# Towards Foundation Models for Zero-Shot Time Series Anomaly Detection: Leveraging Synthetic Data and Relative Context Discrepancy

## Abstract

Time series anomaly detection (TSAD) is a critical task, but developing models that generalize to unseen data in a zero-shot manner remains a major challenge. Prevailing foundation models for TSAD predominantly rely on reconstruction-based objectives, which suffer from a fundamental objective mismatch: they struggle to identify subtle anomalies while often misinterpreting complex normal patterns, leading to high rates of false negatives and positives. To overcome these limitations, we introduce `TimeRCD`, a novel foundation model for TSAD built upon a new pre-training paradigm: Relative Context Discrepancy (RCD). Instead of learning to reconstruct inputs, `TimeRCD` is explicitly trained to identify anomalies by detecting significant discrepancies between adjacent time windows. This relational approach, implemented with a standard Transformer architecture, enables the model to capture contextual shifts indicative of anomalies that reconstruction-based methods often miss. To facilitate this paradigm, we develop a large-scale, diverse synthetic corpus with token-level anomaly labels, providing the rich supervisory signal necessary for effective pre-training. Extensive experiments demonstrate that `TimeRCD` significantly outperforms existing general-purpose and anomaly-specific foundation models in zero-shot TSAD across diverse datasets. Our results validate the superiority of the RCD paradigm and establish a new, effective path toward building robust and generalizable foundation models for time series anomaly detection. The code is available in `https://anonymous.4open.science/r/TimeRCD-5BE1/`

## 1 Introduction

Time series anomaly detection (TSAD) is a crucial task in domains such as finance (Ahmed et al., 2016), healthcare (Kaji et al., 2019), industrial monitoring (Lan et al., 2025), and cloud operations (Ren et al., 2019). The accurate detection of rare and unexpected events is vital for ensuring system reliability and safety. Despite recent progress driven by deep learning, most existing approaches are trained in a dataset- and model-specific manner, which restricts their scalability and hampers generalization across diverse domains in a *zero-shot* way.

The success of foundation models in natural language processing and computer vision has motivated efforts to establish similar paradigms for TSAD. Existing approaches can be broadly categorized into two directions: (i) general-purpose time series foundation models designed for multiple tasks such as classification, forecasting, and anomaly detection (Gao et al., 2024; Goswami et al., 2024; Woo et al., 2024; Ekambaram et al., 2025; Xie et al., 2024), and (ii) anomaly-specific foundation models tailored explicitly for TSAD (Shentu et al., 2024). The more related work discussion is in Appx. B. Despite their differences, both types of mod-

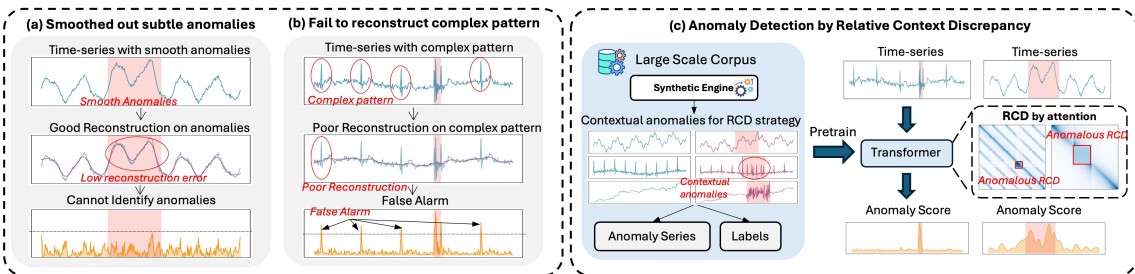

Figure 1: **Limitations of Reconstruction-based TSAD and Our Proposed TimeRCD.** (a) The model accurately reconstructs a smooth anomaly, resulting in a low error score and a missed detection (false negative). (b) The model fails to reconstruct a complex normal pattern unseen in the training dataset, leading to a high error score and a false alarm (false positive). **(c)** TimeRCD adopts RCD with a large-scale corpus and a standard Transformer-Encoder.

els predominantly rely on reconstruction-based objectives trained on real-world data, where anomalies are inferred indirectly from deviations between reconstructed and observed sequences.

While intuitive, reconstruction-based methods suffer from a fundamental **objective mismatch**: they are optimized to reconstruct normal patterns in a latent space where anomalous structure is assumed to be lost (Wong et al., 2022). This leads to critical limitations, as illustrated in Figure 1(a,b). First, subtle and contextual anomalies are often smoothed out and thus missed, resulting in low reconstruction error and false negatives (Wong et al., 2022; Wu et al.). Second, complex but normal sequences deviate from the "average" patterns learned during training, yielding high reconstruction errors and false alarms (Yahya et al., 2025). These weaknesses are further exacerbated in zero-shot settings, where unseen yet normal patterns make reconstruction-based scores particularly unreliable. Our experiments confirm this gap: reconstruction-based foundation models achieve only 15.4% Standard-F1 on contextual anomalies, compared to 82.7% for our RCD approach (Section 3.3.)

Additional fragilities arise from the limitations of real-world training data. First, labeled anomalies are inherently **scarce**, providing the model with few examples to learn abnormal behavior. Second, training data often lacks **diversity**, covering only a subset of real-world patterns. In zero-shot scenarios, these data limitations leave models unexposed to many unseen normal and abnormal sequences, hindering their ability to generalize and detect novel anomalies. To mitigate these issues, some approaches have focused on data augmentation, where artificial anomalies are injected into real-world time series to enrich the training set (Shentu et al., 2024; Darban et al., 2025; Cai et al., 2024). However, these methods are still fundamentally dependent on the availability and diversity of the underlying real data they seek to enhance. Consequently, in zero-shot and cross-domain scenarios, these data limitations leave models unexposed to many unseen normal and abnormal sequences, hindering their ability to generalize and detect novel anomalies.

To address these limitations, we introduce `TimeRCD`, a novel foundation model for TSAD built on a new pre-training paradigm, as illustrated in Figure 1(c). Our approach abandons indirect reconstruction-based objectives in favor of explicitly learning to detect anomalies through **Relative Context Discrepancy** (RCD). The fundamental insight behind RCD is that many anomalies, particularly subtle or contextual ones, are best identified not in isolation but as a significant discrepancy between the patterns of adjacent time windows. By capturing these relational differences, our model can detect shifts that single-window analysis would otherwise miss.

For our pre-training process, we employ a standard Transformer backbone without any architectural modifications. We treat each time window of a time series as an input token, which allows the self-attention

mechanism to naturally compute the inter-token relationships. As shown in Figure 1(c), the model's attention weights learn to capture the discrepancy between contexts, effectively identifying anomalous RCD. An anomaly scoring head then uses these learned discriminative features to produce a final score. To teach the model this explicit detection strategy, we provide it with a rich, supervised signal by first leveraging a synthetic engine. This engine generates a large-scale, diverse, and fully-labeled corpus of time series data that is specifically designed to contain a wide variety of contextual anomalies, enabling the model to learn the RCD task from the ground up. Our main contribution are threefold:

- **The Relative Context Discrepancy (RCD) strategy and the TimeRCD model** We introduce a novel pre-training paradigm for time series anomaly detection that moves beyond reconstruction by explicitly learning to identify anomalies through RCD. This strategy is instantiated in our foundation model, TimeRCD, which uses a standard Transformer to capture relational differences between time windows. This design achieves strong zero-shot generalization through a simple yet powerful architecture.

- **A large-scale, fully-labeled synthetic corpus for foundation models** To enable the RCD pre-training paradigm and its rigorous evaluation, we construct a comprehensive synthetic corpus. It provides token-level annotations for a diverse spectrum of anomalies, including point, contextual, and collective types with cross-variate propagation, offering the essential supervision for building and evaluating zero-shot TSAD models on this corpus.

- **Extensive empirical evaluation** Experiments on diverse corpora demonstrate consistent gains over existing reconstruction-based and general-purpose time-series foundation models. Ablation studies confirm the contributions of both the synthetic corpus and the RCD framework to zero-shot performance.

## 2 METHODOLOGY: THE TIMERCD FRAMEWORK

In this section, we first formally define the zero-shot time series anomaly detection task. We then present the RCD strategy and the foundation model architecture (Section 2.1), which employs an encoder-only Transformer. Finally, we introduce our synthetic data generation engine (Section 2.2), producing a rich, diverse, and precisely annotated training corpus.

**Problem Definition** For the zero-shot time series anomaly detection problem, we observe a multivariate $d$-channel time series $\mathbf{X} = (\mathbf{x}_1, \ldots, \mathbf{x}_n)$ with $\mathbf{x}_t \in \mathbb{R}^d$ for each time step $t \in [n] := \{1, 2, \ldots, n\}$. The objective is to produce a binary annotation sequence $\hat{\mathbf{y}} = (\hat{y}_1, \ldots, \hat{y}_n) \in \{0, 1\}^n$ such that $\hat{y}_t = 1$ if and only if time $t$ is anomalous. In the zero-shot setting, the model must detect anomalies on unseen target sequences without any additional training or fine-tuning, distinguishing normal from anomalous behavior.

### 2.1 RCD STRATEGY AND FOUNDATION MODEL ARCHITECTURE

**Relative Context Discrepancy** We introduce RCD to redefine zero-shot anomaly detection in time series. Rather than learning discriminative mappings from individual samples to labels, RCD formulates detection as comparing a set of time windows to extract discriminative relational patterns. This approach derives anomaly scores from relational comparisons, enabling the model to detect subtle and comparative anomalies in unseen sequences under the zero-shot setting. Detected mathematical formulation of RCD is provided in Appx. C.

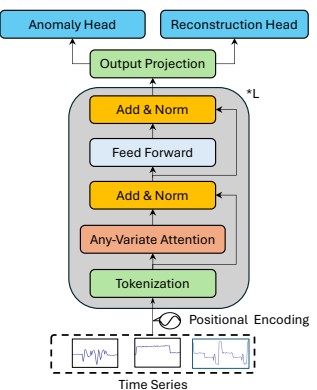

Figure 2: TimeRCD Architecture

Concretely, when each time window is treated as an input token, the Transformer's self-attention effectively implements RCD: attention weights naturally capture relational discrepancies among windows within the sequence context. This demonstrates that RCD-based anomaly detection can be directly realized using standard Transformer blocks. Building on this, we propose a novel foundation model, TimeRCD, illustrated in Fig. 2, which adopts an encoder-only Transformer (Vaswani et al., 2017) with input tokenization, Transformer blocks, and output projection. Crucially, our approach leverages the existing Transformer architecture directly, requiring no structural modifications, which underscores both its simplicity and broad applicability.

**Variate-Window Tokenization**  We adopt the common practice (Nie et al., 2022) and treat a window of continuous observations as an input token. Since multivariate anomaly detection is a critical task (Zamanzadeh Darban et al., 2024), we build on the design introduced by Moirai (Woo et al., 2024), which flattens multivariate time series so that all variates are represented within a single sequence. This design allows the subsequent Transformer blocks to capture both intra-variate dependencies and inter-variate dependencies. Specifically, given a normalized multivariate time series $\tilde{\mathbf{X}} \in \mathbb{R}^{n \times d}$, we partition it into $\lceil n/W \rceil \times d$ non-overlapping windows, where $W$ denotes the window length. The resulting windows are are then flattened andlinearly projected into input token embeddings $\mathbf{H}_{inp} \in \mathbb{R}^{\lceil n/W \rceil d \times D_v}$.

**Transformer Blocks**  We stack $L$ transformer blocks, each consisting of layer normalization, feed-forward network, and self-attention modules. The architecture imposes no special requirements on the Transformer block itself; however, the self-attention mechanism is repurposed to compute the RCD. Each token attends to other tokens, capturing inter-tokens differences that serve as contrastive features for subsequent anomaly score computation. Specifically, we leverage the any-variate attention (Woo et al., 2024) and formulate the output token embeddings as $\mathbf{H}_{out} \in \mathbb{R}^{\lceil n/\tilde{W} \rceil d \times D_v}$.

**Output Projection and Anomaly/Reconstruction Head**  To derive anomaly scores at the original temporal resolution, the output token embeddings $\mathbf{H}_{out}$ from the Transformer blocks are projected back into the observation space. Concretely, we define two heads during training: $\mathbf{X}_{rec} = \mathbf{H}_{out}\mathbf{W}_s\mathbf{W}_{rec}$ and $\mathbf{X}_{ano} = \mathbf{H}_{out}\mathbf{W}_s\mathbf{W}_{ano}$, where $\mathbf{W}_s \in \mathbf{R}^{D_v \times D_v}$ is the shared embedding projection, $\mathbf{W}_{rec} \in \mathbf{R}^{D_v \times W}$ and $\mathbf{W}_{ano} \in \mathbf{R}^{D_v \times W}$ are the reconstruction and anomaly projection, respectively. The reconstruction head predicts masked portions of the input series. To foster robust contextual learning, we employ a **patched masking strategy** where continuous segments of the input time series are masked (with a 15% ratio), as opposed to random point masking. This design prevents information leakage from adjacent timestamps and compels the model to reconstruct missing patterns by aggregating global context. Concurrently, the anomaly head outputs window-level anomaly scores. We optimize the model using a joint loss function:

$$\mathcal{L} = \|\mathbf{M} \odot (\mathbf{X}_{rec} - \mathbf{X})\|_2^2 + \text{BCE}(\sigma(\mathbf{X}_{ano}), \mathbf{y}),$$

where $\mathbf{M}$ is the binary mask, $\mathbf{y}$ denotes ground-truth labels, and $\sigma$ is the sigmoid function. During inference, no masking is applied. The model processes the full, uncorrupted time series, and the Reconstruction Head is discarded. Only the Anomaly Head is used to compute anomaly scores. Although the reconstruction head is discarded at inference, it acts as a crucial auxiliary task during training, encouraging the transformer blocks to learn rich, stable embeddings beyond pairwise relational discrepancies. This effect is empirically validated in Appx. F.6.

We design a synthetic engine to generate multivariate time series with rich, controllable contextual structures, creating a TSAD benchmark that encourages models to recognize anomalies relative to context. The pipeline proceeds in three hierarchical stages: first, defining univariate contextual patterns (Stage 1); next, integrating them into a multivariate system with causal dependencies (Stage 2); and finally, injecting context-aware anomalies that depend on the system's structure (Stage 3). A summary is shown in Fig. 3, with full details in Appx. D.

## 2.2 SYNTHETIC DATA GENERATION

**Stage 1: Context-Template Generation**  For each channel, we generate normal context data from an additive template: $x_{\mathrm{base}}(t) = T(t) + S(t) + \varepsilon(t), t = 0, 1, \ldots, n - 1$. The trend blends deterministic and stochastic components, i.e., $T(t) = (1 - \rho_T) T_{\mathrm{det}}(t) + \rho_T T_{\mathrm{stoc}}(t)$. Seasonality is a mixture of periodic atoms, i.e., $S(t) = \sum_{k=1}^{K} A_k w_k(2\pi f_k t + \varphi_k; \theta_k)$, where $w_k$ spans sinusoid, square/triangle waves, and wavelet atoms with amplitudes $A_k$, frequencies $f_k$, phases $\varphi_k$, and shape parameters $\theta_k$. Noise is zero-mean with optional piecewise volatility, e.g., $\varepsilon(t) \sim \mathcal{N}(0, \sigma^2(t))$ and $\sigma(t)$ allowing bursty segments. The full description for constructing trend and seasonality is in Appx. D.2 and D.3.

**Stage 2: Joint-Context Fusion**  To simulate realistic inter-channel dependencies, we fuse per-channel contexts into a coherent multivariate system via a causal graphical model. We first sample a DAG $G = (V, E)$ over $N$ channels (nodes) and define a latent causal process $z_i$ for each node. Discretizing first-order ODE dynamics by Euler ($\Delta t = 1$) gives an ARX system: $z_i[t] = a_i z_i[t - 1] + \sum_{j \in P(i)} b_{ij} x_j[t - \ell_{ij}] + c_i, |a_i| \leq 0.8$, with parent set $P(i)$, lags $\ell_{ij}$, gains $b_{ij}$, and bias $c_i$. The observed signal mixes baseline and causal channels $x_i[t] = (1 - \alpha_i) x_{\mathrm{base},i}(t) + \alpha_i z_i[t], \alpha_i \in [0, 1]$.

**Stage 3: Causal-contextual Anomaly Injection**
Most existing works inject anomalies via simple pointwise corruption or handcrafted signal perturbations—procedures that ignore both temporal context and cross-variate dependencies. We term this class **exogenous injections**, where a multivariate normal system is first generated and then a window in one channel is overwritten, i.e., $x'_{\mathrm{anom},i}[t] = x_i[t] + \Delta(t)$ for $t \in [t_s, t_e)$. Beyond this, we introduce a novel **endogenous injection mechanism** that intervenes prior to causal mixing. Specifically, we perturb the baseline signal $x_{\mathrm{base},i}$ of a parent node, i.e. $x'_{\mathrm{base},i}(t) = x_{\mathrm{base},i}(t) + \Delta(t)$ for $t \in [t_s, t_e)$, allowing abnormal effects to propagate to downstream nodes $x'_{\mathrm{anom},j}[t]$ organically via the ARX dynamics. This process emulates internal system failures and yields multivariate anomalies with coherent temporal and structural footprints. We design more than 20 types of anomalies for $\Delta(t)$ and also consider a type of contextual anomalies which modifies the periodic structure, e.g., replace $S(t)$

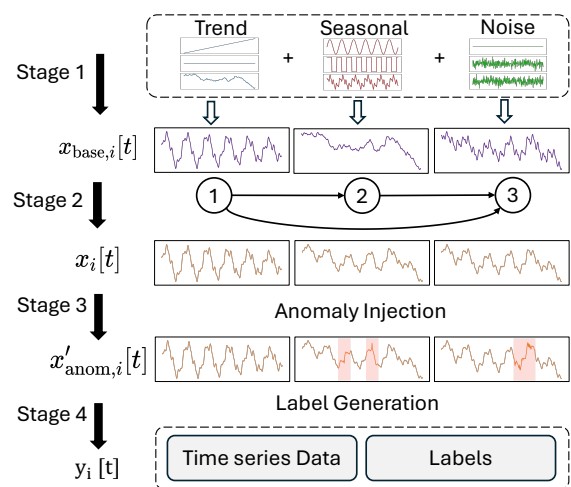

Figure 3: Synthetic data generation procedure.

with $S'(t)$ to realize frequency or phase shifts within $[t_s, t_e)$. The full taxonomy and parameterizations are detailed in Appx. D.4.

**Stage 4: Labels and Masks**  We generate token-level binary labels. For exogenous injections, positive labels correspond to the intervention window. For endogenous injections, we label the root-cause window and extend it to descendant channels based on their causal lags, as implied by the DAG and ARX lags $\{\ell_{ij}\}$. While our generation process tracks channel-specific anomalies (root-cause vs. propagated effects), for this work, we aggregate these into temporal localization labels (marking when an anomaly occurs across any channel). This aligns with standard zero-shot anomaly detection benchmarks, which typically provide timestep-level but not channel-level ground truth. Sequence lengths, DAG sparsity, ARX coefficients, and

signal regimes (trend, seasonality, noise) are sampled from configurable priors (Appx. D.1), generating rich, interpretable dynamics designed for zero-shot learning.

**Relative Context Discrepancy** This layered generation process is designed to compel a model to learn RCD. Rather than identifying anomalies via absolute thresholds or isolated patterns, a model must determine whether a data point is abnormal relative to its context. This context is multifaceted, including a variable's own temporal dynamics (e.g., trend, seasonality), its causal relationships with other variables, and the characteristics of the anomaly itself. Our **endogenous anomaly** injection mechanism exemplifies this challenge: a deviation in a downstream "child" variable may be a propagated effect of an upstream "parent" failure. Correctly identifying only the root-cause anomaly requires the model to evaluate multiple variables in light of their causal dependencies. This encourages reasoning beyond simple pattern recognition toward a relational, system-level understanding—the essence of assessing relative discrepancy.

## 3 EXPERIMENTS

To validate the effectiveness of TimeRCD, we design a comprehensive evaluation to answer three research questions: **RQ1**: How well does TimeRCD perform in strict zero-shot anomaly detection compared with existing time-series foundation models and with full-shot, dataset-specific baselines? **RQ2**: How does the RCD-based strategy exploit long contextual windows, and what is its impact on detecting contextual anomalies and on window-size sensitivity? **RQ3**: How do our synthetic generator and injection design affect performance, and how does accuracy scale with pre-training data size?

### 3.1 EXPERIMENTAL SETTINGS

**Datasets** Our evaluation is conducted on a comprehensive suite of 16 public time-series anomaly detection datasets, covering a wide range of real-world and synthetic scenarios. Details about the benchmark datasets can be found in the Appx. E.1.

**Baselines** We benchmark TimeRCD against methods from two primary settings: (1) **Zero-shot models**, which include our approach and other foundation models (DADA† (Shentu et al., 2024), TS-Pulse (Ekambaram et al., 2025), MOMENT† (Goswami et al., 2024), TimesFM (Das et al., 2024), Chronos (Ansari et al., 2024), Time MOE (Shi et al., 2024)). (2) **Full-shot models**, which are fitted on a per-dataset basis. This category includes deep learning methods (TranAD (Tuli et al., 2022), USAD (Audibert et al., 2020), OmniAnomaly (Su et al., 2019), Sub-PCA (Liu & Paparrizos, 2024), DCdetector (Yang et al., 2023), TF-MAE (Fang et al., 2024)) and classical statistical algorithms (LOF (Breunig et al., 2000), IForest (Liu et al., 2008)). Note that models marked with (†) were excluded where necessary due to potential data leakage under zero-shot setting (Appx. E.3). Additionally details about all of the baseslines can be found at the Appx. E.2.

**Evaluation Protocol** We evaluate model performance using four standard metrics: Affiliation-F1, F1-T, Standard-F1, and VUS-PR. We deliberately avoid using the Point-Adjusted F1 score, as recent work has demonstrated that Point Adjustment (PA) can lead to inflated and misleading performance evaluations (Huet et al., 2022; Wang et al., 2023a). More details about the metrics are shown in the Appx. E.4.

### 3.2 MAIN RESULTS: TSAD ACCURACY (RQ1)

Our evaluation includes two comparisons: a direct **zero-shot** test against foundation models, and a **full-shot** test against baselines trained on target data. We stress that **TimeRCD** is strictly zero-shot in all settings, testing true out-of-the-box performance. Results are shown in Table 1. In zero-shot comparisons, TimeRCD

achieves clear SOTA: on 64 evaluation cases (16 datasets × 4 metrics) it ranks first in **41** and second in **6**. Even against full-shot baselines with access to target data, TimeRCD is highly competitive, ranking first in **32** and second in **3**. This strong showing highlights the power and generalizability of our pre-training framework.

Table 1: Performance of **TimeRCD** against zero-shot and full-shot baselines. TimeRCD operates in a strictly zero-shot capacity in all comparisons. Best result is in **red**, second-best is in blue. Asterisked (*) results are excluded from ranking due to data leaking.

| Metric | Model | IOPS | MGAB | NAB | NEK | Power | SED | Stock | TODS | UCR | WSD | YAHOO | MSL | PSM | SMAP | SMD | SWaT | Total 1st | Total 2nd |
|---|---|---|---|---|---|---|---|---|---|---|---|---|---|---|---|---|---|---|---|
| | | | | | | Univariate Datasets | | | | | | | Multivariate Datasets | | | | | | |
| | | | | | | **Zero-Shot Models** | | | | | | | | | | | | | |
| Affiliation-F | TimeRCD | 83.28 | 70.69 | 82.48 | 79.73 | 85.51 | 96.87 | 71.84 | 86.37 | 84.63 | 90.33 | 96.65 | 81.16 | 81.61 | 87.73 | 92.58 | 71.55 | 09 | 02 |
| | DADA† | 89.37* | 67.66* | 86.56 | 95.40 | 69.79 | 65.18 | 98.77 | 76.89 | 72.21 | 93.92 | 92.20* | 76.57 | 81.27 | 76.92 | 83.74 | 76.18 | 03 | 06 |
| | TS-Pulse | 68.76 | 67.33 | 70.80 | 73.05 | 69.94 | 67.44 | 67.93 | 67.90 | 67.70 | 68.22 | 70.05 | 70.14 | 70.28 | 69.21 | 68.21 | 71.18 | 00 | 00 |
| | MOMENT† | 87.54* | 66.76* | 90.45* | 92.26 | 75.97 | 59.13 | 45.26 | 59.76 | 75.77 | 95.39 | 79.99* | 74.55* | 65.79 | 77.42* | 74.00* | 70.17 | 01 | 02 |
| | TimesFM | 81.88 | 66.95 | 79.73 | 90.49 | 69.88 | 67.14 | 97.53 | 89.08 | 70.03 | 78.97 | 91.28 | 20.35 | 71.24 | 45.44 | 62.85 | 44.37 | 00 | 02 |
| | Chronos | 90.12 | 67.89 | 86.66 | 93.63 | 69.72 | 67.89 | 96.85 | 91.96 | 74.35 | 90.98 | 96.34 | 75.52 | 70.88 | 72.22 | 75.31 | 70.43 | 03 | 04 |
| | Time MOE | 76.34 | 67.23 | 80.51 | 80.50 | 71.19 | 60.98 | 63.28 | 54.68 | 73.56 | 80.25 | 69.70 | 69.85 | 54.74 | 74.38 | 69.97 | 64.37 | 00 | 00 |
| F1-T | TimeRCD | 28.44 | 1.81 | 38.85 | 35.87 | 28.47 | 69.43 | 31.73 | 65.89 | 34.30 | 35.04 | 85.86 | 42.47 | 37.98 | 33.74 | 53.91 | 30.28 | 11 | 01 |
| | DADA† | 42.50* | 0.91* | 37.24 | 47.98 | 19.80 | 9.56 | 95.49 | 35.18 | 7.22 | 48.46 | 79.52* | 34.58 | 31.84 | 30.42 | 40.80 | 35.13 | 03 | 05 |
| | TS-Pulse | 4.10 | 0.81 | 34.61 | 27.07 | 19.90 | 9.71 | 15.98 | 13.45 | 5.12 | 4.57 | 5.50 | 23.57 | 25.39 | 12.34 | 9.15 | 28.58 | 00 | 00 |
| | MOMENT† | 33.15* | 0.80* | 52.27* | 63.66 | 19.91 | 9.54 | 18.04 | 17.47 | 13.02 | 41.98 | 11.69* | 25.97* | 27.77 | 17.93* | 28.68* | 28.76 | 01 | 03 |
| | TimesFM | 48.95 | 0.93 | 36.74 | 36.63 | 19.80 | 9.58 | 88.94 | 51.13 | 10.78 | 41.38 | 83.46 | 7.83 | 25.42 | 11.64 | 18.65 | 21.39 | 01 | 01 |
| | Chronos | 45.45 | 1.10 | 36.10 | 33.16 | 19.90 | 13.18 | 89.30 | 53.90 | 10.88 | 39.82 | 79.00 | 15.59 | 25.42 | 11.72 | 17.32 | 28.88 | 00 | 04 |
| | Time MOE | 25.95 | 0.63 | 38.70 | 15.78 | 19.85 | 17.73 | 34.13 | 20.91 | 8.29 | 22.60 | 37.11 | 23.92 | 26.82 | 14.22 | 19.90 | 30.11 | 00 | 02 |
| Standard-F1 | TimeRCD | 24.22 | 1.62 | 27.70 | 33.05 | 28.59 | 69.88 | 32.61 | 67.02 | 28.13 | 31.96 | 87.02 | 30.66 | 26.00 | 30.48 | 44.89 | 28.73 | 11 | 01 |
| | DADA† | 32.76* | 0.80* | 26.91 | 48.24 | 15.99 | 2.69 | 95.59 | 28.18 | 3.36 | 45.06 | 79.30* | 22.13 | 24.07 | 26.75 | 34.98 | 34.78 | 03 | 05 |
| | TS-Pulse | 3.54 | 0.73 | 21.61 | 23.96 | 18.27 | 8.84 | 15.46 | 12.45 | 2.05 | 2.17 | 4.00 | 12.56 | 22.31 | 7.44 | 8.00 | 23.84 | 00 | 01 |
| | MOMENT† | 30.69* | 0.67* | 44.75* | 63.85 | 16.39 | 3.36 | 19.38 | 14.64 | 9.00 | 41.42 | 10.54* | 14.43* | 23.83 | 12.92* | 29.78* | 21.30 | 01 | 02 |
| | TimesFM | 34.28 | 0.83 | 26.46 | 38.15 | 16.73 | 2.96 | 89.13 | 40.08 | 7.86 | 38.50 | 84.44 | 5.75 | 22.18 | 10.46 | 18.65 | 22.84 | 01 | 01 |
| | Chronos | 32.69 | 0.99 | 26.22 | 33.54 | 17.47 | 8.74 | 89.41 | 40.52 | 8.21 | 34.58 | 78.89 | 11.63 | 22.27 | 9.62 | 17.50 | 24.03 | 00 | 04 |
| | Time MOE | 26.52 | 0.45 | 26.20 | 11.47 | 12.16 | 17.73 | 34.32 | 16.38 | 4.09 | 20.09 | 27.50 | 12.85 | 24.80 | 9.01 | 21.62 | 23.58 | 00 | 02 |
| VUS-PR | TimeRCD | 20.23 | 1.05 | 24.32 | 27.88 | 21.25 | 80.75 | 77.28 | 93.46 | 23.09 | 21.77 | 84.41 | 20.45 | 18.69 | 22.68 | 37.03 | 17.58 | 10 | 02 |
| | DADA† | 24.97* | 0.56* | 24.73 | 46.85 | 10.61 | 6.42 | 99.51 | 64.83 | 2.94 | 33.42 | 70.74* | 12.74 | 17.17 | 20.02 | 25.98 | 21.13 | 03 | 06 |
| | TS-Pulse | 4.64 | 0.56 | 16.40 | 19.39 | 11.72 | 9.11 | 70.95 | 45.86 | 1.20 | 1.83 | 9.93 | 7.41 | 14.48 | 3.99 | 4.56 | 15.67 | 00 | 01 |
| | MOMENT† | 37.35* | 0.56* | 45.38* | 67.74 | 10.50 | 4.31 | 76.97 | 56.45 | 6.17 | 15.26 | 30.81* | 9.32* | 16.48 | 8.97* | 15.96* | 14.90 | 02 | 00 |
| | TimesFM | 19.56 | 0.58 | 24.01 | 35.02 | 10.44 | 6.13 | 98.39 | 72.89 | 6.03 | 21.57 | 86.78 | 11.84 | 14.76 | 16.95 | 13.02 | 19.43 | 01 | 04 |
| | Chronos | 19.00 | 0.60 | 23.76 | 31.80 | 10.95 | 8.65 | 97.49 | 70.66 | 6.56 | 18.81 | 83.54 | 8.25 | 14.61 | 5.18 | 10.22 | 16.44 | 00 | 02 |
| | Time MOE | 16.63 | 0.52 | 22.62 | 19.76 | 9.34 | 10.87 | 74.78 | 48.78 | 2.10 | 10.93 | 20.90 | 7.82 | 15.68 | 4.98 | 11.12 | 16.20 | 00 | 01 |
| **TimeRCD Grand Total (Zero-Shot)** | | | | | | | | | | | | | | | | | | 41 | 06 |
| | | | | | | **Full-Shot Models** | | | | | | | | | | | | | |
| Affiliation-F | TimeRCD | 83.28 | 70.69 | 82.48 | 79.73 | 85.51 | 96.87 | 71.84 | 86.37 | 84.63 | 90.33 | 96.65 | 81.16 | 81.61 | 87.73 | 92.58 | 71.55 | 10 | 01 |
| | TranAD | 83.19 | 67.28 | 90.28 | 85.02 | 71.56 | 61.03 | 57.94 | 52.76 | 73.31 | 84.34 | 76.08 | 79.91 | 73.83 | 87.39 | 92.20 | 75.37 | 00 | 04 |
| | USAD | 71.08 | 67.81 | 91.54 | 71.13 | 76.48 | 55.60 | 35.92 | 47.90 | 76.00 | 65.10 | 53.05 | 81.86 | 57.86 | 87.25 | 85.09 | 75.06 | 00 | 01 |
| | OmniAnomaly | 80.32 | 67.35 | 92.35 | 86.30 | 78.16 | 61.26 | 75.24 | 50.73 | 73.53 | 78.02 | 71.31 | 83.15 | 58.17 | 91.38 | 85.82 | 73.39 | 03 | 02 |
| | LOF | 81.06 | 68.44 | 75.75 | 84.74 | 66.76 | 63.85 | 69.74 | 60.58 | 73.53 | 81.29 | 75.63 | 84.35 | 61.98 | 63.32 | 64.13 | 56.34 | 01 | 00 |
| | IForest | 52.81 | 68.82 | 39.84 | 71.15 | 0.00 | 70.09 | 0.06 | 44.17 | 50.56 | 41.24 | 33.30 | 63.36 | 63.78 | 59.96 | 69.71 | 0.00 | 00 | 01 |
| | Sub-PCA | 75.39 | 66.90 | 89.29 | 97.10 | 71.37 | 67.14 | 70.63 | 72.75 | 76.66 | 76.45 | 75.85 | 84.25 | 71.49 | 90.08 | 85.80 | 76.29 | 02 | 04 |
| | DCdetector | 71.83 | 67.91 | 72.21 | 62.31 | 69.75 | 72.20 | 57.81 | 70.18 | 72.79 | 67.77 | 67.74 | 67.32 | 67.10 | 69.55 | 71.07 | 00 | 01 |
| | TFMAE | 78.25 | 67.50 | 75.99 | 76.91 | 70.30 | 68.17 | 56.39 | 62.83 | 70.60 | 80.25 | 76.87 | 75.70 | 70.07 | 75.36 | 70.85 | 75.72 | 00 | 02 |
| F1-T | TimeRCD | 28.44 | 1.81 | 38.85 | 35.87 | 28.47 | 69.43 | 31.73 | 65.89 | 34.30 | 35.04 | 85.86 | 42.47 | 37.98 | 33.74 | 53.91 | 30.28 | 08 | 01 |
| | TranAD | 22.63 | 1.65 | 37.28 | 69.97 | 22.36 | 9.57 | 16.73 | 13.51 | 7.75 | 20.94 | 8.41 | 39.42 | 25.49 | 29.12 | 37.98 | 49.58 | 00 | 01 |
| | USAD | 20.99 | 4.07 | 61.46 | 70.64 | 28.23 | 9.54 | 16.86 | 20.85 | 14.63 | 14.18 | 9.35 | 48.71 | 28.96 | 43.94 | 50.41 | 50.41 | 03 | 01 |
| | OmniAnomaly | 51.17 | 1.61 | 40.09 | 82.20 | 23.48 | 9.68 | 36.22 | 14.33 | 8.47 | 34.79 | 24.16 | 49.36 | 30.42 | 46.63 | 51.84 | 46.64 | 03 | 04 |
| | LOF | 27.97 | 1.15 | 35.76 | 63.57 | 19.80 | 9.60 | 66.14 | 31.63 | 8.31 | 24.38 | 55.93 | 38.97 | 25.58 | 21.81 | 10.13 | 30.62 | 01 | 02 |
| | IForest | 7.64 | 0.84 | 21.44 | 65.56 | 0.00 | 9.54 | 1.10 | 11.06 | 6.36 | 4.28 | 4.90 | 20.73 | 25.39 | 14.32 | 16.20 | 0.00 | 00 | 00 |
| | Sub-PCA | 32.75 | 0.98 | 54.15 | 84.18 | 20.30 | 9.54 | 20.10 | 18.40 | 18.90 | 24.62 | 11.57 | 49.02 | 30.38 | 44.31 | 51.87 | 46.65 | 01 | 06 |
| | DCdetector | 6.61 | 1.32 | 32.72 | 29.21 | 21.13 | 10.53 | 16.07 | 16.40 | 6.62 | 7.32 | 6.81 | 23.24 | 25.34 | 15.73 | 9.47 | 28.64 | 00 | 00 |
| | TFMAE | 19.41 | 1.07 | 33.04 | 31.11 | 20.18 | 11.82 | 22.15 | 16.48 | 5.90 | 19.11 | 23.85 | 25.28 | 25.36 | 19.39 | 10.13 | 28.46 | 00 | 01 |
| Standard-F1 | TimeRCD | 24.22 | 1.62 | 27.70 | 33.05 | 28.59 | 69.88 | 32.61 | 67.02 | 28.13 | 31.96 | 87.02 | 30.66 | 26.00 | 30.48 | 44.89 | 28.73 | 06 | 01 |
| | TranAD | 34.85 | 1.46 | 27.33 | 60.36 | 22.36 | 2.63 | 16.23 | 11.94 | 4.40 | 20.23 | 5.70 | 29.60 | 25.63 | 25.11 | 43.99 | 61.86 | 00 | 02 |
| | USAD | 30.66 | 3.89 | 56.15 | 62.91 | 28.24 | 3.41 | 17.99 | 23.87 | 10.74 | 13.20 | 7.21 | 38.71 | 28.41 | 38.66 | 53.06 | 62.82 | 03 | 02 |
| | OmniAnomaly | 47.05 | 1.44 | 28.81 | 74.03 | 23.50 | 0.43 | 38.59 | 12.65 | 5.11 | 29.57 | 21.40 | 39.10 | 30.43 | 40.50 | 57.06 | 55.93 | 04 | 04 |
| | LOF | 30.28 | 1.05 | 24.04 | 56.92 | 12.18 | 4.11 | 66.20 | 25.77 | 4.70 | 22.62 | 48.95 | 30.65 | 18.80 | 18.70 | 8.41 | 29.08 | 01 | 02 |
| | IForest | 8.37 | 0.73 | 29.41 | 58.10 | 19.77 | 3.81 | 16.91 | 13.35 | 4.09 | 2.07 | 3.20 | 14.68 | 24.15 | 13.61 | 16.89 | 26.76 | 00 | 00 |
| | Sub-PCA | 33.96 | 0.83 | 46.71 | 85.43 | 16.05 | 9.56 | 21.77 | 18.72 | 15.12 | 24.74 | 11.06 | 38.29 | 30.26 | 38.90 | 57.22 | 55.98 | 02 | 00 |
| | DCdetector | 5.19 | 1.21 | 24.02 | 17.37 | 21.10 | 10.54 | 16.97 | 17.85 | 3.18 | 4.64 | 4.14 | 14.08 | 25.33 | 10.67 | 8.99 | 27.02 | 00 | 00 |
| | TFMAE | 9.48 | 0.97 | 23.78 | 19.74 | 20.14 | 11.87 | 22.34 | 14.88 | 2.63 | 14.88 | 12.58 | 15.68 | 25.39 | 12.58 | 9.16 | 27.08 | 00 | 01 |
| VUS-PR | TimeRCD | 20.23 | 1.05 | 24.32 | 27.88 | 21.25 | 80.75 | 77.28 | 93.46 | 23.09 | 21.77 | 84.41 | 20.45 | 18.69 | 22.68 | 37.03 | 17.58 | 08 | 00 |
| | TranAD | 21.61 | 0.64 | 24.82 | 61.63 | 13.04 | 5.75 | 78.08 | 47.33 | 2.25 | 12.20 | 25.78 | 14.78 | 16.49 | 13.37 | 28.34 | 47.37 | 01 | 01 |
| | USAD | 16.58 | 0.75 | 55.03 | 58.53 | 18.68 | 4.37 | 74.53 | 56.36 | 8.85 | 10.00 | 14.15 | 29.95 | 17.59 | 26.37 | 34.53 | 44.73 | 01 | 04 |
| | OmniAnomaly | 25.35 | 0.64 | 27.17 | 74.51 | 14.32 | 6.20 | 48.55 | 45.55 | 2.40 | 16.37 | 29.26 | 31.57 | 18.58 | 28.07 | 37.44 | 42.97 | 04 | 03 |
| | LOF | 19.43 | 0.57 | 21.18 | 58.52 | 9.31 | 6.81 | 83.07 | 49.14 | 2.39 | 12.85 | 41.37 | 24.67 | 13.58 | 10.59 | 4.40 | 14.50 | 00 | 02 |
| | IForest | 8.59 | 0.62 | 23.57 | 56.50 | 11.56 | 7.71 | 70.99 | 46.62 | 2.88 | 2.06 | 10.47 | 11.29 | 15.85 | 7.55 | 8.88 | 15.49 | 00 | 00 |
| | Sub-PCA | 23.02 | 0.60 | 46.08 | 88.91 | 10.49 | 3.72 | 80.86 | 54.16 | 12.92 | 16.41 | 21.57 | 31.43 | 18.52 | 26.42 | 37.50 | 43.02 | 02 | 06 |
| | DCdetector | 5.83 | 0.59 | 16.60 | 14.03 | 12.32 | 9.37 | 74.16 | 46.66 | 1.53 | 3.23 | 10.17 | 7.01 | 14.49 | 4.21 | 4.66 | 15.04 | 00 | 00 |
| | TFMAE | 5.32 | 0.64 | 15.68 | 17.81 | 11.90 | 9.55 | 73.54 | 48.79 | 2.57 | 5.36 | 25.93 | 8.25 | 14.22 | 5.76 | 4.77 | 15.38 | 00 | 01 |
| **TimeRCD Grand Total (Full-Shot)** | | | | | | | | | | | | | | | | | | 32 | 03 |

## 3.3 RCD STRATEGY EFFICIENCY (RQ2)

**Qualitative Analysis of Contextual Understanding** A key architectural feature of TimeRCD is its ability to process long context windows, allowing it to learn complex temporal dependencies. Many existing

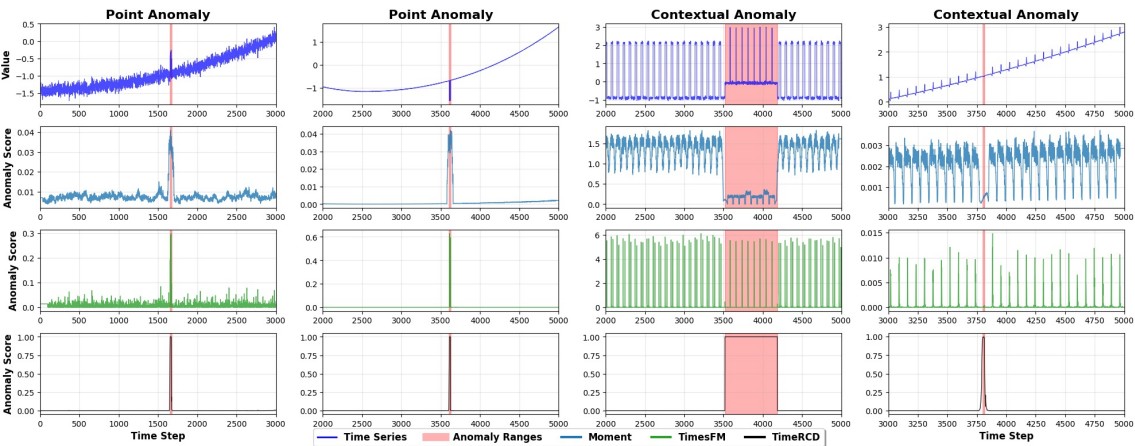

Figure 4: Qualitative comparison of anomaly scores.

zero-shot methods, particularly those based on reconstruction with small look-up windows, are effective at detecting abrupt **point anomalies**—short-term deviations from an immediate pattern (two left charts in Fig. 4). However, these models often fail on subtle **contextual**, where the anomalous behavior is a deviation from a long-term pattern (two right charts in Fig. 4). Their limited context prevents them from distinguishing normal long-term variations from true anomalous segments. In contrast, as our qualitative results in Fig. 4 show, TimeRCD's ability to view the entire series allows it to learn the complex relationships between distant points.

**Quantitative Analysis of Contextual Understanding** We create specialized, unseen datasets containing either purely point or contextual anomalies, ensuring a fair zero-shot evaluation (details in Appx. F.2). The results are in Fig. 5. While TimeRCD's performance on point anomalies is highly competitive with other top zero-shot models, it is substantially superior on contextual anomalies. On this task, our model achieves a `Standard-F1` of 0.827, whereas all other models suffer a significant performance collapse. This performance disparity provides strong evidence that TimeRCD's ability to leverage long-range context is a key capability, allowing it to detect complex deviations that are challenging for methods with a more limited contextual view.

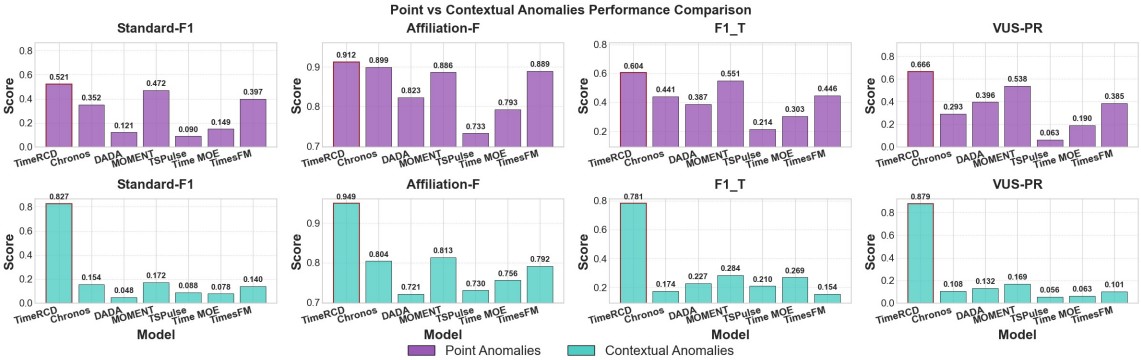

Figure 5: Comparison on specially-created datasets containing either point or contextual anomalies.

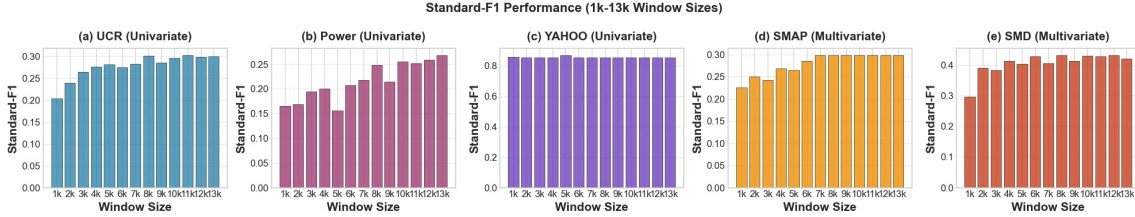

Figure 6: Performance on univariate (top) and multivariate (bottom) datasets as a function of the input window size, varied from 1k to 13k.

**Impact of Context Window Size** TimeRCD's ability to process variable context lengths is a core architectural feature. To analyze its impact, we evaluate performance with input window sizes from 1k to 13k. As shown in Fig. 6, the results confirm that the optimal context length is task-dependent. For datasets with long-term patterns like **UCR**, **Power**, **SMAP**, and **SMD** Fig. 6 (a, b, d, e), performance generally improves with a larger window, as this allows the model to establish a more robust baseline of "normal" behavior. Conversely, on inherently short series like **YAHOO** Fig. 6(c), performance remains flat, as the series length itself becomes the effective context limit. A detailed breakdown for all datasets is available in Appx. F.3.

### 3.4 Synthetic Data Efficiency (RQ3)

**Ablation Study on Pre-training Data** To validate our data generation framework, we train on the same TimeRCD (architecture, hyperparameters, epochs) on three 350M-point datasets, each with 350M points to match the real-world data scale: (1) our synthetic data with *in-context* anomaly injection, (2) the same synthetic series but with DADA-injected anomalies (Shentu et al., 2024), and (3) real-world data (Godahewa et al.) augmented with DADA. Table 2 shows weighted averages over 9 univariate benchmarks (full results in Appx. F.4). Models trained on augmented real data performed far worse, confirming the necessity of a high-quality synthetic curriculum. Comparing the two synthetic variants reveals a key trade-off: though using DADA-injection achieves similar Affiliation-F and slightly higher VUS-PR, it causes sharp drops in finer-grained metrics (F1-T ↓6.4%, Standard-F1 ↓6.1%). This indicates our in-context injection generates more challenging and robust training signals.

Table 2: Weighted average performance across 9 univariate benchmarks.

| Pre-Training Dataset | Affiliation-F | F1-T | Standard-F1 | VUS-PR |
|---|---|---|---|---|
| Our Synthetic Data | **0.878** | **0.569** | **0.523** | 0.478 |
| Our Synthetic + DADA Injection | **0.878** | 0.505 | 0.462 | **0.487** |
| Real-world Data + DADA Injection | 0.716 | 0.073 | 0.062 | 0.102 |

**Dataset Scaling** To investigate the effect of pre-training data scale on performance, we train TimeRCD on increasingly larger subsets of our synthetic dataset: 350M, 700M, and the full 2.5B data points. The results, shown as a weighted average across our benchmark datasets in the Fig. 7, demonstrate a clear and positive scaling law. As the amount of pre-training data increases, the model's performance consistently improves across all four evaluation metrics.

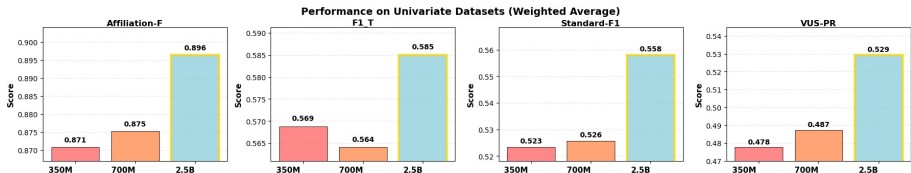

Figure 7: Demonstration of positive scaling laws. The figure shows weighted average performance across our benchmarks when training on datasets of increasing size (350M, 700M, 2.5B).

## 4 CONCLUSION

In conclusion, TimeRCD successfully addresses the objective mismatch in reconstruction-based methods by introducing the principle of Relative Context Discrepancy. This work establishes a new and effective pre-training paradigm for zero-shot TSAD, with its conceptual simplicity and strong empirical results opening several promising avenues for future research. Looking forward, the extensibility of our framework invites investigation into efficiently fine-tuning the pre-trained TimeRCD on domain-specific datasets for critical applications. As a current limitation and future direction, our implementation relies on a standard transformer backbone; exploring novel network structures specifically engineered to more efficiently capture RCD could yield further performance gains and is a promising area for subsequent research.

## ETHICS STATEMENT

In alignment with the ICLR Code of Ethics, this research is committed to upholding the highest standards of academic integrity, social responsibility, and fairness by proactively identifying and mitigating potential ethical risks, ensuring transparency in methodology and limitations, respecting privacy and consent in data usage, promoting inclusivity and non-discrimination, and avoiding any form of harm or misuse, while fostering a constructive and respectful scholarly environment for all participants and the broader community.

## REPRODUCIBILITY STATEMENT

We have made every effort to ensure that the results presented in this paper are reproducible. We ensure reproducibility by fully specifying our synthetic data generator (Appx. D). All evaluation datasets and splits follow the TSB-AD protocol described in Appx. E.1. Training configurations—model architecture, optimization, masking strategy, early stopping, and windowing—are detailed in Appx. E.5. Source code is available in `https://anonymous.4open.science/r/TimeRCD-5BE1/`

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
