## A  USE OF LLMs

In preparing our manuscript, we used large language models (LLMs) only to aid or polish the writing. Their use was restricted to improving grammar, readability, and style, without contributing to the methodology, theoretical results, algorithmic implementation, or experimental outcomes. The involvement of LLMs does not affect the reproducibility of our findings.

## B  RELATED WORK

**State-of-the-art TSAD algorithm**  Traditional unsupervised TSAD operates by training models on domain-specific normal data to identify deviations in unseen test sets. Within this paradigm, two primary approaches have been dominant. Reconstruction-based methods identify anomalies by quantifying the error when attempting to rebuild an input signal, with higher errors indicating anomalous behavior (Audibert et al., 2020; Su et al., 2019; Zhang et al., 2019; Tuli et al., 2022; Nam et al., 2024). In parallel, contrastive-based methods learn discriminative representations by maximizing the distinction between normal and anomalous patterns or distributions (Xu et al., 2021; Yang et al., 2023; Shen et al., 2020; Darban et al., 2025; Lan et al., 2025; Wilson et al., 2023). However, the reliance of these methods on in-domain normal data presents a fundamental barrier to generalization, leading to significant performance degradation in zero-shot scenarios where the model encounters data from new, unseen domains. While the recent emergence of foundation models aims to overcome this transferability issue, the predominant architecture remains anchored in the reconstruction-based framework (Shentu et al., 2024; Gao et al., 2024; Goswami et al., 2024), which suffers from **objective dismatch**. We are the first to develop a foundation model for zero-shot TSAD built upon the principle of relative contextual discrimination, thereby circumventing the limitations of reconstruction-based approaches in zero-shot TSAD.

**Synthetic dataset for Time series analysis**  The paradigm of leveraging synthetic data for pre-training time series models has been extensively validated, particularly in the domain of forecasting, where it has led to significant performance gains across various benchmarks (Xie et al., 2025; Cai et al., 2024; Dooley et al., 2023; Hoo et al., 2025; Taga et al., 2025; Bhethanabhotla et al., 2024). Following this trend, several foundation models for general time series analysis have been successfully trained on a hybrid corpus of real-world and synthetic data to enhance their generalization capabilities (Ansari et al., 2024; Das et al., 2024; Shi et al., 2024; Goswami et al., 2024). In contrast, the application of purely synthetic data for TSAD remains substantially less explored. Existing efforts in this area have primarily focused on data augmentation, where artificial anomalies are injected into real-world time series to enrich the training set (Shentu et al., 2024; Darban et al., 2025). While effective, these methods are still fundamentally dependent on the availability and diversity of the underlying real data. Other studies have discussed the generation of entirely synthetic anomalous time series but have not applied this data to the end-to-end training of a dedicated TSAD model (Cai et al., 2024; Xie et al., 2024). To the best of our knowledge, this work is the first to propose and successfully implement a pre-training framework for a foundation model dedicated to TSAD that relies exclusively on a large-scale, fully synthetic dataset.

## C  MATHEMATICAL FORMULATION OF RELATIVE CONTEXT DISCREPANCY

### C.1  DEFINITION OF RCD

Traditional anomaly-detection methods often evaluate each data point in isolation, for example by measuring its deviation from expected values or by assessing reconstruction errors. *Relative Context Discrepancy (RCD)* frames the problem differently. Rather than focusing on the individual value of a token, it evaluates whether the *relationships* a token forms with its surrounding context remain consistent with normal relational

patterns. Under this view, an anomaly emerges as a disruption in these relational structures, and RCD therefore emphasizes relational inconsistency rather than absolute deviation.

## C.2 MATHEMATICAL FORMULATION OF RCD

Consider a time-series sequence

$$X_1, X_2, \ldots, X_T,$$

where each $X_t$ denotes the observation at time $t$. RCD does not describe $X_t$ by its value alone, but by how it interacts with the other tokens in the sequence.

To formalize this, introduce two functions. The first, $f(X_j, X_t)$, is a pairwise interaction function representing how token $X_j$ contributes to the interpretation of token $X_t$. This function captures the *relative* aspect of RCD by explicitly modeling token-to-token relationships:

$$f(X_j, X_t).$$

Anomalies, however, typically arise not from a single unusual interaction, but from an *unusual pattern* of interactions. This motivates an aggregation function $g$ that synthesizes all pairwise contributions into a contextual embedding for token $t$:

$$\text{embedding}_{X_t} = g\big(f(X_1, X_t), f(X_2, X_t), \ldots, f(X_T, X_t)\big).$$

This embedding summarizes the relational profile that determines how token $t$ fits into its broader temporal context. Under this formulation, an anomaly corresponds to an embedding that deviates from those produced under normal relational patterns.

The combination of $f$ and $g$ therefore operationalizes the central RCD principle: *the meaning of a token arises from its relationships with the rest of the sequence*.

## C.3 INSTANTIATION IN TIMERCD VIA SELF-ATTENTION

The RCD formulation aligns naturally with the mechanism of self-attention. In a Transformer, each token $X_t$ produces a query vector

$$q_t = X_t W^Q,$$

while each token $X_j$ generates

$$k_j = X_j W^K, \qquad v_j = X_j W^V.$$

The pairwise interaction between tokens $t$ and $j$ is expressed through the scaled dot-product score

$$s_{tj} = \frac{q_t \cdot k_j}{\sqrt{d_k}}.$$

The corresponding unnormalized contribution can be written as

$$f(X_j, X_t) = v_j \, \exp(s_{tj}),$$

which directly instantiates the pairwise interaction function $f$.

Self-attention aggregates these contributions using softmax normalization, which serves as a concrete realization of the aggregation function $g$:

$$\text{embedding}_{X_t} = \frac{\sum_j v_j \exp\left(\frac{q_t \cdot k_j}{\sqrt{d_k}}\right)}{\sum_j \exp\left(\frac{q_t \cdot k_j}{\sqrt{d_k}}\right)}.$$

Within the RCD perspective, discrepancy corresponds to deviations in the attention pattern of token $X_t$ relative to the relational patterns learned from normal data. When a token is anomalous, its query–key interactions produce an atypical similarity distribution $s_{tj}$, leading to a context embedding that lies outside the manifold of embeddings observed during normal behavior. The anomaly head is then trained to map such deviations in the embedding to a high anomalous probability, thereby converting relational irregularity directly into an anomaly score.

### C.4 ANOMALY HEAD, RECONSTRUCTION HEAD, AND TRAINING OBJECTIVE

TimeRCD employs two heads during training: an anomaly head and a reconstruction head. The anomaly head applies a lightweight MLP to each contextual embedding and outputs logits for the normal and anomalous classes:

$$\text{logits}_{x_t} = \text{AnomalyHead}(\text{embedding}_{X_t}),$$

and the anomaly score is defined as the softmax probability of the anomalous class:

$$\text{AnomalyScore}_{x_t} = \text{softmax}(\text{logits}_{x_t})[1].$$

The reconstruction head is a 3-layer MLP that predicts a scalar reconstruction for each timestep:

$$\widehat{x}_t = \text{ReconstructionHead}(\text{embedding}_{X_t}).$$

The total loss combines a cross-entropy loss for anomaly prediction and an MSE loss for reconstruction:

$$L_{\text{total}} = \lambda_{\text{anomaly}} \cdot \text{CrossEntropy}(\text{logits}_{x_t}, y_t) + \lambda_{\text{recon}} \cdot \text{MSE}(\widehat{x}_t, x_t),$$

where $y_t$ and $x_t$ are the label and value at timestamp $t$, respectively. In all experiments, both weighting coefficients are set to $\lambda_{\text{anomaly}} = \lambda_{\text{recon}} = 1$.

## D DATA GENERATION

Our synthetic data generation framework is designed as a curriculum to instill a universal understanding of temporal dynamics and causality in a model. The four-stage process is meticulously designed to first build a complex, predictable "normal" context (Stages 1-2) and then introduce meaningful, context-violating anomalies (Stage 3). Finally, we generate causally-aware ground-truth labels that teach the model to link an anomaly's root cause to its downstream effects (Stage 4), preparing it for zero-shot detection on unseen systems.

**Stage 1: Synthesizing the "Normal" Univariate Context** Every time series, whether it stands alone or is a component in a multivariate system, begins as a univariate baseline signal, $x_{\text{base}}(t)$. This signal is constructed as an additive composite of trend, seasonal, and noise components:

$$x_{\text{base}}(t) = T(t) + S(t) + \varepsilon(t), \quad t = 0, 1, \ldots, n-1.$$

**Sequence length.** we sample $n$ from 100 to 10000 to match variable-length sequences.

**Trend $T(t)$.** We mix deterministic and stochastic trends: $T_{\text{det}}(t)$ and $T_{\text{stoc}}(t)$. Set $T(t) = (1 - \rho_T) T_{\text{det}}(t) + \rho_T T_{\text{stoc}}(t)$ with $\rho_T \in [0, 1]$. The detailed construction of trend is shown in Appendix D.2

**Seasonality $S(t)$.** A mixture of $K$ periodic atoms:

$$S(t) = \sum_{k=1}^{K} A_k w_k(2\pi f_k t + \varphi_k; \theta_k),$$

where $w_k$ is selected from a library (sinusoid, square/triangle wave, wavelet atom), with amplitude $A_k > 0$, frequency $f_k$, phase $\varphi_k$, and shape parameters $\theta_k$ (e.g., duty cycle for square, mother/scale for wavelets). Frequencies may be harmonic $f_k = m_k f_0$ or incommensurate. The detailed construction of seasonality is shown in Appendix D.3

**Noise** $\varepsilon(t)$**.** Zero-mean noise with optional piecewise volatility:

$$\varepsilon(t) \sim \mathcal{N}\big(0, \sigma^2(t)\big), \qquad \sigma(t) = \sigma_0 \prod_{r=1}^{R} \big(1 + v_r \, \mathbf{1}_{[a_r, b_r)}(t)\big),$$

allowing volatility bursts over windows $[a_r, b_r)$. This captures heteroskedastic segments..

**Stage 2: Weaving the Multivariate Causal Context via a Discretized ODE System** This stage creates a multivariate system with clear causal dependencies. We first conceptualize continuous-time causal dynamics (ODE), then discretize to an AutoRegressive with eXogenous inputs (ARX) model via Euler.

1. **Generate a Causal Structure (DAG):** Sample a directed acyclic graph $G = (V, E)$ over $N$ nodes by first drawing an Erdős–Rényi graph and then orienting edges along a random topological order to avoid cycles. Edge presence is controlled by $p_{\text{edge}} \in (0, 1)$. An edge $(j, i) \in E$ means series $j$ causally influences $i$.

2. **Define the Continuous-Time Causal Dynamics (ODE Model):** We model the evolution of the **causal influence** on each node, $z_i(t)$, as a first-order linear ODE. This equation describes how the influence on node $i$ changes over time based on its own internal state and forcing from its parent nodes $P(i)$:
$$\frac{dz_i(t)}{dt} = -\gamma_i z_i(t) + \sum_{j \in P(i)} \beta_{ij} x_j(t - \tau_{ij}) + C_i$$
Here, $-\gamma_i z_i(t)$ represents an internal decay term, the sum represents the forcing from parent nodes $x_j$, and $C_i$ is a constant bias.

3. **Discretize to an ARX Model for Implementation:** With Euler and $\Delta t = 1$, the discrete ARX is
$$z_i[t] = a_i z_i[t-1] + \sum_{j \in P(i)} b_{ij} x_j[t - \ell_{ij}] + c_i, \quad |a_i| \le 0.8,$$
where $a_i = 1 - \gamma_i$ encodes decay stability, delays $\ell_{ij} \in \{0, \dots, \ell_{\max}\}$, and $b_{ij}$ are exogenous gains. We sample $a_i \sim \mathcal{U}[-0.8, 0.8]$, $\ell_{ij} \sim \mathcal{U}\{0, \dots, \ell_{\max}\}$, and $b_{ij} \sim \mathcal{N}(0, \sigma_b^2)$ (variance scaled by in-degree).

4. **Mix the Signals:** The observed node mixes baseline with causal channel:
$$x_i[t] = (1 - \alpha_i) x_{\text{base},i}(t) + \alpha_i z_i[t], \quad \alpha_i \in [0, 1].$$
In vector form ($x[t], z[t] \in \mathbb{R}^N$),
$$z[t] = A z[t-1] + \sum_{\delta=0}^{\ell_{\max}} B^{(\delta)} x[t - \delta] + c, \quad x[t] = (I - A') x_{\text{base}}(t) + A' z[t],$$
where $A = \text{diag}(a_i)$, $A' = \text{diag}(\alpha_i)$, and $B^{(\delta)}$ encodes lagged influences.

This process, performed sequentially according to the topological sort, creates a system where a change in a parent node will naturally propagate to its child nodes after a specified delay.

**Stage 3: Principled Anomaly Injection: Simulating Causal and Contextual Complexity**   Our primary contribution in data generation lies in a synthetic framework designed to address critical gaps in existing anomaly detection benchmarks, namely the lack of causal realism and contextual complexity. To achieve this, our methodology introduces a novel causal framework for generating anomalies based on two distinct, complementary philosophies:

- **Endogenous Anomaly (Internal System Failure):** This strategy simulates how a single root-cause anomaly propagates through an interconnected system. An anomaly is introduced into the baseline series of a parent node *before* causal mixing. Its effects are then passed to descendant nodes based on the ARX model, creating a realistic cascade of causally-linked anomalous events.

- **Exogenous Anomaly (External Shock):** This mirrors more conventional techniques, modeling an external event impacting an otherwise normal system. The normal multivariate system is fully generated first, and then a segment of a single series is overwritten with an anomalous template.

**Stage 4: Label Generation and Causal Propagation**   We generate token-level binary labels. For exogenous injections, positive labels correspond to the intervention window. For endogenous injections, we label the root-cause window and extend it to descendant channels based on their causal lags, as implied by the DAG and ARX lags $\{\ell_{ij}\}$. To accurately capture the duration of these propagated effects, we rely on the physical properties of the generated system. The duration is mathematically determined by the half-life parameter derived from the ODE dynamics in Stage 2. Specifically, the ARX decay coefficient $a_i$ determines the half-life $t_{1/2} \approx \ln(2)/(1 - |a_i|)$ of the impulse response. We define the effective anomaly window for descendant nodes by extending the root-cause duration by a factor of the half-life (e.g., $3 \times t_{1/2}$), ensuring the label encompasses the period where the causal shock remains significant before decaying. While our generation process tracks channel-specific anomalies (root-cause vs. propagated effects), for this work, we aggregate these into temporal localization labels (marking when an anomaly occurs across any channel). This aligns with standard zero-shot anomaly detection benchmarks, which typically provide timestep-level but not channel-level ground truth. Sequence lengths, DAG sparsity, ARX coefficients, and signal regimes (trend, seasonality, noise) are sampled from configurable priors (Appx. D.1), generating rich, interpretable dynamics designed for zero-shot learning.

In addition to causal dynamics, we move beyond simple point anomalies by creating a rich taxonomy of contextual anomalies designed to violate specific, established patterns within the data. These include:

- **Local/Change Anomalies:** Additive templates, $\Delta(t)$, applied over a window, representing events like a sudden sensor glitch (**Spike**), a component failure shifting its baseline (**Level Shift**), or mechanical vibrations (**Shake**).

$$x'_{\text{anom}}(t) = x_{\text{norm}}(t) + \Delta(t) \quad \text{for } t \in [t_s, t_e)$$

- **Seasonal Anomalies:** The normal seasonal component $S(t)$ is replaced with an altered version $S'(t)$. They simulate systemic issues like a machine operating at the wrong speed (**Frequency Change**) or a process desynchronization (**Phase Shift**).

$$x'_{\text{anom}}(t) = T(t) + S'(t) + \varepsilon(t) \quad \text{for } t \in [t_s, t_e)$$

In contrast to methods that inject isolated anomalies into independent time series, our dual-pronged approach provides a more challenging and realistic benchmark. By generating complex scenarios with both contextual violations and causal propagation, our dataset enables a more robust evaluation of a model's ability to understand system-wide dynamics. Details of constructing different abnormal types in the D.4

## D.1 HYPERPARAMETERS AND SAMPLING MAPPINGS

To make the generation process transparent yet non-programmatic, we group hyperparameters into: dataset-level controls (Table 3); attribute families and their sampling weights (Table 4); multivariate graph and causal dynamics (Table 5).

Table 3: Dataset-level generation controls and intended effects.

| Name | Typical range / rule | Applies to | Intended effect |
|---|---|---|---|
| Number of samples | positive integer | whole dataset | Total number of generated sequences. |
| Sequence length | uniformly in $\{100, \dots, 10000\}$ | per sample | Time horizon of each sequence. |
| Anomalous-sample ratio | real in $[0, 1]$ | per sample | Probability a sample contains anomalies. |
| Multivariate flag | $\{$True, False$\}$ | per sample | Whether to generate multi-channel series. |
| Number of features | fixed or uniformly in $\{2, \dots, 50\}$ | multivariate only | Channel count when multivariate. |
| Random seed (optional) | user-provided integer | whole run | Reproducibility of the random draws. |

Table 4: Attribute categories, options, and relative probability.

| Category | Options | Probability |
|---|---|---|
| Seasonality type | no periodic fluctuation | 0.3 |
| | sine | 0.3 |
| | square | 0.05 |
| | triangle | 0.05 |
| | wavelet | 0.3 |
| Trend type | decrease | 0.2 |
| | increase | 0.2 |
| | keep steady | 0.2 |
| | multiple | 0.3 |
| | ARIMA | 0.1 |
| Frequency regime | high frequency | 0.5 |
| | low frequency | 0.5 |
| Noise level | almost no noise | 0.25 |
| | low | 0.25 |
| | moderate | 0.25 |
| | high | 0.25 |

Table 5: Multivariate and causal-dynamics hyperparameters and intended roles.

| Name | Typical range / rule | Applies to | Intended role |
|---|---|---|---|
| Number of nodes | Uniformly in 1 to 50 | system graph | Dimensionality of the multivariate system. |
| Number of edges | roughly linear in nodes (acyclic) | system graph | Connectivity while preserving acyclicity. |
| Parent lags | nonnegative integers up to a small fraction of length | per edge | Causal delays between channels. |
| ARX autoregression $a_i$ | real in $[-0.8, 0.8]$ | per node | Decay/stability of latent causal influence. |
| ARX exogenous gains $b_{ij}$ | zero-mean with finite variance | per edge | Strength of parent-to-child coupling. |
| ARX bias $c_i$ | bounded real interval | per node | Baseline offset in causal channel. |
| Mixing weight $\alpha_i$ | real in $[0, 1]$ | per node | Balance between baseline and causal channel. |

## D.2 TREND TYPES AND FORMAL DEFINITIONS

We instantiate the trend component $T(t)$ using five archetypes aligned with the generator configuration. Let $t \in \{0, \dots, n-1\}$.

**Increase / Decrease / Keep steady.** We use an affine model

$$T(t) = k_0 + k_1 t,$$

with constraints: *increase* uses $k_1 > 0$, *decrease* uses $k_1 < 0$, and *keep steady* uses $k_1 = 0$ (constant baseline).

**Multiple (piecewise trend).** To capture mixed regimes, we allow $P$ change-points at integer knots $0 < \tau_1 < \cdots < \tau_P < n$ and define a piecewise-linear trend with slope updates:

$$T(t) = k_0 + k_1 t + \sum_{p=1}^{P} \delta_p \left[ t \geq \tau_p \right] (t - \tau_p)_+,$$

where $(u)_+ = \max\{u, 0\}$ and $\delta_p$ adjusts slope after each $\tau_p$. This matches the additive hinge construction used in Stage 1.

**ARIMA (stochastic trend).** For stochastic trends we use a differenced ARMA process. Let $\nabla^d$ be the $d$-th difference operator. A general ARIMA$(p, d, q)$ satisfies

$$\phi(B)\,\nabla^d T(t) = \theta(B)\,\varepsilon_T(t), \qquad \varepsilon_T(t) \sim \mathcal{N}(0, \sigma_T^2),$$

with backshift $B$, AR polynomial $\phi(B) = 1 - \sum_{i=1}^{p} \phi_i B^i$, and MA polynomial $\theta(B) = 1 + \sum_{j=1}^{q} \theta_j B^j$. In practice we use small orders (e.g., $p, q \leq 2$) and modest variance.

### D.3 SEASONALITY TYPES AND FORMAL DEFINITIONS

We support five seasonal archetypes. Let the base period be $P > 0$ (in samples), frequency $f = 1/P$, amplitude $A > 0$, and phase $\varphi \in [0, 2\pi)$.

**No periodic fluctuation.** No seasonality is present:

$$S(t) = 0.$$

**Sin periodic fluctuation.** A sinusoidal component:

$$S(t) = A \sin\big(2\pi f t + \varphi\big).$$

**Square periodic fluctuation.** A rectangular wave with duty cycle $d \in (0, 1)$:

$$S(t) = A \operatorname{sgn}\big(\sin(2\pi f t + \varphi)\big),$$

or equivalently by its Fourier series $S(t) = \frac{4A}{\pi} \sum_{m \text{ odd}} m^{-1} \sin(2\pi m f t + \varphi)$; duty-cycle variants modulate on/off durations per period.

**Triangle periodic fluctuation.** A symmetric triangle wave of period $P$:

$$S(t) = \frac{8A}{\pi^2} \sum_{m \text{ odd}} \frac{(-1)^{(m-1)/2}}{m^2} \sin(2\pi m f t + \varphi),$$

equivalently realizable as a piecewise-linear ramp that rises and falls within each period.

**Wavelet periodic fluctuation.** A sum of localized atoms (e.g., db, sym, coif, haar, bior, dmey families):

$$S(t) = \sum_{\ell=1}^{L} A_\ell\, \psi_\ell\!\left(\frac{t - \tau_\ell}{s_\ell}\right), \qquad A_\ell \in \mathbb{R},\ s_\ell > 0,\ \tau_\ell \in \mathbb{R}.$$

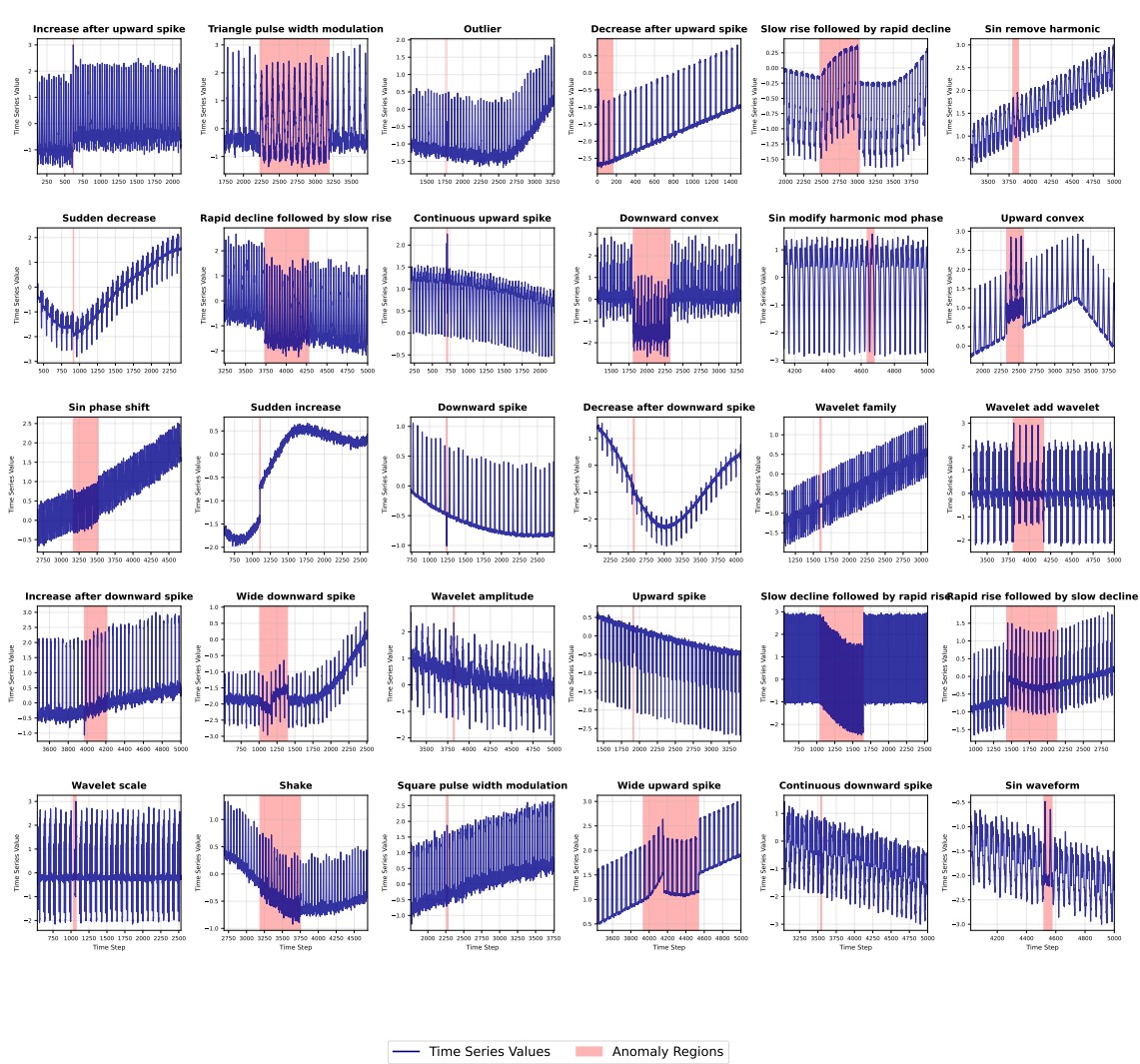

Multi-Anomaly Type Visualization from Synthetic Dataset

## D.4 A TAXONOMY OF SYNTHETIC ANOMALY ARCHETYPES

Following the high-level description, we now provide complete, self-contained definitions and formulas for each anomaly type supported by our generator. We divide them into two families: Local/Change Anomalies (abrupt, windowed changes) and Seasonal Anomalies (contextual violations of periodic structure). We avoid shorthand; every symbol is explicitly defined. A comprehensive summary of all anomaly archetypes with intents and formal definitions is provided in Table 6.

**Notation and primitives.** We use discrete time index $t \in \{0, 1, \ldots, n-1\}$. The normal (anomaly-free) series is denoted by $x_{\text{norm}}(t)$. An anomaly acts on a closed-open window $[t_s, t_e)$ with integers $0 \le t_s < t_e \le n$. The indicator function is

$$\mathbf{1}_{[t_s, t_e)}(t) = \begin{cases} 1, & t_s \le t < t_e, \\ 0, & \text{otherwise.} \end{cases}$$

The Heaviside step function at integer times is $H(t; t_0) = \mathbf{1}_{[t_0, \infty)}(t)$. For any scalar $u$, the logistic sigmoid is $\sigma(u) = \frac{1}{1+e^{-u}}$. For a rectangular window we use $h(t; t_s, t_e) = \mathbf{1}_{[t_s, t_e)}(t)$. All amplitudes $A, B, \{A_m\}$ are real scalars; all durations and lags (e.g., $w, \tau_r, \tau_f, d$) are positive integers unless otherwise stated.

### D.4.1 LOCAL/CHANGE ANOMALIES

All local anomalies are applied additively within a window using a template $\Delta(t)$:

$$x'_{\text{anom}}(t) = x_{\text{norm}}(t) + \Delta(t)\, \mathbf{1}_{[t_s, t_e)}(t), \quad t = 0, 1, \ldots, n-1.$$

**Spikes and bursts**

- **Upward spike** (a single sharp positive glitch). Parameters: center time $t_0 \in [t_s, t_e)$, half-width $w \in \mathbb{N}^+$, amplitude $A > 0$. Define

$$\Delta(t) = A \max\left(1 - \frac{|t - t_0|}{w}, 0\right), \quad t \in \mathbb{Z}.$$

- **Downward spike** (a single sharp negative glitch). Same parameters as above with negative amplitude:

$$\Delta(t) = -A \max\left(1 - \frac{|t - t_0|}{w}, 0\right).$$

- **Continuous upward spikes** (a burst of multiple positive spikes). Parameters: spike count $M \in \mathbb{N}^+$, centers spaced by stride $d \in \mathbb{N}^+$, per-spike amplitudes $A_m > 0$, per-spike widths $w_m \in \mathbb{N}^+$. Define

$$\Delta(t) = \sum_{m=0}^{M-1} A_m \max\left(1 - \frac{|t - (t_0 + md)|}{w_m}, 0\right).$$

- **Continuous downward spikes** (a burst of negative spikes). Use

$$\Delta(t) = \sum_{m=0}^{M-1} (-A_m) \max\left(1 - \frac{|t - (t_0 + md)|}{w_m}, 0\right).$$

- **Wide upward spike** (longer rise, short plateau, and fall). Parameters: rise length $\tau_r \in \mathbb{N}^+$, fall length $\tau_f \in \mathbb{N}^+$, amplitude $A > 0$. Define

$$\Delta(t) = \begin{cases} A \frac{t - t_s}{\tau_r}, & t \in [t_s, t_s + \tau_r), \\ A, & t \in [t_s + \tau_r, t_e - \tau_f), \\ A\left(1 - \frac{t - (t_e - \tau_f)}{\tau_f}\right), & t \in [t_e - \tau_f, t_e), \\ 0, & \text{else.} \end{cases}$$

- **Wide downward spike** (longer fall then recovery). Use the same piecewise definition with amplitude $-A$.

- **Outlier** (single-point shock). Parameters: position $t_0 \in [t_s, t_e)$, amplitude $A \in \mathbb{R}$. Define the Kronecker impulse
$$\Delta(t) = \begin{cases} A, & t = t_0, \\ 0, & t \neq t_0. \end{cases}$$

**Level shifts and plateaus**

- **Sudden increase** (permanent level up-shift). Parameters: slope parameter $\kappa > 0$, amplitude $A > 0$, change time $t_0 \in [t_s, t_e)$. Define
$$\Delta(t) = A\,\sigma\big(\kappa(t - t_0)\big), \quad \sigma(u) = \frac{1}{1 + e^{-u}}.$$

- **Sudden decrease** (permanent level down-shift). Use
$$\Delta(t) = -A\,\sigma\big(\kappa(t - t_0)\big).$$

- **Convex/Concave plateau** (temporary level change using a raised-cosine window). Parameters: window $[t_s, t_e]$, amplitude $A > 0$, sign $s \in \{+1, -1\}$ (convex when $s = +1$, concave when $s = -1$). Define
$$\Delta(t) = s\,A\,\tfrac{1}{2}\left(1 - \cos\frac{\pi(t - t_s)}{t_e - t_s}\right)\mathbf{1}_{[t_s, t_e)}(t).$$

**Asymmetric transients**

- **Rapid rise followed by slow decline**. Parameters: rise time constant $\tau_r \in \mathbb{N}^+$, fall time constant $\tau_f \in \mathbb{N}^+$, peak time $t_p \in [t_s, t_e]$, amplitude $A > 0$. Define
$$\Delta(t) = A\Big(1 - e^{-\frac{t - t_s}{\tau_r}}\Big)\mathbf{1}_{[t_s, t_p)}(t) + A\,e^{-\frac{t - t_p}{\tau_f}}\,\mathbf{1}_{[t_p, t_e)}(t), \quad \tau_r \ll \tau_f.$$

- **Slow rise followed by rapid decline**. Use the same definition with $\tau_r \gg \tau_f$.

- **Rapid decline followed by slow rise**. Use the above definition with amplitude $-A$ and swap rise/fall roles accordingly.

- **Slow decline followed by rapid rise**. Use the rapid rise–slow decline formula with amplitude $-A$ and $\tau_r \gg \tau_f$.

**Spike + level interaction**

- **Decrease after upward spike**. Parameters: spike template $\Delta_{\mathrm{spk}}(t)$ (e.g., the triangular spike above), post-spike shift magnitude $B > 0$, shift start $t_1 \geq t_s$. Define
$$\Delta(t) = \Delta_{\mathrm{spk}}(t) - B\,H(t; t_1), \quad H(t; t_1) = \mathbf{1}_{[t_1, \infty)}(t).$$

- **Increase after downward spike**. Using a negative spike template $\Delta_{\mathrm{nspk}}(t)$ and a positive shift $B > 0$:
$$\Delta(t) = \Delta_{\mathrm{nspk}}(t) + B\,H(t; t_1).$$

- **Increase after upward spike**. Use
$$\Delta(t) = \Delta_{\mathrm{spk}}(t) + B\,H(t; t_1).$$

- **Decrease after downward spike**. Use

$$\Delta(t) = \Delta_{\text{nspk}}(t) - B\,H(t; t_1).$$

**High-frequency burst**

- **Shake** (localized high-frequency vibration). Parameters: carrier frequency $f_h \in \mathbb{R}_{>0}$ in cycles per sample, phase $\varphi \in [0, 2\pi)$, amplitude $A > 0$. With a rectangular window $h(t; t_s, t_e)$,

$$\Delta(t) = A\,h(t; t_s, t_e)\,\sin(2\pi f_h t + \varphi).$$

### D.4.2 SEASONAL ANOMALIES

Seasonal anomalies are implemented by replacing the seasonal component $S(t)$ with a modified component $S'(t)$ strictly within the window $[t_s, t_e)$, while leaving the rest unchanged:

$$x'_{\text{anom}}(t) = \begin{cases} T(t) + S'(t) + \varepsilon(t), & t_s \leq t < t_e, \\ T(t) + S(t) + \varepsilon(t), & \text{otherwise.} \end{cases}$$

**Universal transforms (any waveform)**

- **Waveform inversion** (polarity flip). For $t \in [t_s, t_e)$, set $S'(t) = -S(t)$.

- **Amplitude scaling**. For $t \in [t_s, t_e)$, set $S'(t) = r\,S(t)$ with scaling factor $r > 0$.

- **Frequency change**. Let the original period be $P > 0$ and frequency $f = 1/P$. Choose a period multiplier $\rho > 0$ and set the new period $P' = \rho P$ and frequency $f' = 1/P'$. For a sinusoid $S(t) = A\sin(2\pi f t + \varphi)$, define $S'(t) = A\sin(2\pi f't + \varphi)$ for $t \in [t_s, t_e)$.

- **Noise injection**. For $t \in [t_s, t_e)$, set $S'(t) = S(t) + \eta(t)$ with i.i.d. Gaussian noise $\eta(t) \sim \mathcal{N}(0, \sigma_S^2)$.

- **Waveform change from sine to square/triangle**. If $S(t)$ is sinusoidal with amplitude $A$ and period $P$, replace it by a square/triangle waveform $w_{\text{new}}(t; A, P, \theta)$ of the same amplitude and period (with shape parameters $\theta$) for $t \in [t_s, t_e)$.

**Sinusoidal-specific (harmonics and phase)** Let a sinusoidal seasonal component be represented by a finite harmonic sum

$$S(t) = \sum_{n=1}^{N} A_n \sin\left(2\pi n f\,t + \varphi_n\right),$$

and optionally include amplitude modulation for each harmonic via

$$A_n(t) = \frac{A_n}{n}\left(1 + d_n \sin(\omega_m t + \psi_n)\right), \quad d_n \in [0, 1),\ \omega_m > 0,\ \psi_n \in [0, 2\pi).$$

- **Phase shift**. For $t \in [t_s, t_e)$, update all phases by a fixed shift $\Delta\varphi \in (0, 2\pi)$: $\varphi_n \mapsto \varphi_n + \Delta\varphi$.

- **Add harmonic**. For $t \in [t_s, t_e)$, add a new harmonic of order $m \in \mathbb{N}^+$: $S'(t) = S(t) + A_h \sin(2\pi m f\,t + \varphi_h)$ with amplitude $A_h > 0$ and phase $\varphi_h \in [0, 2\pi)$.

- **Remove harmonic**. For $t \in [t_s, t_e)$, remove one existing harmonic index $n^* \in \{1, \ldots, N\}$ from the sum, keeping the others unchanged.

- **Modify harmonic phase**. For $t \in [t_s, t_e)$, choose one index $n^*$ and set $\varphi_{n^*} \mapsto \varphi'_{n^*}$ such that $|\varphi'_{n^*} - \varphi_{n^*}| \geq \delta$ for a given threshold $\delta > 0$.

- **Modify amplitude-modulation depth**. For $t \in [t_s, t_e)$, choose index $n^*$ and set $d_{n^*} \mapsto d'_{n^*}$ within $[0, 1)$.

- **Modify modulation frequency**. For $t \in [t_s, t_e)$, update $\omega_m \mapsto \omega'_m$ with $\omega'_m > 0$.

- **Modify modulation phase**. For $t \in [t_s, t_e)$, choose index $n^*$ and set $\psi_{n^*} \mapsto \psi'_{n^*}$.

**Square/Triangle-specific (pulse geometry)** For a square or triangle waveform with period $P > 0$, define the duty cycle $d \in (0, 1)$ as the fraction of each period the waveform is at its high level, and define a cycle start parameter $\delta \in [0, 1)$ that shifts the waveform phase relative to each period.

- **Pulse shift**. For $t \in [t_s, t_e)$, apply a phase shift $\Delta \in (-1, 1)$ cycles: $\delta \mapsto (\delta + \Delta) \bmod 1$.

- **Pulse width modulation**. For $t \in [t_s, t_e)$, scale the duty cycle by a factor $\lambda > 0$: $d \mapsto d' = \min\{1, \max\{0, \lambda d\}\}$.

**Wavelet-based (atoms and families)** Let a wavelet-based seasonal component be represented as a finite sum of atoms

$$S(t) = \sum_{\ell=1}^{L} A_\ell \, \psi_\ell \left( \frac{t - \tau_\ell}{s_\ell} \right), \quad A_\ell \in \mathbb{R}, \ s_\ell > 0, \ \tau_\ell \in \mathbb{R}.$$

- **Family change**. For $t \in [t_s, t_e)$, replace each mother wavelet $\psi_\ell$ taken from a family $\mathcal{F}$ by a corresponding wavelet $\tilde{\psi}_\ell$ from a different family $\tilde{\mathcal{F}}$ while keeping $A_\ell, s_\ell, \tau_\ell$ unchanged.

- **Scale change**. For $t \in [t_s, t_e)$, update scales by a factor $\lambda > 0$: $s_\ell \mapsto s'_\ell = \lambda s_\ell$ for selected indices.

- **Shift change**. For $t \in [t_s, t_e)$, update translations by $\tau_\ell \mapsto \tau'_\ell = \tau_\ell + \Delta_{\tau,\ell}$.

- **Amplitude change**. For $t \in [t_s, t_e)$, rescale coefficients by $A_\ell \mapsto r_\ell A_\ell$ with $r_\ell > 0$.

- **Add wavelet**. For $t \in [t_s, t_e)$, append a new atom $A_+ \psi_+ \left( \frac{t - \tau_+}{s_+} \right)$ with specified parameters.

- **Remove wavelet**. For $t \in [t_s, t_e)$, remove one existing atom index from the sum.

Table 6: Summary of anomaly archetypes, intents, and formal definitions.

| Family | Type | Intent (one sentence) | Formal definition (within $[t_s, t_e]$) |
|---|---|---|---|
| Local | Upward spike | A single sharp positive glitch | $\Delta(t) = A \max(1 - |t - t_0|/w, 0)$ |
| | Downward spike | A single sharp negative glitch | $\Delta(t) = -A \max(1 - |t - t_0|/w, 0)$ |
| | Continuous upward spikes | Rapid consecutive positive spikes | $\sum_{m=0}^{M-1} A_m \max(1 - |t - (t_0 + md)|/w_m, 0)$ |
| | Continuous downward spikes | Rapid consecutive negative spikes | $\sum_{m=0}^{M-1} (-A_m) \max(1 - |t - (t_0 + md)|/w_m, 0)$ |
| | Wide upward spike | Longer rise/plateau/fall pulse | piecewise linear rise/hold/fall with amplitude $A$ |
| | Wide downward spike | Longer fall/plateau/rise pulse | piecewise linear with amplitude $-A$ |
| | Outlier | Single-point shock | $\Delta(t) = A$ if $t = t_0$, else $0$ |
| | Sudden increase | Permanent upward level shift | $\Delta(t) = A \sigma(\kappa(t - t_0))$ |
| | Sudden decrease | Permanent downward level shift | $\Delta(t) = -A \sigma(\kappa(t - t_0))$ |
| | Convex/Concave plateau | Temporary level change | $sA\frac{1}{2}\left(1 - \cos(\pi(t - t_s)/(t_e - t_s))\right)$ |
| | Rapid rise → slow decline | Fast excitation then slow decay | $A(1 - e^{-(t-t_s)/\tau_r})\mathbf{1}_{[t_s, t_p]} + Ae^{-(t-t_p)/\tau_f}\mathbf{1}_{[t_p, t_e]}$ |
| | Slow rise → rapid decline | Slow excitation then fast decay | same as above with $\tau_r \gg \tau_f$ |
| | Rapid decline → slow rise | Fast drop then slow recovery | negative-amplitude version of above |
| | Slow decline → rapid rise | Slow drop then fast recovery | negative-amplitude version of above |
| | Decrease after upward spike | Spike followed by downward shift | $\Delta_{\text{spk}}(t) - B\,H(t; t_1)$ |
| | Increase after downward spike | Negative spike then upward shift | $\Delta_{\text{nspk}}(t) + B\,H(t; t_1)$ |
| | Increase after upward spike | Spike followed by upward shift | $\Delta_{\text{spk}}(t) + B\,H(t; t_1)$ |
| | Decrease after downward spike | Negative spike then downward shift | $\Delta_{\text{nspk}}(t) - B\,H(t; t_1)$ |
| | Shake | Localized high-frequency vibration | $A \sin(2\pi f_h t + \varphi)$ within $[t_s, t_e]$ |
| Seasonal | Waveform inversion | Polarity flip of seasonal component | $S'(t) = -S(t)$ |
| | Amplitude scaling | Change seasonal magnitude | $S'(t) = r\,S(t)$ |
| | Frequency change | Alter base period/frequency | $S'(t)$ with $P' = \rho P$ (e.g., $f' = 1/P'$) |
| | Noise injection | Add Gaussian noise to seasonality | $S'(t) = S(t) + \eta(t),\ \eta \sim \mathcal{N}(0, \sigma_S^2)$ |
| | Waveform change (sin→square/triangle) | Replace waveform family | $S'(t) = w_{\text{new}}(t; A, P, \theta)$ |
| | Phase shift (sinusoid) | Global phase offset | $\varphi_n \mapsto \varphi_n + \Delta\varphi$ |
| | Add harmonic (sinusoid) | Introduce new harmonic | $S'(t) = S(t) + A_h \sin(2\pi mft + \varphi_h)$ |
| | Remove harmonic (sinusoid) | Remove one harmonic | drop index $n^*$ term in $\sum_n$ |
| | Modify harmonic phase | Change one harmonic phase | $\varphi_{n^*} \mapsto \varphi'_{n^*}$ |
| | Modify amp-mod depth | Change amplitude modulation depth | $d_{n^*} \mapsto d'_{n^*}$ |
| | Modify modulation frequency | Change modulation frequency | $\omega_m \mapsto \omega'_m$ |
| | Modify modulation phase | Change modulation phase | $\psi_{n^*} \mapsto \psi'_{n^*}$ |
| | Pulse shift (square/triangle) | Shift cycle start | $\delta \mapsto (\delta + \Delta) \bmod 1$ |
| | Pulse width modulation | Change duty cycle | $d \mapsto d' = \min\{1, \max\{0, \lambda d\}\}$ |
| | Wavelet family change | Swap mother wavelet family | $\psi_\ell \in \mathcal{F} \to \tilde{\psi}_\ell \in \tilde{\mathcal{F}}$ |
| | Wavelet scale change | Rescale atom width | $s_\ell \mapsto s'_\ell = \lambda s_\ell$ |
| | Wavelet shift change | Shift atom translation | $\tau_\ell \mapsto \tau'_\ell$ |
| | Wavelet amplitude change | Rescale atom amplitude | $A_\ell \mapsto r_\ell A_\ell$ |
| | Add wavelet atom | Add new atom | append $(A_+, s_+, \tau_+, \psi_+)$ |
| | Remove wavelet atom | Remove existing atom | delete one summand index |

# E  DETAILS OF EXPERIMENTS

## E.1  BENCHMARK DATASETS

Our evaluation uses a selection of datasets from one primary source, the TSB-AD benchmark by Liu & Pa-
parrizos (2024). Our dataset selection prioritizes evaluation quality and avoids redundancy. Approximately
50% of the TSB-AD benchmark's "univariate" section consists of decomposed multivariate datasets (where
dimensions of multivariate series are treated as separate univariate ones), which introduces bias and in-
flates performance metrics. To ensure a rigorous and unbiased evaluation, we deliberately filtered out these
decomposed series, focusing on **genuinely univariate** datasets for the univariate track and selecting five
standard multivariate datasets for the multivariate track (testing them in their original form). This selection
covers diverse domains (e.g., medical, industrial, server metrics, finance, weather) while preventing "double
counting" and providing a clearer picture of model capabilities.

- **Univariate Datasets**: We utilize a diverse collection of univariate datasets including IOPS, MGAB,
  NAB, NEK, Power, SED, Stock, TODS, UCR, WSD, and YAHOO.

- **Multivariate Datasets**: For multivariate anomaly detection, we use the MSL, PSM, SMAP, SMD,
  and SWaT datasets. These are sourced from industrial control systems and spacecraft telemetry,
  presenting complex, multi-dimensional dependencies.

The specific characteristics of these datasets, including their domain, number of time series (TS), average
length, and anomaly ratio, are summarized in Table 7 and Table 8.

Table 7: Univariate Datasets

| Name | Domain | #TS | Avg Length | AR (%) |
|------|--------|-----|-----------|--------|
| UCR | Misc. | 228 | 67818.7 | 0.6 |
| NAB | Web | 28 | 5099.7 | 10.6 |
| YAHOO | Web | 259 | 1560.2 | 0.6 |
| IOPS | Operations | 17 | 72792.3 | 1.3 |
| MGAB | Sensor | 9 | 97777.8 | 0.2 |
| SED | Energy | 3 | 23332.3 | 4.1 |
| Stock | Finance | 20 | 15000 | 9.4 |
| TODS | Traffic | 15 | 5000.0 | 6.3 |
| NEK | Weather | 9 | 1073.0 | 8.0 |
| Power | Power Grid | 1 | 35040.0 | 8.5 |
| WSD | Weather | 111 | 17444.5 | 0.6 |

Table 8: Multivariate Datasets

| Name | Domain | #TS | Avg Length | AR (%) |
|------|--------|-----|-----------|--------|
| MSL | Space | 16 | 3119.4 | 5.1 |
| PSM | Sensor | 1 | 217624.0 | 11.2 |
| SMAP | Space | 27 | 7855.9 | 2.9 |
| SMD | Server | 22 | 25466.4 | 3.8 |
| SWaT | ICS | 2 | 207457.5 | 12.7 |

## E.2 BASELINES

Our study compares a range of baseline models, categorized into zero-shot and full-shot detectors, following the methodology from TSB-AD benchmark (Liu & Paparrizos, 2024) and the DADA (Shentu et al., 2024).

**Zero-Shot Models** These models are pre-trained on large-scale datasets and can be applied directly to new time series without fine-tuning.

- **DADA**: This model, from the paper by (Shentu et al., 2024), is a pre-trained general anomaly detector. We set the window size to 100.

- **TS-Pulse**: An IBM foundation model that uses a TSMixer architecture (Ekambaram et al., 2025). It is pre-trained on a massive corpus for multi-task applications, including anomaly detection. Following the suggestion from the paper, we set the window size with a length of 96.

- **MOMENT**: A foundation model that utilizes a patch-based pre-training strategy to learn rich representations from diverse time series data (Goswami et al., 2024). We use a window size of 64.

- **TimesFM**: A decoder-only transformer model from Google trained on a large time series corpus using a patching strategy, enabling strong zero-shot generalization (Das et al., 2024). We set the window size to 96.

- **Chronos**: A generative model that frames time series analysis as a language modeling task, using a transformer-based architecture to learn and predict time series values (Ansari et al., 2024). Its window size is 100.

- **Time MOE**: A decoder-only transformer model with a sparse Mixture-of-Experts (MoE) architecture. It is pre-trained on a large time series corpus for forecasting and multi-task learning (Shi et al., 2024). We set the window size to 96.

**Full-Shot Models** These models require training on the target dataset.

- **TranAD**: A transformer-based model that uses a reconstructive approach to detect anomalies by comparing original and reconstructed time series (Tuli et al., 2022). It is configured with a window size of 10.

- **USAD**: An autoencoder-based model that employs an adversarial training mechanism to enhance its reconstruction capability and anomaly detection (Audibert et al., 2020). We set the window size to 100.

- **OmniAnomaly**: A deep learning model that uses a Variational Autoencoder (VAE) with a Gated Recurrent Unit (GRU) to learn normal patterns and detect deviations (Su et al., 2019). The model is configured with a window size of 100.

- **Sub-PCA**: A subspace-based method that detects anomalies by identifying deviations in principal component subspaces (Liu & Paparrizos, 2024). The model decomposes multivariate time series into subspaces and flags anomalies based on reconstruction errors in these subspaces.

- **DCdetector**: A deep learning approach that combines dual attention mechanisms and contrastive learning for time series anomaly detection (Yang et al., 2023). The model uses both temporal and channel-wise attention to capture complex dependencies.

- **TFMAE**: A transformer-based masked autoencoder that learns representations by reconstructing masked time series segments (Fang et al., 2024). The model leverages self-supervised pre-training to learn robust temporal patterns for anomaly detection.

- **LOF**: A traditional statistical method that measures the local deviation of a data point from its neighbors, identifying anomalies with lower local density (Breunig et al., 2000). For univariate

datasets, we set n neighbors to 50, and for multivariate, we set n neighbors to 50 with metric as euclidean.

- **IForest**: An ensemble of Isolation Trees that isolates anomalies based on the number of random partitions required to separate them from the rest of the data (Liu et al., 2008). For univariate datasets, we set n estimators to 200, and for multivariate, we set n estimators to 25 and max features to 0.8.

### E.3 BASELINES DATA LEAKAGE

We exclude certain baseline results (specifically from DADA and MOMENT) because their pre-training corpora heavily overlap with our zero-shot test benchmarks, violating the zero-shot evaluation setting.

- DADA: Its pre-training data includes IOPS, MGAB, and YAHOO (Shentu et al. (2024)).
- MOMENT: Uses the "Time Series Pile," which contains IOPS, MGAB, NAB, YAHOO, MSL, SMAP, and SMD (Goswami et al. (2024)).

### E.4 EVALUATION METRIC CALCULATIONS

Our performance evaluation is conducted using four metrics: Affiliation F1 ($F1_A$), Temporal F1 ($F1_T$), Standard F1 ($F1$), and VUS-PR. We adopt a selection of metrics from the TSB-AD benchmark (Liu & Paparrizos, 2024) for Affiliation F1 ($F1_A$), Standard F1 ($F1$), and VUS-PR. For Temporal F1, we follow the implmentation of Sarfraz et al. (2024).

- **Standard-F1** The Standard F1-score is a widely used metric that provides a harmonic mean of precision and recall. It is calculated using point-wise True Positives (TP), False Positives (FP), and False Negatives (FN).
- **F1-T** Temporal F1, as described by Sarfraz et al. (2024), is a range-based metric that evaluates anomaly detection performance by considering the temporal context of anomalies. It is a variant of the F1-score that addresses common issues in time series evaluation, such as overlapping predictions and temporal proximity.
- **Affiliation-F** ($F1_A$) Affiliation F1 is a distance-based metric that measures the "affiliation" or proximity between the ground truth and predicted anomaly points. It is designed to be less sensitive to minor temporal shifts in the predicted anomalies. The score is calculated by finding the optimal one-to-one mapping between the ground truth and detected anomalies and then computing the F1-score based on these affiliations (Huet et al., 2022).
- **VUS-PR** (Volume Under Surface - Precision/Recall) is a threshold-independent, parameter-free metric for time series anomaly detection. Unlike point-wise metrics, VUS-PR is robust to time lags and measures the area under a 3D surface plot of precision, recall, and a buffer parameter (Paparrizos et al., 2022). It addresses the limitations of standard F1 scores by creating a continuous buffer region around each anomaly, thus providing a more reliable and nuanced evaluation of model performance.

### E.5 TRAINING DETAILS

**Pre-training Setup** The pretraining setup for our time series model is built around a transformer architecture. This encoder features 8 transformer layers, each with 8 attention heads. The internal dimension (d_model) of the model is 512, which is then projected down to a dimension of 256 (d_proj). The input

time series is divided into patches of size 16. The model is trained using the `AdamW` optimizer with a learning rate of 5e-4 and a weight decay of 1e-5. The total loss is a sum of two components: a Mean Squared Error loss for a masked reconstruction task and a Cross-Entropy loss for an anomaly detection task. Training runs for a maximum of 50 epochs with a batch size that starts at 64 but is adjusted dynamically, and it includes an early stopping mechanism that halts training if the validation loss doesn't improve for 7 epochs.

**Pre-training Dataset Details**    We trained our model **only** on our custom synthetic dataset, which comprises a total of 2.5B data points. This dataset is composed of a vast number of individual time series with widely varying lengths. Figure 8 illustrates the distribution of these lengths. As the histogram shows, our pre-training curriculum follows a long-tailed distribution. It is dominated by a large volume of shorter series, which teaches the model common, localized patterns. Critically, it also includes a significant and diverse tail of longer series, with lengths extending up to 10,000 time steps. Regarding the number of variates for multivariable model, our training data includes a diverse mixture of dimensions (1, 8, 12, 16, and 20 variables) to ensure robustness across varying multivariate settings.

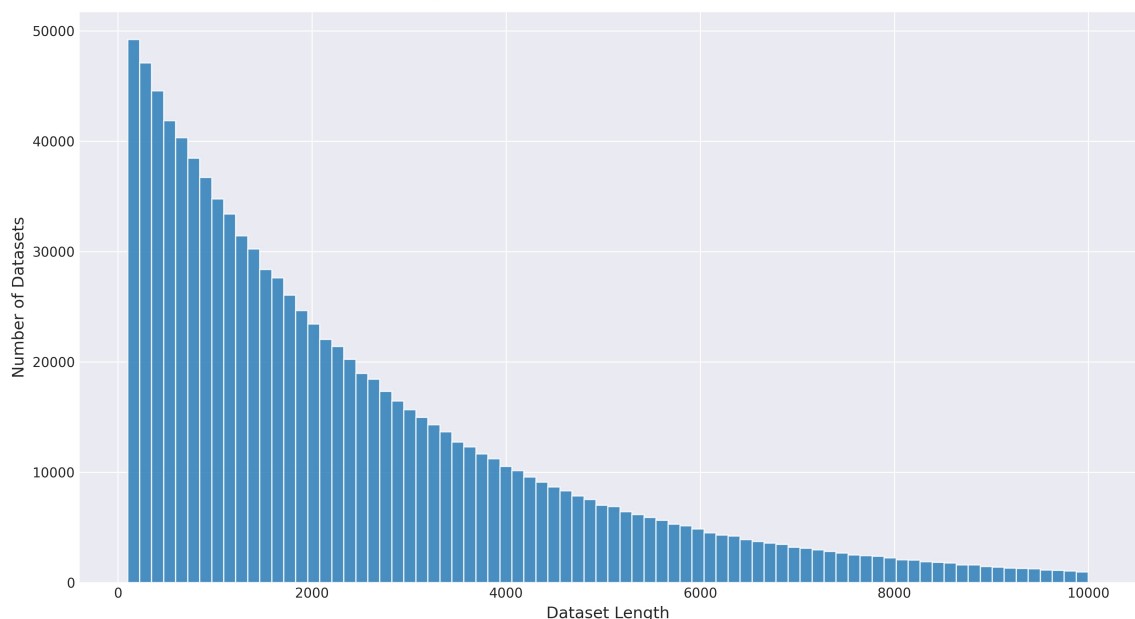

Figure 8: Distribution of time series lengths within our 2.5B point synthetic pre-training dataset. The histogram follows a long-tailed distribution, demonstrating that the pre-training curriculum is highly diverse. It contains a large volume of shorter series (under 4,000 steps) to teach common patterns, while also including a significant number of longer series (up to 10,000 steps) to ensure the model learns long-range temporal dependencies.

# F EXTENDED RESULTS AND ANALYSIS

## F.1 FULL RESULTS

This section provides the complete, unabridged performance results that support the summary claims made in the main text.

**Experimental Setup and Zero-Shot Protocol**  Our evaluation involves two distinct comparisons: (1) a direct **zero-shot** comparison against other foundation models, and (2) a comparison against **full-shot** models where specialized baselines are trained on the target data. We emphasize that **our model remains strictly zero-shot in all settings**, rigorously testing its out-of-the-box performance. In a direct comparison against other leading foundation models, our model demonstrates clear state-of-the-art (SOTA) capabilities. On 64 evaluation cases (16 datasets × 4 metrics), our model ranks first in **41** cases and second in **6**. Even more impressively, when benchmarked against full-shot models that have the advantage of training on target-specific data, our zero-shot model proves highly competitive, securing the top rank in **32** evaluations and second place in another **3**.

**Context Window Size Parameters for Main Results**  For the main results presented in Table 1, a context window size was selected for each dataset based on the analysis in Appendix F.3 to optimize performance. Specifically, for most univariate datasets, a window size of **5k** was used. Exceptions were made for the **NAB** dataset, where a smaller **1k** window was found to be optimal, and the long-series **Power** dataset, where the maximum tested **13k** window yielded the best results. For all five multivariate datasets, a context window of **13k** was applied to best capture their complex, inter-channel dependencies.

## F.2 CONTEXTUAL ANALYSIS

To provide a controlled environment for testing our model's contextual understanding, we generated a new suite of specialized, unseen test datasets. The objective was to create two distinct sets: one containing only **point anomalies** and another containing only **contextual anomalies**. This section details their construction process.

**General Construction Process**  Each dataset was generated using the framework described in the main paper, with a crucial modification: only one type of anomaly was injected per set. We first synthesized a complex "normal" baseline series, combining trend, seasonal, and noise components. We then injected specific anomalies into these clean series to create our test cases. These datasets were not used in any part of the pre-training process.

**Point Anomaly Datasets**  Point anomalies are defined as short-term, localized deviations that can be identified by comparing a data point to its immediate neighbors, without requiring long-range context.

**Contextual Anomaly Datasets**  Contextual anomalies are defined as deviations from a long-term, established pattern, such as a disruption in a seasonal cycle. Detecting them requires the model to understand the global structure of the time series.

**Dataset Summary**  The final collection consists of two distinct sets of time series, with their key characteristics summarized in Table 9.

Table 9: Quantitative comparison on specialized Point and Contextual anomaly datasets. The table is grouped by anomaly type, with metrics listed vertically for each. Our model demonstrates competitive performance on point anomalies and substantially superior performance on contextual anomalies across all metrics. Best results in each row are highlighted in **bold**.

| Type | Metric | Our Model | Chronos | DADA | MOMENT | TSPulse | Time MOE | TimesFM |
|---|---|---|---|---|---|---|---|---|
| **Point** | Affiliation-F | **0.912** | 0.899 | 0.823 | 0.886 | 0.733 | 0.793 | 0.889 |
| | F1$_T$ | **0.604** | 0.441 | 0.387 | 0.551 | 0.214 | 0.303 | 0.446 |
| | Standard-F1 | **0.521** | 0.352 | 0.121 | 0.472 | 0.090 | 0.149 | 0.397 |
| | VUS-PR | **0.666** | 0.293 | 0.396 | 0.538 | 0.063 | 0.190 | 0.385 |
| **Contextual** | Affiliation-F | **0.949** | 0.804 | 0.721 | 0.813 | 0.730 | 0.756 | 0.792 |
| | F1$_T$ | **0.781** | 0.174 | 0.227 | 0.284 | 0.210 | 0.269 | 0.154 |
| | Standard-F1 | **0.827** | 0.154 | 0.048 | 0.172 | 0.088 | 0.078 | 0.140 |
| | VUS-PR | **0.879** | 0.108 | 0.132 | 0.169 | 0.056 | 0.063 | 0.101 |

## F.3 WINDOW SIZE SENSITIVITY ANALYSIS

In this section, we show the detailed performance of our model across different context window sizes (from 1k to 13k) for each of the four primary evaluation metrics. The following figures provide a per-dataset breakdown, which supports the summarized analysis presented in the main paper.

The results in Figures 9 through 12 empirically validate our key finding: the optimal context length is task-dependent, and datasets with long-term or complex structures see a clear benefit from larger window sizes.

- **Positive Scaling Trend:** A strong positive correlation between window size and performance is evident for several datasets across most metrics. For example, **Power**, **UCR**, **MGAB**, **SMD**, and **SMAP** all demonstrate a general upward trend, especially on the F1-T, Standard-F, and VUS-PR metrics. This confirms that for these series, a larger context provides a more robust understanding of "normalcy", leading to better anomaly detection.

- **Flat Performance on Short Series:** In contrast, datasets with inherently short time series, such as **YAHOO**, **NEK**, and **TODS**, show a flat performance profile. Their scores remain largely unchanged regardless of the window size. This is the expected behavior, as the length of the time series itself acts as the natural context limit, and the model correctly avoids computing attention over non-existent data.

- **Task-Dependent Fluctuation:** Other datasets, like **NAB**, **PSM**, and **SWaT**, exhibit more fluctuating or non-monotonic behavior. For these, the ideal window size varies, with performance sometimes peaking at intermediate lengths. This underscores the value of having a flexible architecture where the context window can be tuned as a hyperparameter to achieve optimal results for a specific task.

- **Metric Sensitivity:** It is also noteworthy that the 'Affiliation-F' metric (Figure 9) generally appears less sensitive to changes in window size compared to the other three metrics. This suggests that while a larger context may be crucial for the precise, point-wise accuracy captured by metrics like 'Standard-F1', the broader, proximity-based evaluation of 'Affiliation-F' is less affected.

These detailed results collectively reinforce the conclusion that the model's ability to handle extensive, variable-length contexts is a powerful and necessary feature for a general-purpose time-series anomaly detector.

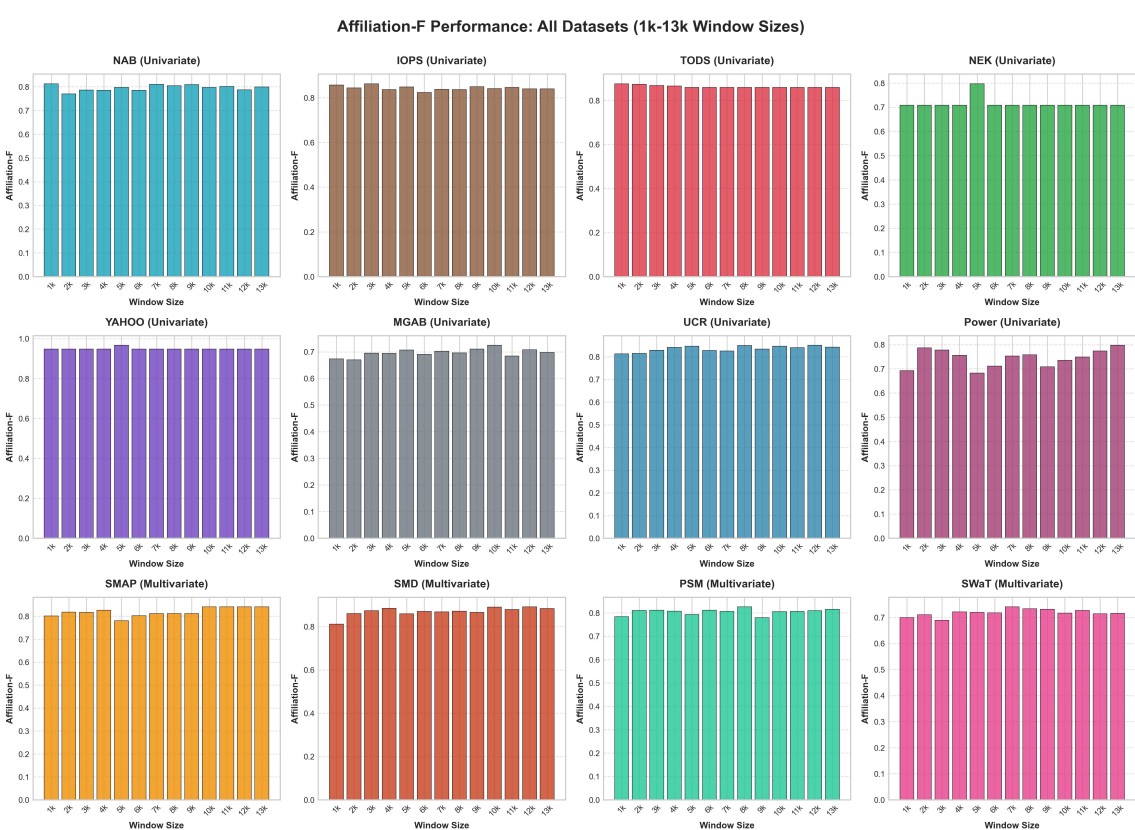

Figure 9: Affiliation-F performance across all 14 datasets as a function of the input window size from 1k to 13k.

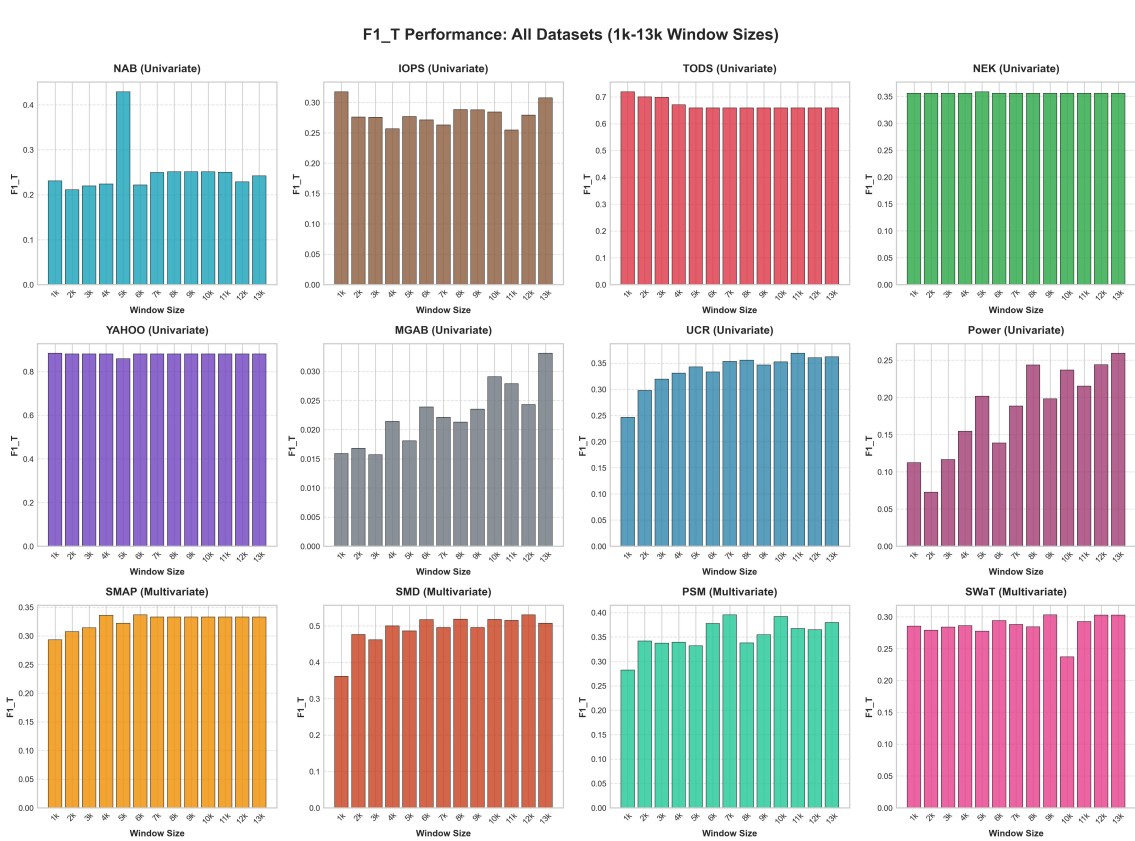

Figure 10: $F1_T$ performance across all 14 datasets as a function of the input window size from 1k to 13k.

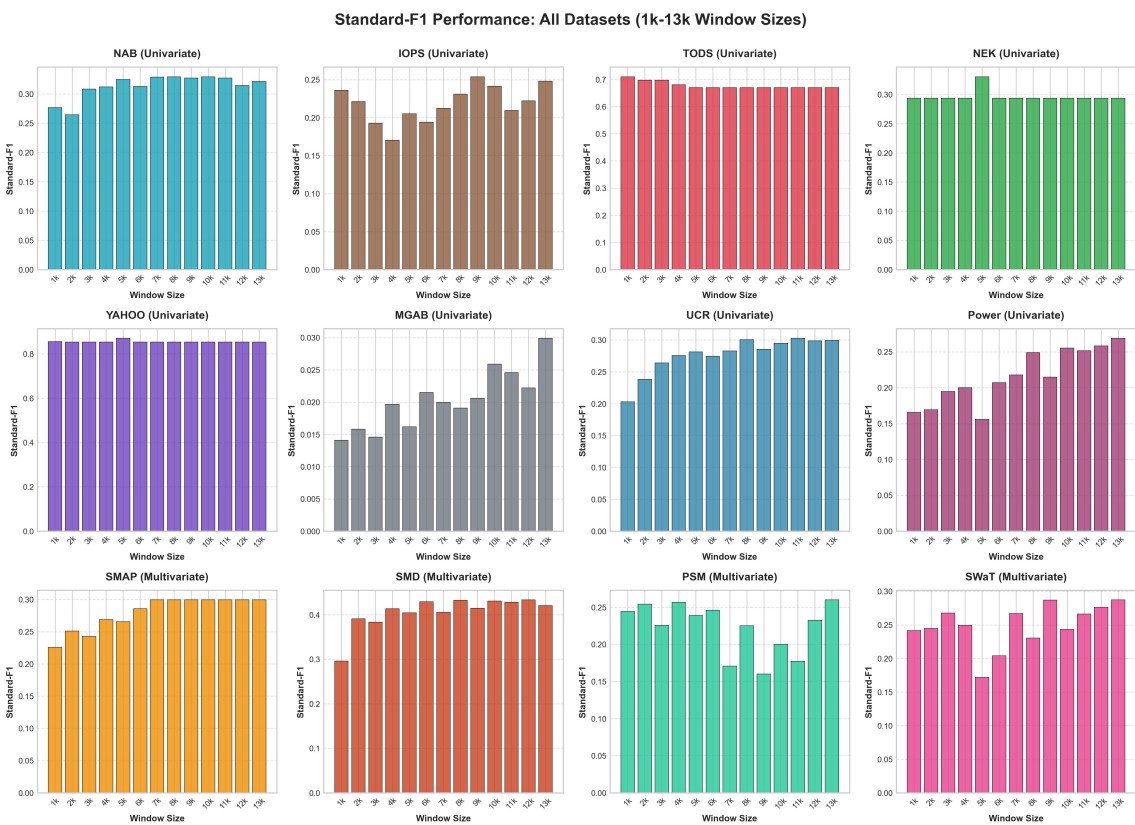

Figure 11: Standard-F1 performance across all 14 datasets as a function of the input window size from 1k to 13k.

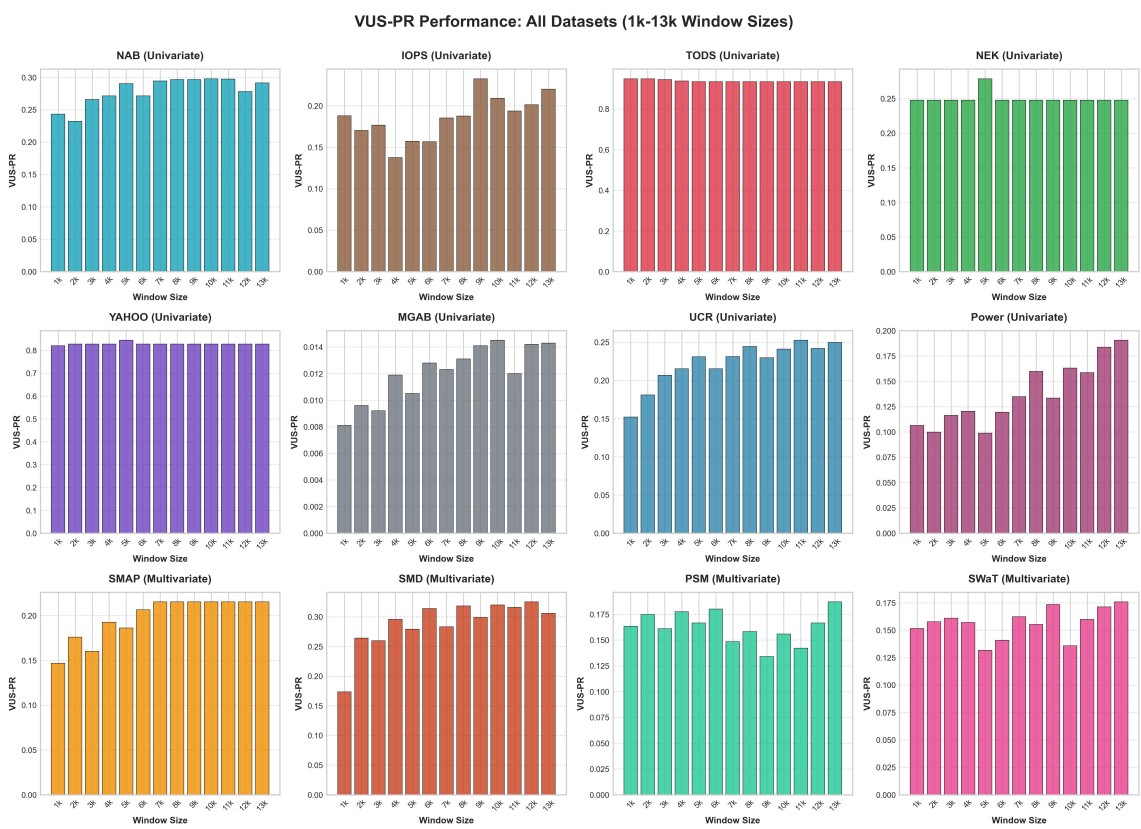

Figure 12: VUS-PR performance across all 14 datasets as a function of the input window size from 1k to 13k.

Table 10: Impact of Context Window Size: A comprehensive performance comparison of our model with varying context window sizes across all univariate and multivariate benchmark datasets. Best results for each dataset-metric combination are highlighted in **red**, and second-best are in blue.

| Metric | Model | Univariate Datasets | | | | | | | | | Multivariate Datasets | | | | | Total 1st | Total 2nd |
|---|---|---|---|---|---|---|---|---|---|---|---|---|---|---|---|---|---|
| | | IOPS | MGAB | NAB | NEK | Power | SED | TODS | UCR | YAHOO | MSL | PSM | SMAP | SMD | SWaT | | |
| Affiliation-F | 1k | 85.64 | 67.32 | 81.28 | 70.87 | 69.30 | 87.07 | 87.56 | 81.34 | 94.73 | 78.50 | 78.36 | 80.16 | 81.11 | 69.93 | 02 | 02 |
| | 2k | 84.41 | 66.97 | 77.01 | 70.87 | 78.73 | 91.51 | 87.24 | 81.54 | 94.79 | 79.63 | 81.04 | 81.80 | 85.99 | 71.00 | 00 | 04 |
| | 3k | 86.25 | 69.50 | 78.60 | 70.87 | 77.83 | 92.15 | 86.73 | 82.88 | 94.79 | 80.27 | 81.12 | 81.77 | 87.30 | 68.87 | 01 | 02 |
| | 4k | 83.61 | 69.41 | 78.48 | 70.87 | 75.57 | 94.57 | 86.48 | 84.12 | 94.79 | 80.75 | 80.75 | 82.68 | 88.48 | 72.12 | 00 | 04 |
| | 5k | 84.79 | 70.69 | 79.70 | 79.73 | 68.30 | 96.87 | 85.91 | 84.63 | 96.65 | 79.90 | 79.35 | 78.04 | 85.94 | 71.93 | 02 | 00 |
| | 6k | 82.30 | 69.08 | 78.49 | 70.87 | 71.10 | 96.92 | 85.91 | 82.75 | 94.79 | 79.90 | 81.11 | 80.25 | 87.07 | 71.79 | 00 | 02 |
| | 7k | 83.72 | 70.24 | 81.00 | 70.87 | 75.32 | 96.92 | 85.91 | 82.55 | 94.79 | 79.90 | 80.67 | 81.23 | 86.76 | 74.03 | 01 | 03 |
| | 8k | 83.65 | 69.58 | 80.40 | 70.87 | 75.77 | 95.22 | 85.91 | 84.95 | 94.79 | 79.90 | 82.56 | 81.23 | 87.14 | 73.29 | 01 | 04 |
| | 9k | 84.90 | 71.07 | 80.97 | 70.87 | 70.83 | 97.66 | 85.91 | 83.46 | 94.79 | 79.90 | 77.94 | 81.23 | 86.59 | 73.10 | 00 | 04 |
| | 10k | 84.02 | 72.45 | 79.74 | 70.87 | 73.48 | 97.77 | 85.91 | 84.70 | 94.79 | 81.13 | 80.55 | 84.10 | 89.03 | 71.65 | 04 | 03 |
| | 11k | 84.54 | 68.39 | 80.08 | 70.87 | 74.96 | 97.36 | 85.91 | 84.03 | 94.79 | 81.13 | 80.62 | 84.10 | 87.85 | 72.69 | 02 | 02 |
| | 12k | 83.91 | 70.79 | 78.67 | 70.87 | 77.39 | 95.27 | 85.91 | 85.06 | 94.79 | 81.13 | 80.94 | 84.10 | 89.09 | 71.41 | 04 | 02 |
| | 13k | 83.98 | 69.82 | 79.96 | 70.87 | 79.73 | 95.44 | 85.91 | 84.24 | 94.79 | 81.13 | 81.61 | 84.10 | 88.29 | 71.55 | 03 | 03 |
| F1_T | 1k | 31.77 | 01.59 | 23.07 | 35.60 | 11.22 | 51.63 | 71.91 | 24.66 | 88.38 | 39.60 | 28.22 | 29.30 | 36.13 | 28.53 | 03 | 01 |
| | 2k | 27.60 | 01.68 | 21.11 | 35.60 | 07.27 | 57.90 | 69.97 | 29.79 | 88.12 | 40.45 | 34.16 | 30.72 | 47.63 | 27.91 | 00 | 03 |
| | 3k | 27.58 | 01.57 | 21.94 | 35.60 | 11.66 | 58.75 | 69.80 | 31.96 | 88.12 | 39.99 | 33.75 | 31.43 | 46.19 | 28.39 | 00 | 02 |
| | 4k | 25.68 | 02.14 | 22.36 | 35.60 | 15.44 | 64.61 | 67.01 | 33.10 | 88.12 | 42.47 | 33.95 | 33.59 | 49.98 | 28.60 | 01 | 03 |
| | 5k | 27.70 | 01.81 | 42.90 | 35.87 | 20.17 | 69.43 | 65.89 | 34.30 | 85.86 | 41.54 | 33.19 | 32.22 | 48.64 | 27.75 | 02 | 01 |
| | 6k | 27.13 | 02.39 | 22.17 | 35.60 | 13.87 | 72.40 | 65.85 | 33.31 | 88.12 | 41.54 | 37.75 | 33.66 | 51.73 | 29.42 | 01 | 03 |
| | 7k | 26.31 | 02.21 | 24.95 | 35.60 | 18.83 | 70.80 | 65.85 | 35.33 | 88.12 | 41.54 | 39.53 | 33.28 | 49.54 | 28.80 | 01 | 03 |
| | 8k | 28.84 | 02.13 | 25.09 | 35.60 | 24.36 | 70.86 | 65.85 | 35.57 | 88.12 | 41.54 | 33.77 | 33.28 | 51.83 | 28.44 | 00 | 04 |
| | 9k | 28.78 | 02.35 | 25.12 | 35.60 | 19.80 | 74.65 | 65.85 | 34.68 | 88.12 | 41.54 | 35.46 | 33.28 | 49.54 | 30.30 | 02 | 04 |
| | 10k | 28.42 | 02.91 | 25.08 | 35.60 | 23.66 | 73.61 | 65.85 | 35.27 | 88.12 | 41.54 | 39.22 | 33.28 | 51.77 | 23.73 | 00 | 06 |
| | 11k | 25.45 | 02.79 | 24.99 | 35.60 | 21.51 | 71.50 | 65.85 | 36.89 | 88.12 | 41.54 | 36.75 | 33.28 | 51.52 | 29.27 | 01 | 03 |
| | 12k | 27.90 | 02.43 | 22.86 | 35.60 | 24.40 | 69.70 | 65.85 | 36.03 | 88.12 | 41.54 | 36.48 | 33.28 | 53.02 | 30.28 | 01 | 05 |
| | 13k | 30.77 | 03.31 | 24.19 | 35.60 | 25.92 | 70.33 | 65.85 | 36.23 | 88.12 | 41.54 | 37.98 | 33.28 | 50.73 | 30.28 | 02 | 06 |
| Standard-F1 | 1k | 23.61 | 01.41 | 27.68 | 29.38 | 16.57 | 51.89 | 71.00 | 20.30 | 85.54 | 27.22 | 24.47 | 22.56 | 29.59 | 24.17 | 01 | 02 |
| | 2k | 22.10 | 01.58 | 26.45 | 29.38 | 16.92 | 58.24 | 69.76 | 23.84 | 85.33 | 28.58 | 25.43 | 25.11 | 39.08 | 24.48 | 00 | 02 |
| | 3k | 19.26 | 01.46 | 30.84 | 29.38 | 19.46 | 59.15 | 69.74 | 26.39 | 85.33 | 29.48 | 22.58 | 24.27 | 38.33 | 26.76 | 00 | 01 |
| | 4k | 17.04 | 01.97 | 31.20 | 29.38 | 20.02 | 65.02 | 68.05 | 27.58 | 85.33 | 30.66 | 25.68 | 26.89 | 41.35 | 24.96 | 01 | 02 |
| | 5k | 20.51 | 01.62 | 32.46 | 33.05 | 15.59 | 69.88 | 67.01 | 28.13 | 87.02 | 29.95 | 23.91 | 26.54 | 40.45 | 17.18 | 02 | 01 |
| | 6k | 19.40 | 02.15 | 31.27 | 29.38 | 20.70 | 72.71 | 67.01 | 27.46 | 85.33 | 29.95 | 24.61 | 28.57 | 42.90 | 20.40 | 00 | 03 |
| | 7k | 21.23 | 02.00 | 32.90 | 29.38 | 21.76 | 71.16 | 67.01 | 28.28 | 85.33 | 29.95 | 17.09 | 29.93 | 40.56 | 26.70 | 01 | 02 |
| | 8k | 23.08 | 01.91 | 32.91 | 29.38 | 24.88 | 71.26 | 67.01 | 30.04 | 85.33 | 29.95 | 22.50 | 29.93 | 43.25 | 23.07 | 01 | 05 |
| | 9k | 25.38 | 02.06 | 32.70 | 29.38 | 21.47 | 75.00 | 67.01 | 28.53 | 85.33 | 29.95 | 16.00 | 29.93 | 41.44 | 28.69 | 03 | 03 |
| | 10k | 24.14 | 02.59 | 32.94 | 29.38 | 25.52 | 74.07 | 67.01 | 29.50 | 85.33 | 29.95 | 20.01 | 29.93 | 43.08 | 24.36 | 02 | 04 |
| | 11k | 20.96 | 02.46 | 32.72 | 29.38 | 25.15 | 71.89 | 67.01 | 30.26 | 85.33 | 29.95 | 17.74 | 29.93 | 42.80 | 26.63 | 02 | 02 |
| | 12k | 22.21 | 02.22 | 31.45 | 29.38 | 25.84 | 70.16 | 67.01 | 29.85 | 85.33 | 29.95 | 23.27 | 29.93 | 43.33 | 27.63 | 02 | 03 |
| | 13k | 24.81 | 02.99 | 32.13 | 29.38 | 26.88 | 70.76 | 67.01 | 29.92 | 85.33 | 29.95 | 26.00 | 29.93 | 42.05 | 28.73 | 05 | 03 |
| VUS-PR | 1k | 18.80 | 0.8100 | 24.32 | 24.78 | 10.64 | 48.68 | 94.89 | 15.22 | 82.02 | 17.45 | 16.33 | 14.68 | 17.35 | 15.16 | 01 | 01 |
| | 2k | 17.03 | 0.9600 | 23.23 | 24.78 | 09.97 | 59.79 | 94.86 | 18.11 | 82.78 | 18.23 | 17.48 | 17.59 | 26.41 | 15.77 | 00 | 03 |
| | 3k | 17.66 | 0.9200 | 26.58 | 24.78 | 11.64 | 65.57 | 94.46 | 20.69 | 82.78 | 18.94 | 16.10 | 16.02 | 25.97 | 16.09 | 00 | 02 |
| | 4k | 13.74 | 01.19 | 27.13 | 24.78 | 12.05 | 73.53 | 93.80 | 21.53 | 82.78 | 20.45 | 17.76 | 19.27 | 29.57 | 15.70 | 01 | 02 |
| | 5k | 15.71 | 01.05 | 29.02 | 27.88 | 09.88 | 80.75 | 93.46 | 23.09 | 84.41 | 19.61 | 16.67 | 18.62 | 27.89 | 13.17 | 02 | 01 |
| | 6k | 15.67 | 01.28 | 27.13 | 24.78 | 11.96 | 86.59 | 93.46 | 21.52 | 82.78 | 19.61 | 18.01 | 20.66 | 31.37 | 14.08 | 00 | 05 |
| | 7k | 18.53 | 01.23 | 29.43 | 24.78 | 13.50 | 84.32 | 93.14 | 23.14 | 82.78 | 19.61 | 14.85 | 21.53 | 28.31 | 16.24 | 01 | 03 |
| | 8k | 18.75 | 01.31 | 29.66 | 24.78 | 15.98 | 86.40 | 93.46 | 24.45 | 82.78 | 19.61 | 15.82 | 21.53 | 31.83 | 15.53 | 01 | 03 |
| | 9k | 23.26 | 01.41 | 29.67 | 24.78 | 13.35 | 92.86 | 93.46 | 22.99 | 82.78 | 19.61 | 13.39 | 21.53 | 29.91 | 17.35 | 03 | 04 |
| | 10k | 20.91 | 01.45 | 29.79 | 24.78 | 16.30 | 91.36 | 93.46 | 24.10 | 82.78 | 19.61 | 15.59 | 21.53 | 32.00 | 13.59 | 03 | 05 |
| | 11k | 19.38 | 01.20 | 29.74 | 24.78 | 15.86 | 87.48 | 93.46 | 25.26 | 82.78 | 19.61 | 14.22 | 21.53 | 31.59 | 16.00 | 02 | 04 |
| | 12k | 20.13 | 01.42 | 27.82 | 24.78 | 18.38 | 85.60 | 93.46 | 24.20 | 82.78 | 19.61 | 16.65 | 21.53 | 32.51 | 17.13 | 02 | 04 |
| | 13k | 22.00 | 01.43 | 29.14 | 24.78 | 19.05 | 86.77 | 93.46 | 24.99 | 82.78 | 19.61 | 18.70 | 21.53 | 30.58 | 17.58 | 04 | 06 |

## F.4 ANALYSIS FOR ABLATION STUDY ON PRE-TRAINING DATA

In this section, we provide a detailed, per-dataset analysis of the ablation study on our pre-training data, which supports the weighted average results discussed in the main paper.

**Experimental Setup** To rigorously validate our data generation framework, we conducted a controlled experiment. We trained our model on three distinct pre-training datasets, each standardized to 350M data points. A completely identical training protocol—including the same model architecture, hyperparameters, and number of training epochs—was used for each run to isolate the impact of the data itself. The three datasets were:

1. **Our Synthetic Data:** Generated entirely by our proposed framework, including our custom, context-aware anomaly injection techniques.

2. **Our Synthetic Data + DADA Injection:** The same base time series as above, but with anomalies introduced using the generic DADA injection method described by Shentu et al. (2024).

3. **Real-world Data + DADA Injection:** A dataset created by applying DADA injection to real-world time series from the Monash forecasting repository Godahewa et al..

**Univariate Setting Justification** This experiment was conducted exclusively in a univariate setting for two primary reasons. First, the large-scale real-world dataset available for creating a comparable baseline (*Monash*) is predominantly comprised of univariate series. Second, the DADA anomaly injection technique, as described in their original paper, is designed for and implemented on univariate time series. This constraint ensures a fair and direct comparison between the different data generation strategies.

**Detailed Results and Analysis** Table 11 presents the full, per-dataset results for the nine univariate benchmarks used in this study. These results offer a more granular view than the weighted averages in the main text.

The per-dataset results first confirm the main conclusion: the model pre-trained on augmented real-world data demonstrates significantly inferior performance across most datasets and metrics, particularly on the F1-based scores. This validates our hypothesis that a purpose-built synthetic curriculum is superior to simply augmenting existing real-world series.

More interestingly, the table reveals the nuances behind the weighted average comparison between our two synthetic strategies. While the "Our Synthetic Data + DADA Injection" strategy secures more first-place finishes on a per-dataset basis (e.g., strong performance on IOPS and NAB), our custom injection method ("Our Synthetic Data") achieves higher scores on other key datasets like TODS and YAHOO. The fact that our method yields a better weighted average on $F1_T$ and Standard-F1 (as mentioned in the main paper) suggests that its performance is more stable or that it excels on datasets that have a greater influence on the overall average. This highlights a critical trade-off: DADA's generic injections can be highly effective for specific data profiles, whereas our context-aware injection methods appear to provide a more consistent and robust performance across a wider range of scenarios, leading to better overall performance on the most stringent metrics. Ultimately, this detailed analysis reinforces that our end-to-end synthetic data generation framework provides the most effective training curriculum for achieving state-of-the-art zero-shot performance.

Table 11: Performance comparison across different models and metrics. Best results for each dataset-metric combination are highlighted in **red**, and second-best are in blue.

| Metric | Model | Datasets | | | | | | | | | Total 1st | Total 2nd |
|---|---|---|---|---|---|---|---|---|---|---|---|---|
| | | IOPS | MGAB | NAB | NEK | Power | SED | TODS | UCR | YAHOO | | |
| **Affiliation-F** | **Our Synthetic Data** | 89.37 | 68.55 | 88.74 | 79.27 | 63.31 | 89.24 | **83.26** | **83.88** | 92.35 | 02 | 05 |
| | **Our Synthetic Data + DADA Injection** | **90.50** | **68.67** | **88.80** | 79.60 | **78.52** | **94.04** | 80.92 | 82.43 | **93.63** | 06 | 03 |
| | **Real-world Data + DADA Injection** | 70.98 | 67.59 | 81.99 | **82.61** | 70.73 | 62.18 | 55.56 | 74.80 | 68.41 | 01 | 01 |
| **F1_T** | **Our Synthetic Data** | 36.08 | 01.08 | 25.69 | 55.06 | 06.23 | 53.92 | **58.76** | 31.59 | **85.99** | 02 | 04 |
| | **Our Synthetic Data + DADA Injection** | **53.11** | **01.91** | **34.91** | 59.59 | **13.16** | **61.45** | 57.66 | **32.74** | 68.57 | 06 | 03 |
| | **Real-world Data + DADA Injection** | 06.80 | 01.35 | 23.18 | **70.65** | 10.01 | 0.1700 | 09.67 | 05.25 | 05.37 | 01 | 02 |
| **Standard-F1** | **Our Synthetic Data** | 30.63 | 01.13 | 30.12 | 50.89 | 17.39 | 54.18 | **61.66** | 24.38 | **82.16** | 02 | 06 |
| | **Our Synthetic Data + DADA Injection** | **46.46** | **01.74** | **34.65** | 48.90 | **19.53** | **61.43** | 59.62 | **26.20** | 65.66 | 06 | 02 |
| | **Real-world Data + DADA Injection** | 11.02 | 01.32 | 26.17 | **66.97** | 17.09 | 03.08 | 11.99 | 03.83 | 03.53 | 01 | 01 |
| **VUS-PR** | **Our Synthetic Data** | 24.07 | 0.7000 | 27.16 | 48.66 | 09.66 | 61.63 | 88.62 | **20.59** | 74.68 | 01 | 07 |
| | **Our Synthetic Data + DADA Injection** | **45.24** | **0.9500** | **32.22** | 48.52 | **12.60** | **65.09** | **93.48** | 19.17 | **75.67** | 07 | 01 |
| | **Real-world Data + DADA Injection** | 08.88 | 0.6400 | 24.54 | **69.75** | 11.09 | 06.69 | 45.82 | 02.12 | 12.18 | 01 | 01 |

## F.5 ADDITIONAL ANALYSIS FOR SCALING LAWS

In this section, we provide the full, per-dataset results for our investigation into the impact of pre-training data scale on model performance. These granular results support the weighted average trends and conclusions presented in the main paper.

**Experimental Setup** To investigate the effect of pre-training data scale, we trained our model on increasingly larger subsets of our synthetic dataset: 350M, 700M, and the full 2.5B data points. Each model was trained using an identical protocol to isolate the effect of the data volume. Due to the predominantly univariate nature of the large-scale datasets used to construct our synthetic curriculum, this scaling analysis was conducted exclusively on the nine univariate benchmark datasets.

**Detailed Results and Analysis** The results, presented in Table 12, confirm the existence of a strong and positive scaling law. As the volume of pre-training data increases, the model's overall performance consistently improves. This is most evident in the "Total 1st" place rankings, where the full **2.5B** model overwhelmingly outperforms the smaller models across all four metrics, securing the top rank in 24 out of 36 total evaluations. While the general trend is clear, the per-dataset results reveal some interesting nuances. The scaling is not perfectly monotonic for every dataset-metric combination. For instance, the **700M** model achieves the best 'F1-T score on the IOPS and YAHOO datasets, and the **350M** model shows strong performance on the NEK dataset. These variations likely stem from a combination of dataset-specific characteristics and the inherent stochasticity of the training process. However, the dominant performance of the 2.5B model in the aggregate rankings strongly validates our approach of leveraging a large-scale pre-training curriculum. It also suggests that the model's capabilities are not yet saturated, with the potential for further improvement with even more data.

Table 12: Performance comparison across different models and metrics. Best results for each dataset-metric combination are highlighted in red, and second-best are in blue.

| Metric | Model | Datasets | | | | | | | | | Total 1st | Total 2nd |
| | | IOPS | MGAB | NAB | NEK | Power | SED | TODS | UCR | YAHOO | | |
|---|---|---|---|---|---|---|---|---|---|---|---|---|
| **Affiliation-F** | 2.5B | 84.79 | **70.69** | 79.70 | **79.73** | 68.30 | **96.87** | **85.91** | **84.63** | **96.65** | 06 | 01 |
| | 700M | 86.46 | 68.87 | **86.43** | 72.74 | **73.02** | 92.07 | 83.95 | 82.23 | 93.74 | 02 | 05 |
| | 350M | **88.24** | 68.19 | 86.21 | 79.27 | 63.30 | 89.21 | 83.22 | 82.49 | 92.35 | 01 | 03 |
| **F1_T** | 2.5B | 27.70 | **01.81** | **42.90** | 35.87 | **20.17** | 69.43 | 65.89 | **34.30** | 85.86 | 06 | 00 |
| | 700M | **40.56** | 01.49 | 27.82 | 42.31 | 05.13 | 59.02 | 64.07 | 28.88 | **86.88** | 02 | 05 |
| | 350M | 36.08 | 01.08 | 25.69 | **55.06** | 06.23 | 53.92 | 58.76 | 31.59 | 85.99 | 01 | 04 |
| **Standard-F1** | 2.5B | 20.51 | **01.62** | **32.46** | 33.05 | 15.59 | **69.88** | **67.01** | **28.13** | **87.02** | 06 | 01 |
| | 350M | **30.63** | 01.13 | 30.12 | **50.89** | **17.39** | 54.18 | 61.66 | 24.38 | 82.16 | 03 | 01 |
| | 700M | 30.08 | 01.51 | 32.17 | 42.68 | 12.63 | 59.31 | 66.45 | 22.71 | 83.92 | 00 | 07 |
| **VUS-PR** | 2.5B | 15.71 | **01.05** | 29.02 | 27.88 | **09.88** | **80.75** | **93.46** | **23.09** | **84.41** | 06 | 01 |
| | 700M | **26.51** | 0.7200 | **30.18** | 36.27 | 08.51 | 67.37 | 91.62 | 18.35 | 78.48 | 02 | 05 |
| | 350M | 24.07 | 0.7000 | 27.16 | **48.66** | 09.66 | 61.63 | 88.62 | 20.59 | 74.68 | 01 | 03 |

## F.6 ADDITIONAL ANALYSIS FOR ABLATION STUDY

**Ablation Study of Anomaly/Reconstruction Head** To analyze the contribution of our model's key components, we conducted an ablation study on the representative UCR dataset. We compare three distinct versions: (1) the **Full Model**, which includes both our dedicated anomaly head and the auxiliary reconstruction head; (2) **Anomaly Head Only**, where the reconstruction task is removed; and (3) **Reconstruction Only**, where scores are derived from reconstruction error. The results, presented in Table 13, reveal the critical synergy between our model's two heads. The Full Model substantially outperforms both ablated versions

across all metrics. Interestingly, the Anomaly Head Only model sees a near-total collapse in performance, performing significantly worse than even the Reconstruction Only model. This result strongly suggests that the auxiliary reconstruction task is not merely helpful, but essential. It provides a foundational learning signal that teaches the model to understand the basic structure of the time series, without which the dedicated anomaly head cannot function effectively. Therefore, our architecture's state-of-the-art performance stems directly from the combination of these two components: the reconstruction head learns the representation, and the anomaly head refines it to achieve superior detection capabilities.

Table 13: Ablation study results on the UCR dataset. The synergy of the Full Model provides the best performance, while the reconstruction head proves essential for the anomaly head to function effectively.

| Model Version | Affiliation-F | $F1_T$ | Standard-F1 | VUS-PR |
|---|---|---|---|---|
| **Full Model** | **0.846** | **0.343** | **0.281** | **0.231** |
| Anomaly Head Only | 0.709 | 0.043 | 0.024 | 0.014 |
| Reconstruction Only | 0.742 | 0.109 | 0.075 | 0.044 |

## G  STATISTICAL ANALYSIS SUMMARY

| Scenario | Affiliation-F | F1_T | Standard-F1 | VUS-PR | Avg Rank |
|---|---|---|---|---|---|
| **Zero-shot Univariate** | 2nd (MR=2.73, M=0.844) | 1st (MR=2.45, M=0.414) | 1st (MR=2.64, M=0.393) | 1st (MR=2.27, MD=0.243) | **1.8 / 7** |
| **Zero-shot Multivariate** | 1st (MR=1.2, M=0.829) | 1st (MR=1.2, M=0.397) | 1st (MR=1.2, M=0.322) | 1st (MR=1.4, M=0.233) | **1.2 / 7** |
| **Full-shot Univariate** | 1st (MR=1.82, M=0.844) | 1st (MR=2.27, MD=0.350) | 1st (MR=2.64, MD=0.320) | 1st (MR=2.55, MD=0.243) | **2.3 / 9** |
| **Full-shot Multivariate** | 4th (MR=3.2, MD=0.816) | 4th (MR=3.2, M=0.397) | 4th (MR=4.4, M=0.322) | 4th (MR=3.6, M=0.233) | **3.6 / 9** |

Table 14: TimeRCD Performance Summary across All Scenarios. Legend: MR = Mean Rank, M = Mean, MD = Median

TimeRCD demonstrates consistent state-of-the-art performance across evaluation scenarios (Table 14). The model achieves first place on 10 out of 16 metric-scenario combinations, with particularly strong results in zero-shot settings (average rank 1.2-1.8 out of 7 models). In full-shot scenarios, TimeRCD maintains competitive performance, ranking first on univariate tasks (average rank 2.3 out of 9 models) and fourth on multivariate tasks (average rank 3.6 out of 9 models). The results indicate that TimeRCD's architecture successfully generalizes across both univariate and multivariate time series without requiring domain-specific training. Details ranking of each metric and setting can be found from Figure 13 to Figure 16.

## H  EFFICIENCY AND SCALABILITY OF TIMERCD

This appendix provides additional profiling results for TimeRCD in comparison with two strong baselines, TimeMOE and DADA. The goal is to characterize the computational feasibility, scalability, and memory behavior of TimeRCD across a wide range of sequence lengths, including regimes beyond those reported in the main text.

TimeRCD is evaluated on sequences up to 15,000 timesteps. For a sequence of 10,000 timesteps, TimeRCD achieves an inference latency of 21.8 ms, corresponding to a speedup of 12.6× over TimeMOE and 95.2× over DADA under the same hardware and profiling setup. Even at 15,000 timesteps, the inference time remains 86.22 ms, which is compatible with real-time or near real-time monitoring requirements in many practical applications. Empirically, the observed time complexity is sub-linear in sequence length and substantially more favorable than the quadratic behavior of standard Transformer architectures.

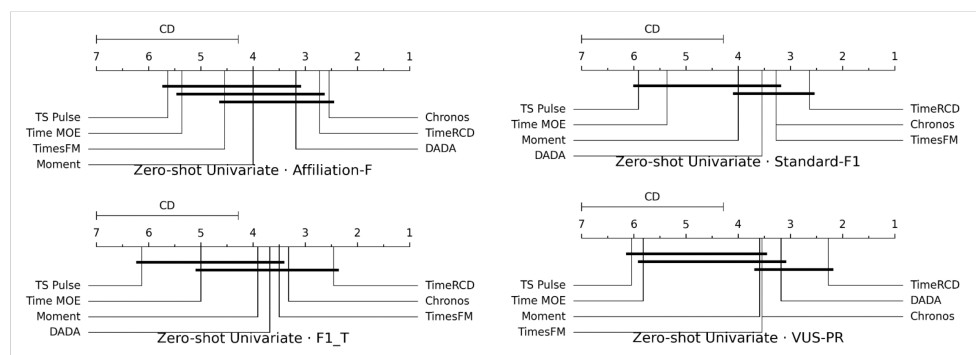

Figure 13: Statistical ranking visualization for zero-shot univariate anomaly detection.

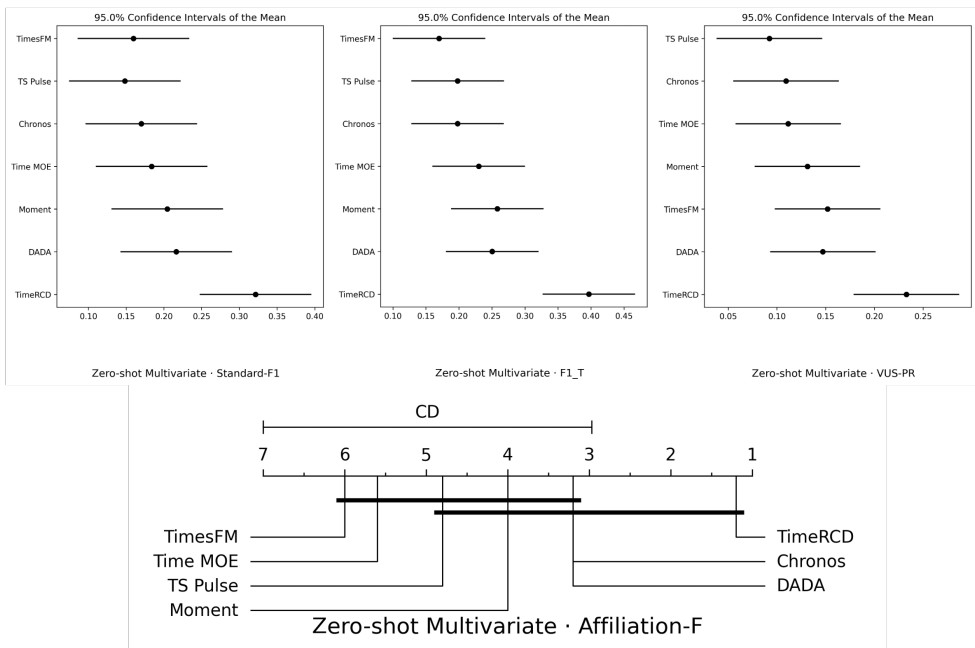

Figure 14: Statistical ranking visualization for zero-shot multivariate anomaly detection.

In terms of memory, TimeRCD processes the entire input sequence rather than using a fixed-size sliding window. This design provides a longer effective receptive field and richer temporal context at the cost of increased memory usage with sequence length. At 10,000 timesteps, peak memory consumption is approximately 1.5 GB, which remains manageable on widely available GPUs. TimeRCD also uses $3.2\times$ fewer parameters than TimeMOE (35.5M vs. 113.4M), mitigating model storage and deployment overhead while still delivering substantial gains in long-sequence efficiency.

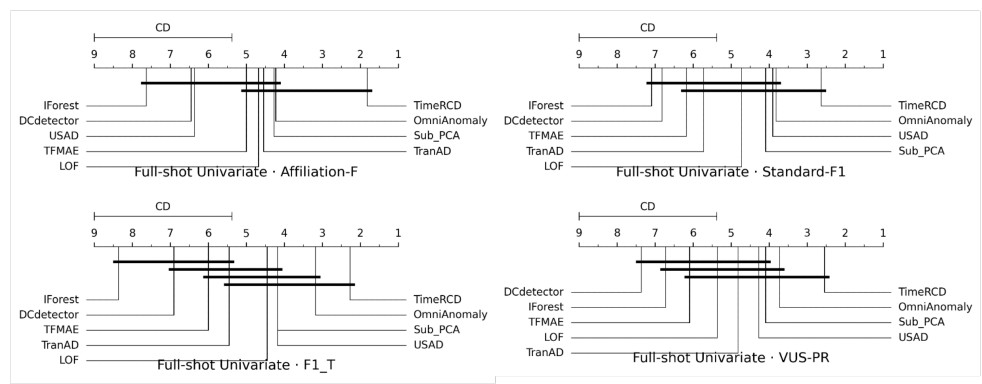

Figure 15: Statistical ranking visualization for full-shot univariate anomaly detection.

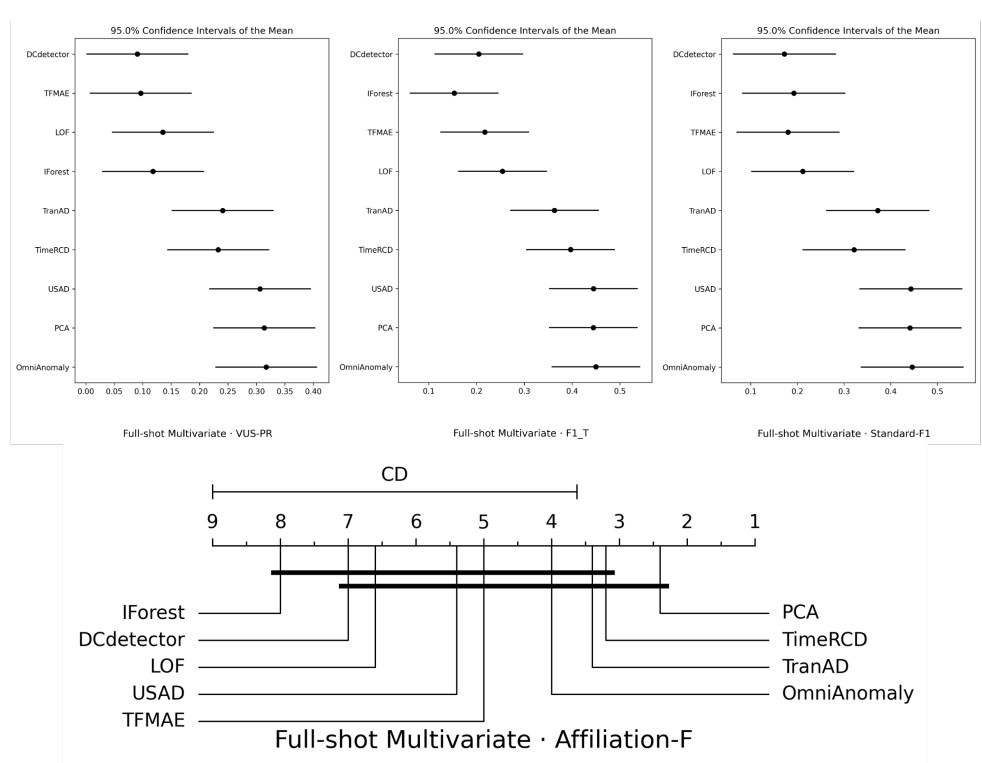

Figure 16: Statistical ranking visualization for full-shot multivariate anomaly detection.

## H.1 INFERENCE TIME (1K–10K TIMESTEPS)

All models are profiled on a single GPU using isolated processes and a global warmup phase to reduce variability. Table 15 reports the end-to-end inference latency as a function of sequence length.

| Model | Seq=1K | Seq=2K | Seq=5K | Seq=10K |
|-------|--------|--------|--------|---------|
| **TimeRCD** | **8.0** | **9.2** | **12.2** | **21.8** |
| TimeMOE | 36.5 | 66.7 | 130.8 | 274.2 |
| DADA | 207.6 | 423.1 | 1042.3 | 2074.0 |

Table 15: Inference time (ms) as a function of sequence length for TimeRCD and baselines.

Across all lengths, TimeRCD exhibits a much slower growth in latency compared to TimeMOE and DADA. This behavior is primarily attributed to its patch-based processing (patch size = 16), which effectively reduces the computational burden for long sequences while preserving temporal coverage.

## H.2 PEAK MEMORY USAGE AND PARAMETER COUNTS

Table 16 summarizes peak memory usage for different sequence lengths together with the parameter counts of each model.

| Model | Seq=1K | Seq=2K | Seq=5K | Seq=10K | Parameters |
|-------|--------|--------|--------|---------|------------|
| **TimeRCD** | **295** | **434** | **858** | **1561** | 35.5M |
| TimeMOE | 462 | 462 | 462 | 462 | 113.4M |
| DADA | 17.8 | 17.8 | 17.8 | 17.8 | 1.8M |

Table 16: Peak memory usage (MB) and parameter counts for TimeRCD and baselines.

TimeMOE and DADA have approximately constant memory usage with respect to sequence length due to their use of fixed-size sliding windows, which limit the effective context. In contrast, TimeRCD processes the full sequence and therefore exhibits increasing memory consumption with longer inputs. Nevertheless, the resulting memory requirements remain well within the capabilities of common consumer-grade GPUs for the sequence lengths considered here.

## H.3 SCALABILITY TO LONG CONTEXTS (UP TO 15K TIMESTEPS)

To further characterize long-context behavior, TimeRCD is evaluated on sequences up to 15,000 timesteps. Table 17 reports the end-to-end inference latency, peak memory usage, and a normalized efficiency measure in milliseconds per 1,000 timesteps.

| Sequence Length | Inference (ms) | Memory (MB) | Efficiency (ms/1k steps) |
|-----------------|----------------|-------------|--------------------------|
| 1,000 | 14.37 | 295 | 14.37 |
| 5,000 | 28.39 | 858 | 5.68 |
| 10,000 | 51.38 | 1561 | 5.14 |
| **15,000** | **86.22** | **2230** | **5.75** |

Table 17: Scalability of TimeRCD up to 15,000 timesteps.

The efficiency metric (milliseconds per 1,000 timesteps) decreases substantially when moving from 1,000 to 5,000 timesteps and then stabilizes around 5–6 ms per 1,000 timesteps for longer sequences. This pattern indicates that fixed overheads are amortized as the sequence length grows, leading to stable per-timestep cost in the long-context regime.

Overall, these extended results corroborate that TimeRCD provides favorable latency and memory characteristics for long-context time series, with sub-linear empirical time scaling and a memory footprint suitable for both research and practical deployments.