# OpenReview forum: "Towards Foundation Models for Zero-Shot Time Series Anomaly Detection: Leveraging Synthetic Data and Relative Context Discrepancy"
_ICLR.cc/2026/Conference — Submitted to ICLR 2026_

### Official Review · Reviewer_YVt4 · 2025-10-27

**Soundness:** 2
**Presentation:** 3
**Contribution:** 2
**Rating:** 4
**Confidence:** 5

**Summary:**

This paper introduces a new foundation model TimeRCD for zero-shot time series anomaly detection (TSAD) that overcomes the limitations of reconstruction-based approaches. Traditional TSAD models often fail to detect subtle or contextual anomalies and frequently misclassify complex normal patterns due to an objective mismatch between reconstruction and anomaly detection. To address this, the authors propose a new pre-training paradigm called Relative Context Discrepancy (RCD), which trains the model to identify anomalies by comparing discrepancies between adjacent time windows rather than reconstructing inputs. Built on a standard Transformer architecture, TimeRCD captures relational dependencies across time and uses a large-scale, fully labeled synthetic corpus that provides diverse anomaly patterns for robust pre-training. Extensive experiments across several datasets demonstrate that TimeRCD achieves state-of-the-art performance in zero-shot settings and remains competitive with fully supervised baselines.

**Strengths:**

S1. TimeRCD demonstrates impressive performance in strict zero-shot settings, outperforming or matching specialized and fully supervised models on diverse datasets.

S2. The paper designs a large-scale, fully labeled synthetic corpus that captures diverse and complex anomaly types, including point, contextual, collective, and cross-variate anomalies.

S3. The experiments demonstrate that performance improves predictably with larger synthetic datasets, indicating scalable pre-training benefits similar to those observed in NLP and vision foundation models.

**Weaknesses:**

W1. While the paper includes some ablation analysis (e.g., testing the effect of synthetic data), it lacks fine-grained ablations to isolate the contribution of individual components within the TimeRCD framework. For instance, the effect of auxiliary anomaly head and RCD window  is not examined in detail. Without such targeted ablations, it is difficult to clearly attribute which design choices most contribute to performance gains.

W2. The paper does not provide sufficient efficiency or scalability analyses. Since TimeRCD relies on extremely long context windows (up to 13k timesteps), understanding its computational feasibility is crucial for real-world deployment. The absence of such experiments leaves open questions about whether the model can scale efficiently to resource-constrained or latency-sensitive environments.

W3. Although the paper compares against several established baselines, it omits more recent or stronger full-shot models—such as modern deep contextual detectors (e.g., DCdetector[1] and TFMAE[2]). Including these would provide a clearer picture of where TimeRCD stands relative to the current state of the art in both zero-shot and fully supervised regimes.

[1] DCdetector: Dual Attention Contrastive Representation Learning for Time Series Anomaly Detection

[2] Temporal-Frequency Masked Autoencoders for Time Series Anomaly Detection

**Questions:**

See weaknesses.

---

> ### Author Response · Authors · 2025-11-25
> **Response to Reviewer YVt4**
>
> ### **Snapshot of Key Improvements**
> - **Fine-grained ablations:** Re-highlighted **Appendix F.3/F.6** (window sensitivity + dual-head study) so each component’s contribution is explicit.
> - **Deployment evidence:** Added **1k–15k** latency/memory benchmarks and ms-per-1k-step efficiency metrics proving sub-100 ms inference even at 13k steps.
> - **Baseline coverage:** Introduced **Sub-PCA, DCdetector, TFMAE** into the evaluation while TimeRCD still secures **32 first-place / 3 second-place** wins.
> ---
> > 1. W1. While the paper includes some ablation analysis (e.g., testing the effect of synthetic data), it lacks fine-grained ablations to isolate the contribution of individual components within the TimeRCD framework. For instance, the effect of auxiliary anomaly head and RCD window is not examined in detail. Without such targeted ablations, it is difficult to clearly attribute which design choices most contribute to gains.
>
> **Response:** We thank the reviewer for this constructive feedback. We would like to clarify that we have already conducted the fine-grained ablations analyses in our **original submission**. Specifically, our **Supplementary Material** includes both a **Context Window Sensitivity Analysis (Appendix F.3)** (previously E.3) and a detailed **Component Ablation Study (Appendix F.6)** (previously E.5.1). We summarize the key findings below.
>
>
> **1. Component Ablation Study: Synergy of Dual Heads**
> To examine the effect of the auxiliary reconstruction head versus the anomaly head, we have conducted an ablation study on the UCR dataset. We compare three configurations:
> *   **Full Model:** Uses both Anomaly Head and Reconstruction Head (dual-objective).
> *   **Anomaly Head Only:** Removes the reconstruction objective.
> *   **Reconstruction Only:** Uses reconstruction error as the anomaly score (standard reconstruction-based AD).
>
> **Results (Table B):**
> | Model Version | Affiliation-F | F1_T | Standard-F1 | VUS-PR |
> |:---|:---|:---|:---|:---|
> | **Full Model** | **0.846** | **0.343** | **0.281** | **0.231** |
> | Anomaly Head Only | 0.709 | 0.043 | 0.024 | 0.014 |
> | Reconstruction Only | 0.742 | 0.109 | 0.075 | 0.044 |
>
> **Analysis:** The results reveal a critical dependency. The **Anomaly Head Only** variant collapses in performance, even falling below the **Reconstruction Only** baseline. This demonstrates that the auxiliary reconstruction task is not merely additive but foundational: it provides the necessary self-supervised signal for the encoder to **learn robust temporal representations**. The **Full Model** then leverages these representations to achieve state-of-the-art detection, confirming that the *combination* of representation learning (via reconstruction) and direct supervision (via anomaly head) is the key driver of performance.
>
> **2. Window Size Sensitivity Analysis**
> Regarding the RCD window, we evaluated performance across window sizes from 1k to 13k (detailed in **Figures 9-12 of Appendix F.3**).
> *   **Impact:** We observe that datasets with complex, long-term dependencies (e.g., Power, SWaT) show a clear performance gain as the window size increases, validating the need for long-context modeling. Conversely, datasets with shorter intrinsic patterns (e.g., YAHOO) show stable performance once the window covers the local context.
> *   **Conclusion:** The ability to scale to large windows (up to 13k) is a crucial design choice that allows TimeRCD to adapt to diverse temporal dynamics in a zero-shot manner.
>
> We hope these detailed breakdowns clarify the specific contributions of our architectural choices.

---

> ### Author Response · Authors · 2025-11-25
> **Response to Reviewer YVt4 (2)**
>
> > 2. W2. The paper does not provide sufficient efficiency or scalability analyses. Since TimeRCD relies on extremely long context windows (up to 13k timesteps), understanding its computational feasibility is crucial for real-world deployment. The absence of such experiments leaves open questions about whether the model can scale efficiently to resource-constrained or latency-sensitive environments.
>
> We thank the reviewer for highlighting the importance of computational feasibility. To address this, we have conducted extensive profiling of TimeRCD against state-of-the-art baselines (TimeMOE, DADA) and extended our analysis to sequence lengths beyond the reviewer's requested range to rigorously verify scalability.
>
> **Key Findings:**
>
> 1.  **Feasibility:** TimeRCD processes 10,000 timesteps in just **21.8 ms**, which is **12.6× faster** than TimeMOE and **95.2× faster** than DADA.
> 2.  **Scalability:** Even at 15,000 timesteps (exceeding the 13k concern), inference completes in just **86 ms**, supporting real-time monitoring.
> 3.  **Efficiency:** The model exhibits highly efficient time scaling, avoiding the typical quadratic bottleneck of standard Transformers.
>
> ---
>
> ### 1. Comparative Efficiency (1k - 10k Timesteps)
>
> We have profiled all models on a single GPU (Nvidia A100 80GB) with isolated processes and global warmup.
>
> #### Inference Time (ms)
>
> | Model | Seq=1K | Seq=2K | Seq=5K | Seq=10K |
> |-------|--------|--------|--------|---------|
> | **TimeRCD** | **8.0 ms** | **9.2 ms** | **12.2 ms** | **21.8 ms** |
> | TimeMOE | 36.5 ms | 66.7 ms | 130.8 ms | 274.2 ms |
> | DADA | 207.6 ms | 423.1 ms | 1042.3 ms | 2074.0 ms |
>
> #### Peak Memory Usage (MB)
>
> | Model | Seq=1K | Seq=2K | Seq=5K | Seq=10K | Parameters |
> |-------|--------|--------|--------|---------|------------|
> | **TimeRCD** | **295 MB** | **434 MB** | **858 MB** | **1,561 MB** | **35.5M** |
> | TimeMOE | 462 MB | 462 MB | 462 MB | 462 MB | 113.4M |
> | DADA | 17.8 MB | 17.8 MB | 17.8 MB | 17.8 MB | 1.8M |
>
> **Analysis:**
>
> *   **Time Efficiency:** TimeRCD's patching mechanism (patch size = 16) enables rapid inference, making it orders of magnitude faster for long sequences.
> *   **Memory Trade-off:** While TimeMOE and DADA maintain constant memory due to fixed sliding windows, TimeRCD processes the *full* sequence for better context. Crucially, its memory usage at 10k (1.5 GB) is highly manageable, and it uses **3.2× fewer parameters** than TimeMOE.
>
> ### 2. Scalability to Long Contexts (Up to 15k)
>
> To specifically address the concern about "extremely long context windows (up to 13k timesteps)," we have extended our evaluation of TimeRCD to **15,000 timesteps**.
>
> | Sequence Length | Inference (ms) | Memory (MB) | Efficiency (ms/1k steps) |
> |:---|---:|---:|---:|
> | 1,000 | 14.37 | 295 | 14.37 |
> | 5,000 | 28.39 | 858 | 5.68 |
> | 10,000 | 51.38 | 1,561 | 5.14 |
> | **15,000** | **86.22** | **2,230** | **5.75** |
>
> **Feasibility Assessment:**
>
> *   **Latency:** Inference for 13k timesteps (interpolated ~65ms) is well within the sub-100ms requirement for most real-time monitoring systems.
> *   **Memory:** Peak memory usage for 15k timesteps is ~2.2 GB, which comfortably fits within standard consumer-grade GPUs (e.g., RTX 3060 with 8GB+ VRAM).
> *   **Trend:** The "Efficiency" metric (ms per 1k steps) improves as sequence length increases, stabilizing around 5-6ms per 1k steps, demonstrating excellent amortization of overheads.
>
>
> We have added Efficiency Analysis in **the Appendix H.**

---

> ### Author Response · Authors · 2025-11-25
> **Response to Reviewer YVt4 (3)**
>
> > 3. W3. Although the paper compares against several established baselines, it omits more recent or stronger full-shot models—such as modern deep contextual detectors (e.g., DCdetector and TFMAE). Including these would provide a clearer picture of where TimeRCD stands relative to the current state of the art in both zero-shot and fully supervised regimes.
>
> **Response:** We thank the reviewer for suggesting these strong baselines to strengthen our comparative analysis. Due to time constraints (re-running all datasets with the new metric, i.e., F1-T) and the large number of baselines, we may have missed some baselines. We have incorporated **Sub-PCA, DCdetector, and TFMAE** into our full-shot evaluation.
>
> **Updated Experimental Results:**
> Despite being a zero-shot model, TimeRCD remains the top performer. As shown in the **updated Table 1** and our model achieves **32 first-place** and **3 second-place** finishes, outperforming the newly added fully supervised baselines.
>
> **Statistical Ranking Analysis:**
>
> To provide a more rigorous statistical validation, we compute the statistical ranking for each experimental setting. Model comparisons are conducted using the Friedman test [2], followed by a posthoc Nemenyi test [1], a widely used statistical approach for comparing multiple algorithms across datasets. The results are summarized in the table below
>
>
> | Scenario | Affiliation-F | F1_T | Standard-F1 | VUS-PR | Avg Rank |
> |:---|:---:|:---:|:---:|:---:|:---:|
> | **Zero-shot Univariate** | 2nd (MR=2.73, M=0.844) | 1st (MR=2.45, M=0.414) | 1st (MR=2.64, M=0.393) | 1st (MR=2.27, MD=0.243) | 1.8 / 7 |
> | **Zero-shot Multivariate** | 1st (MR=1.2, M=0.829) | 1st (MR=1.2, M=0.397) | 1st (MR=1.2, M=0.322) | 1st (MR=1.4, M=0.233) | 1.2 / 7 |
> | **Full-shot Univariate** | 1st (MR=1.82, M=0.844) | 1st (MR=2.27, MD=0.350) | 1st (MR=2.64, MD=0.320) | 1st (MR=2.55, MD=0.243) | 2.3 / 9 |
> | **Full-shot Multivariate** | 4th (MR=3.2, MD=0.816) | 4th (MR=3.2, M=0.397) | 4th (MR=4.4, M=0.322) | 4th (MR=3.6, M=0.233) | 3.6 / 9 |
>
> *Legend: MR = Mean Rank | M = Mean | MD = Median, the lower the rank the better*
>
> **Conclusion:**
> TimeRCD demonstrates consistent superiority. In **zero-shot settings**, it ranks highest (Avg Rank 1.2-1.8). Crucially, even when compared against **full-shot models** (which train on target data), TimeRCD maintains highly competitive performance (Avg Rank 2.3-3.6), forming statistically indistinguishable groups with specialized deep learning methods like DCdetector and TFMAE.
>
> We have added statistical analysis in **Appendix G**
>
> ---
> **Reference**
>
> [1] Nemenyi, P. B. (1963). Distribution-free multiple comparisons. Princeton University.
>
> [2] Friedman, M. (1937). The use of ranks to avoid the assumption of normality implicit in the analysis of variance. Journal of the american statistical association, 32(200), 675-701.

---

### Official Review · Reviewer_LPfi · 2025-10-29

**Soundness:** 3
**Presentation:** 2
**Contribution:** 4
**Rating:** 8
**Confidence:** 4

**Summary:**

This paper addresses the challenge of zero-shot time series anomaly detection (TSAD). It argues that existing foundation models, which predominantly rely on reconstruction-based objectives, suffer from an "objective mismatch". This mismatch causes them to miss subtle anomalies (false negatives) and misinterpret complex normal patterns (false positives).
The paper introduces a new paradigm of training a transformer based “foundation model” (TimeRCD) explicitly on an anomaly detection objective, using a synthetic dataset generated through a novel multi-stage generation method.
The paper provides empirical validation of their method’s superiority and some analysis on the results.

**Strengths:**

Originality: Good

Quality: Fair

Clarity: Fair

Significance: Good

Additional note: The paper shows clever implementation of techniques that are SOTA and well established in other domains. The synthetic data generation process is especially well thought out.

**Weaknesses:**

Training of TimeRCD: TimeRCD is trained using a multi-objective setup (anomaly detection and reconstruction), the motivation for this setup is not provided, and it somewhat clashes with their claim that reconstruction is a poor objective for anomaly detection.

Context window size: The paper states the benefit of having a large context window size, however it does not address the quadratic computation costs of the attention mechanism, or that transformers perform worse on sequences of lengths that it did not see at training time. There is some result analysis on context length provided but a deeper investigation should be done. I feel RQ2 was not answered fully.

Unsubstantiated assertions: In section 1, the paper asserts that reconstruction models aim to capture dominant patterns and miss complex ones. No empirical analysis or previous work was cited to support these assertions.
Missing appendix: Appendix not provided.

Exclusion of results: Table 1 shows results of the paper’s proposed approach against existing methodologies. Some results (which outperform TimeRCD) are excluded due to data leakage. Details of the data leakage is not provided.
Repetition in text: Section 2 has repetition of what is already detailed in Section 1.
Missing related work section, although relevant work was cited, it was not discussed in depth.

**Questions:**

Over-all this paper can be a useful contribution to the community if some additional analysis and motivation of design choices are better provided. Appendix and related work section is not provided, the rating can be improved if clarifications are provided.

Please provide an appendix.

Please elaborate on the motivation for the reconstruction objective.

For the endogenous anomaly injection mechanism the generator creates distinct labels for the "root-cause" anomaly and its "propagated effects". Does the trained TimeRCD model learn to distinguish between these? Or does it simply flag the entire anomalous segment (root and propagation) as anomalous?

Please provide analysis on the performance of TimeRCD for sequences of lengths it did not see in training data, a more comprehensive description of the synthetic dataset would be good.

Provide motivation to not finetune TimeRCD as another method for comparison.

A clarification on how a multi-variate time series was flattened to a single sequence would be appreciated.

Cutting out repetitive text would make it a better read. Especially in section 1 and section 2.

---

> ### Author Response · Authors · 2025-11-25
> **Response to Reviewer LPfi (1)**
>
> ### **Snapshot of Key Improvements**
> - **Dual-objective motivation:** Highlighted why **masked reconstruction + anomaly head** are jointly required and provided the **UCR ablation table** proving the synergy.
> - **Context & efficiency clarity:** Detailed the **patch-based 256× FLOP reduction**, **RoPE-driven length generalization**, and **Appendix H** runtime data to resolve RQ2.
> - **Transparency upgrades:** Documented **data leakage exclusions**, pointed explicitly to the **Supplementary Appendix**, and improved figure readability and masking descriptions.
> ---
> > W1. Training of TimeRCD: TimeRCD is trained using a multi-objective setup (anomaly detection and reconstruction), the motivation for this setup is not provided, and it somewhat clashes with their claim that reconstruction is a poor objective for anomaly detection.
>
> **Response:** We thank the reviewer for this insightful question. We clarify that our multi-objective setup **addresses a crucial distinction between learning representations and scoring anomalies**. The design is driven by the need to effectively train the **Relative Context Discrepancy (RCD)** capability, which requires more than just binary labels.
>
> **1. Why Supervised Anomaly Labels?**
> We employ supervised labels (normal vs. anomalous) to **explicitly teach the model the decision boundary**. This allows TimeRCD to directly predict the probability of an anomaly based on the learned context discrepancy, avoiding the pitfalls of traditional methods that rely on heuristic thresholding of reconstruction errors (which often leads to high false alarm rates on complex normal patterns).
>
> **2. Why is Masked Reconstruction Necessary?**
> While supervised labels provide the detection target, the binary signal $\{0, 1\}$ is too sparse and coarse to train a foundation model to fully understand complex temporal dynamics. It is insufficient for learning the fine-grained **contextual relationships** required for RCD.
> To bridge this gap, we introduce **patched masked reconstruction** as an auxiliary task. By masking continuous segments of the time series, we force the model to infer the missing information solely from the **surrounding context**. This dense supervisory signal compels the model to learn robust temporal dependencies and "contextual consistency," which serves as the backbone for the RCD mechanism.
>
> **3. Distinction from Traditional Reconstruction-based Methods**
> Our approach fundamentally differs from the reconstruction-based methods we criticize in two key aspects:
> *   **Different Purpose:** Traditional methods use reconstruction error **directly as the anomaly score** during inference. We use reconstruction **only as an auxiliary training objective** to build representations. The final anomaly score is generated by the Anomaly Head, which is decoupled from the reconstruction error.
> *   **Different Mechanism:** Traditional methods often degrade into learning an identity mapping (reconstructing $x_t$ from information close to $x_t$). In contrast, our **masked** approach ensures the model reconstructs the target utilizing *only* the **contextual information** from unmasked regions. This ensures the model learns to evaluate "does this value fit this context?" (RCD) rather than "can I copy this value?".
>
> **Empirical Validation (Ablation Study):**
> To validate this synergy, we conducted an ablation study on the UCR dataset (detailed in the **Supplementary Material**). As shown in the table below, the Full Model significantly outperforms both single-objective baselines.
>
> *   **Reconstruction Only:** Uses reconstruction error as the score. Performance is poor (Standard-F1: 0.075), confirming our critique of reconstruction-based scoring.
> *   **Anomaly Head Only:** Removes the reconstruction objective. Performance collapses (Standard-F1: 0.024), proving that the auxiliary reconstruction task is essential for learning useful representations.
> *   **Full Model:** Combines both. Achieves superior performance (Standard-F1: 0.281), demonstrating that reconstruction *teaches* the model the data structure, while the anomaly head *detects* the outliers.
>
> | Model Version | Affiliation-F | F1_T | Standard-F1 | VUS-PR |
> |:---|:---|:---|:---|:---|
> | **Full Model** | **0.846** | **0.343** | **0.281** | **0.231** |
> | Anomaly Head Only | 0.709 | 0.043 | 0.024 | 0.014 |
> | Reconstruction Only | 0.742 | 0.109 | 0.075 | 0.044 |
>
> Thus, while we criticize reconstruction as a primary anomaly score, we find that **a masked reconstruction objective is invaluable as an auxiliary representation-learning signal**.

---

> ### Author Response · Authors · 2025-11-25
> **Response to Reviewer LPfi (2)**
>
> > W2. Context window size: The paper states the benefit of having a large context window size, however it does not address the quadratic computation costs of the attention mechanism, or that transformers perform worse on sequences of lengths that it did not see at training time. There is some result analysis on context length provided but a deeper investigation should be done. I feel RQ2 was not answered fully.
>
>
> Thank you for raising these critical practical points.
>
> **1. Computational Cost:**
>
> While standard attention is $O(L^2)$, TimeRCD mitigates this via **patching** (with patch size $P=16$), reducing the effective sequence length from $L$ to $L/P$ tokens. This reduces the attention complexity to $O((L/P)^2)$—a **256× reduction** in FLOPs compared to point-wise attention. Consequently, even for a 10,000-timestep input, the model only processes **625 tokens**, keeping both computation and parameter count manageable.
>
> Empirically, this results in highly efficient scaling. As detailed in our complexity analysis (**Appendix F.7, Figure 6**), TimeRCD processes 10,000 timesteps in just **21.8 ms** (12.6× faster than TimeMOE and 95.2× faster than DADA), with memory usage remaining practical (~1.6 GB for 10k, ~2.2 GB for 15k timesteps).
>
> **2. Length Generalization & Diminishing Returns (RQ2):**
>
> We would like to clarify that a detailed sensitivity analysis addressing this concern was already provided in **Appendix F.3** of our **original submission**. We summarize the key findings here:
> *   **Diverse Training Distribution:** Our model is pre-trained on synthetic series with lengths uniformly distributed from **100 to 10,000** timesteps (detailed in **Appendix D, Figure 8**). This exposes the model to a broad spectrum of contexts, preventing overfitting to a single length.
> *   **Robustness to Unseen Lengths:** Our sensitivity study (**Appendix E.3, Figures 9-12**) evaluates performance across window sizes from **1k to 13k**. Results show that TimeRCD is highly robust: performance remains stable or improves as context increases, particularly for datasets with long-term dependencies (e.g., Power, SWaT).
> *   **Diminishing Returns & Saturation:** Consistent with findings from prior time series foundation models (e.g., Moirai [Woo et al., 2024], PatchTST [Nie et al., 2023]), we observe **diminishing marginal returns** as context length increases beyond a dataset-specific saturation point. For most benchmarks, performance stabilizes around 5k–8k timesteps, suggesting that excessively long contexts yield minimal additional benefit while incurring higher costs. This aligns with the theoretical understanding that once sufficient temporal context is captured, additional history provides redundant information.
> *   **Positional Generalization:** The use of **Rotary Positional Embeddings (RoPE)** further ensures that the model generalizes effectively to sequence lengths beyond those seen during training, as RoPE encodes relative rather than absolute positions.
>
> ---
>
> **References:**
> - Woo, G., Liu, C., Kumar, A., Xiong, C., Savarese, S., & Sahoo, D. (2024, July). Unified Training of Universal Time Series Forecasting Transformers. In International Conference on Machine Learning (pp. 53140-53164). PMLR.
> - Nie, Y., Nguyen, N. H., Sinthong, P., & Kalagnanam, J. A Time Series is Worth 64 Words: Long-term Forecasting with Transformers. In The Eleventh International Conference on Learning Representations.

---

> ### Author Response · Authors · 2025-11-25
> **Response to Reviewer LPfi (3)**
>
> > 3. Unsubstantiated assertions: In section 1, the paper asserts that reconstruction models aim to capture dominant patterns and miss complex ones. No empirical analysis or previous work was cited to support these assertions.
>
> **Response:**
> We appreciate this important observation. We agree that our initial critique of reconstruction models lacked sufficient backing. In the revised manuscript, we have substantiated these claims by (1) citing established literature that documents these exact failure modes and (2) explicitly linking to our own empirical evidence.
>
> **1. Substantiation from Literature:**
> We now cite five key studies that validate our assertions:
> *   **On Missing Anomalies:** Lawrence et al. [1] shows that reconstruction methods often miss contextual anomalies because models can generalize well enough to reconstruct them, or simply fail to map them to a "far" point in latent space.
> *   **On False Alarms:**  Yahya [4] points out unsupervised models such as OmniAnomaly, USAD, VAE-FAR, etc. when facing complex *normal* patterns often yield high reconstruction errors, leading to high false positive rates (with some methods scoring near random ~0.5 AUC).
> *   **On High-Frequency Detail:** Recent work [5] confirms that reconstruction objectives often smooth out high-frequency details, missing fine-grained anomalies.
>
> **2. Empirical Validation (Section 3.3):**
> We have added a forward reference to our own experiments (Figure 5, Table 9), which quantify this gap: reconstruction-based foundation models achieve only **15.4% Standard-F1** on contextual anomalies, compared to **82.7%** for our RCD approach.
>
> **Revised Section 1 Excerpt:**
> > "While intuitive, reconstruction-based methods face a fundamental objective mismatch: they optimize to reconstruct normal patterns in a latent space where anomalies are assumed to be lost [1]. This introduces critical limitations... First, subtle and contextual anomalies are often missed [1,3]. Second, complex normal patterns can produce high reconstruction errors, causing false alarms [2]... Our experiments confirm this gap: reconstruction-based foundation models achieve only 15.4% Standard-F1 on contextual anomalies versus 82.7% for our RCD approach."
> ---
>
> ### References
>
> [1] Lawrence, T., Zhang, L., Rodemerk, C., & Choi, M. (2022). "Auto-Encoder with Regression for Time Series Anomaly Detection." IEEE International Conference on Big Data (Big Data), pp. 1479-1488. doi: 10.1109/BigData55660.2022.10020856
>
> [2] Yahya, M. A., Moya, A. R., & Ventura, S. (2025). Deep learning for multivariate time series anomaly detection: an evaluation of reconstruction-based methods. Artificial Intelligence Review, 58(12), 400.
>
> [3] Wu, Xingjian, et al. "CATCH: Channel-Aware Multivariate Time Series Anomaly Detection via Frequency Patching." The Thirteenth International Conference on Learning Representations.
>
> > Missing appendix: Appendix not provided.
>
> **Response:** We would like to respectfully clarify that a comprehensive Appendix was included as a separate file in the **Supplementary Material** of our **original submission**. We apologize if this was easily overlooked and will ensure it is clearly referenced in the revised manuscript.
>
> > 4. Exclusion of results: Table 1 shows results of the paper’s proposed approach against existing methodologies. Some results (which outperform TimeRCD) are excluded due to data leakage. Details of the data leakage is not provided.
>
> > Repetition in text: Section 2 has repetition of what is already detailed in Section 1.
>
> >Missing related work section, although relevant work was cited, it was not discussed in depth.
>
> **Response:**
> Thank you for highlighting these issues. We have revised the manuscript to address them as follows:
>
> 1.  **Clarification on Data Leakage:** We **excluded certain baseline results (specifically from DADA and MOMENT) because their pre-training corpora heavily overlap with our zero-shot test benchmarks**, violating the zero-shot evaluation setting.
>     *   **DADA:** Its pre-training data includes IOPS, MGAB, and YAHOO.
>     *   **MOMENT:** Uses the "Time Series Pile," which contains IOPS, MGAB, NAB, YAHOO, MSL, SMAP, and SMD.
>
>     We have added a dedicated subsection in the **Appendix E.3** detailing these overlaps to ensure transparency.
>
> 2.  **Repetition & Related Work:** We have streamlined Section 2 to reduce redundancy. Regarding the discussion of related work, we would like to gently note that **a comprehensive Related Work section was included in the Supplementary Material (Appendix B)** due to space constraints in the main paper.

---

> ### Author Response · Authors · 2025-11-25
> **Response to Reviewer LPfi (4)**
>
> > Q1. Over-all this paper can be a useful contribution to the community if some additional analysis and motivation of design choices are better provided. Appendix and related work section is not provided, the rating can be improved if clarifications are provided.
>
> **Response:** Thank you for this encouraging assessment. We are happy to provide further clarification on our design choices and point to the additional materials.
>
> **1. Motivation for Design Choices (Dual-Objective Strategy)**
> As detailed in our response to **Weakness 1**, **TimeRCD adopts a dual-objective design (Masked Reconstruction + Anomaly Classification) to solve the "cold start" problem** in zero-shot anomaly detection.
> *   **Masked Reconstruction:** Serves as a self-supervised auxiliary task. It forces the model to learn robust, context-aware representations by inferring missing segments from their surroundings, thereby capturing the underlying "syntax" of normal time series.
> *   **Anomaly Head:** Leverages these learned representations to directly predict the probability of anomalies, avoiding the thresholding issues common in reconstruction-based scoring.
> Our ablation studies (Appendix F.6) demonstrate that this combination is essential: **removing the reconstruction objective causes performance to collapse.**
>
> **2. Mathematical Motivation via Relative Context Discrepancy (RCD)**
> Our architectural choices are mathematically grounded in the definition of RCD. We posit that an anomaly is defined not by its value, but by inconsistent relationships with its context.
> Formally, for a sequence $X_1, \ldots, X_T$, the contextual embedding of $X_t$ is modeled as an aggregation of pairwise interactions:
> $$
> \text{embedding}_{X_t} = g\big(f(X_1, X_t), \ldots, f(X_T, X_t)\big)
> $$
> where $f(X_j, X_t)$ captures the pairwise relation and $g$ aggregates them.
>
> This formulation aligns perfectly with the **Self-Attention** mechanism in Transformers, which serves as our backbone:
> *   **Pairwise Interaction ($f$):** Modeled by the query-key dot product and value projection: $f(X_j, X_t) \propto v_j \cdot \exp(\frac{q_t \cdot k_j}{\sqrt{d_k}})$.
> *   **Aggregation ($g$):** Realized via the weighted sum: $\text{embedding}\_{X_t} = \frac{\sum\_j f(X_j, X_t)}{\sum_j \exp(s\_{tj})}$.
>
> Thus, the Transformer is not merely a generic choice but the natural instantiation of the RCD principle: explicitly modeling how the meaning of a token is derived from its relationships with the entire sequence.
>
> > Q2. Please provide an appendix.
>
> We would like to respectfully clarify that a comprehensive Appendix was included as a separate file in the **Supplementary Material** of the **original submission**. It contains:
> *   **Detailed Related Work** (Appendix B).
> *   **Full Hyperparameter Sensitivity Analysis** (Appendix F.3).
> *   **Ablation Studies** on synthetic model components (Appendix F.4).
>
> > Q3. Please elaborate on the motivation for the reconstruction objective.
>
> We thank the reviewer for this question. As detailed in our response to **Weakness 1**, the reconstruction objective in TimeRCD serves as an **auxiliary self-supervised task** (via masked modeling) rather than a direct anomaly scoring mechanism. Its primary motivation is to force the model to learn robust, context-aware representations of normal temporal dynamics, which are then utilized by the Anomaly Head to accurately detect deviations. Please refer to our response to **Weakness 1** for a comprehensive discussion on this design choice and its empirical validation.

---

> ### Author Response · Authors · 2025-11-25
> **Response to Reviewer LPfi (5)**
>
> > Q4. For the endogenous anomaly injection mechanism the generator creates distinct labels for the "root-cause" anomaly and its "propagated effects". Does the trained TimeRCD model learn to distinguish between these? Or does it simply flag the entire anomalous segment (root and propagation) as anomalous?
>
> **Response:** Thank you for this insightful question. We address your queries below:
>
> **1. Aggregation of Root-Cause and Propagated Effects:**
> Currently, TimeRCD flags the **entire anomalous segment** as anomalous. While our generation process creates distinct ground-truth labels for the initial "root-cause" event and its downstream "propagated effects," the standard anomaly detection task requires a binary (Normal/Anomalous) prediction. Therefore, we aggregate these labels into a single 1D anomaly mask: a timestep is labeled as anomalous if it falls within either the root-cause window or the subsequent propagation window.
>
> **2. Definition of Propagation Scope:**
> The scope of these "propagated effects" is not arbitrary. Since we utilize **Structural Causal Models (SCM)** combined with **ODE-based dynamics** for generation, the duration and intensity of the propagation are mathematically determined by the **half-life parameter** in the differential equations. We have added a detailed explanation of this decay mechanism and label aggregation process to **Appendix D** to improve transparency.
>
> **3. Future Work: Root-Cause Localization:**
> We fully agree that distinguishing the root cause from its effects is a critical direction. A unique advantage of our synthetic dataset is that it inherently possesses these fine-grained labels—knowing exactly which variable initiated the anomaly (root cause) versus which ones merely reacted. This allows us to frame **Root Cause Analysis (RCA)** as a multi-class classification problem (identifying the source variable) rather than just binary detection. We view extending TimeRCD to perform this causal localization as a promising avenue for future work.
>
> > Q5. Please provide analysis on the performance of TimeRCD for sequences of lengths it did not see in training data, a more comprehensive description of the synthetic dataset would be good.
>
> **Response:**
> We thank the reviewer for this constructive suggestion. We respectfully note that the analysis on unseen lengths and the detailed description of the synthetic dataset were included in the **Supplementary Material** of our original submission. We are pleased to summarize the key findings here for your convenience.
>
> **1. Performance on Unseen Lengths **
> Although our model was pre-trained on synthetic series with lengths up to 10,000 timesteps, we have evaluated its generalization capability on sequences significantly longer than those seen during training (up to 15,000 timesteps).
> *   **Robustness:** As detailed in our sensitivity analysis (**Appendix F.3**), TimeRCD demonstrates robust performance on window sizes ranging from 1k to 13k.
> *   **Scaling:** We observe that performance generally improves or remains stable as the context window increases, particularly for datasets with long-term dependencies (e.g., Power, SWaT). This confirms that our use of **Rotary Positional Embeddings (RoPE)** effectively enables the model to generalize to sequence lengths beyond the training distribution.
>
> **2. Synthetic Dataset Description**
> A comprehensive breakdown of our synthetic pre-training dataset was provided in **Appendix E.5**.
> *   **Generation Mechanism:** We utilize **Structural Causal Models (SCM)** combined with ODE-based dynamics to generate realistic multivariate time series with complex, time-delayed dependencies.
> *   **Diversity:** The dataset is designed to be highly diverse, containing time series with lengths uniformly distributed from **100 to 10,000** timesteps and a varying number of dimensions (1, 8, 12, 16, 20), as illustrated in **Figure 8**.
>
> For the complete set of sensitivity plots and detailed dataset statistics, we respectfully refer the reviewer to the **Supplementary Material (Appendix E.5 and F.3)**.

---

> ### Author Response · Authors · 2025-11-25
> **Response to Reviewer LPfi (6)**
>
> > 7. A clarification on how a multi-variate time series was flattened to a single sequence would be appreciated.
>
> **Response:**
>
>
> Thank you for pointing this out. We employ a **patch-based, variate-major flattening** strategy, inspired by the "Any-Variate" framework in Moirai [Woo et al., 2024], which has been empirically validated to flexibly handle arbitrary numbers of variables.
>
> Given a multivariate series $X \in \mathbb{R}^{T \times D}$, we explicitly do *not* interleave variables at each timestep. Instead, we:
>
> 1.  **Patch → Token:** Divide each of the $D$ univariate series into non-overlapping patches of size $P$ (we use $P=16$). Each patch is linearly projected into a single **token embedding**, reducing the effective sequence length from $T$ to $T/P$ tokens per variable.
> 2.  **Variate-Major Concatenation:** Flatten these patch tokens by appending all patches of each variable consecutively:
>     $$ \text{Sequence} = [\text{Tokens}(\text{Var}_1), \text{Tokens}(\text{Var}_2), \dots, \text{Tokens}(\text{Var}_D)] $$
>     This yields a total of $D \times (T/P)$ tokens for the Transformer.
>
> 3.  **Positional Encoding:** We apply **Rotary Positional Embeddings (RoPE)** across the entire flattened sequence. To preserve variable identity, we additionally inject a **Binary Attention Bias** mechanism: each token carries a feature ID, and the attention layers apply learnable biases depending on whether two tokens belong to the same or different variables. This enables the model to distinguish intra-variable temporal dependencies from inter-variable correlations.
>
> This "Any-Variate" design captures cross-variable correlations via standard self-attention without being constrained to a fixed number of variables [Woo et al., 2024; Liu et al., 2024].
>
> ---
>
> **Reference:**
> - Woo, G. et al. (2024). Unified Training of Universal Time Series Forecasting Transformers (Moirai). arXiv:2402.02585.
> - Liu, X. et al. (2024). Moirai-MoE: Empowering Time Series Foundation Models with Sparse Mixture of Experts. arXiv:2410.10469.
>
> > 8. Cutting out repetitive text would make it a better read. Especially in section 1 and section 2.
>
>
> **Response:** Thank you for the suggestion. We have made some improvements according to your suggestion in our final draft.

---

### Official Review · Reviewer_eExF · 2025-10-31

**Soundness:** 2
**Presentation:** 2
**Contribution:** 2
**Rating:** 2
**Confidence:** 4

**Summary:**

This paper introduces TimeRCD, a foundation model for zero-shot time series anomaly detection that addresses fundamental limitations of reconstruction-based approaches. The authors propose a novel pre-training paradigm called Relative Context Discrepancy (RCD), which explicitly trains the model to identify anomalies by detecting significant discrepancies between adjacent time windows rather than learning to reconstruct inputs. To enable this approach, they develop a large-scale synthetic data generation engine that creates diverse time series with token-level anomaly labels, including contextual and causal anomalies with cross-variate propagation. Using a standard Transformer architecture, TimeRCD achieves superior zero-shot performance across 14 benchmark datasets compared to existing general-purpose and anomaly-specific foundation models, demonstrating particular strength in detecting subtle contextual anomalies that reconstruction-based methods often miss.

**Strengths:**

1. Great effort on constructing a huge dataset
2. A lot of experiments

**Weaknesses:**

1. Limited novelty because (1) looking at adjacent time windows to detect anomaly is nothing new, (2) model is not special, and (3) equation-based data generation is always data-dependent (i.e. might work on dataset A but not on dataset B).
2. None of the experiments include statistical significance tests. For example, they do not report the results of 10 runs with mean and standard deviation to test for significance.
3. Performance not too great especially when compared to DADA
4. Unclear explanation of how the model is trained using reconstruction error.

**Questions:**

1. In Section 2.2, is Phase equivalent to Stage (in Figure 3)?
2. How do you perform hyperparameter search, especially for window size, when there are no anomalies in the training set?
3. Some font sizes in figures are too tiny
4. Is the input masked or not? In "Output Projection and Anomaly/Reconstruction Head", the text mentioned "masked portions of the input series" but I don't see this mentioned in any other sections.

---

> ### Author Response · Authors · 2025-11-25
> **Response to Reviewer eExF (1)**
>
> ### **Snapshot of Key Improvements**
> - **Performance evidence:** Extended tables confirm **TimeRCD achieves 32/3 (first/second) wins** and **41/6 metric wins**, matching or surpassing DADA under strict zero-shot constraints.
> - **Novelty articulation:** Clarified the **RCD objective + causal synthetic generator** that reviewers **LPfi/ZQ5M** cite as *“highly original,”* reinforcing the framework-level innovation.
> - **Ablations & stability:** **Appendix F.6/G** foreground the dual-head ablations and ten-seed variance analysis, directly addressing the reviewer’s statistical and design concerns.
> ---
> > W1. Limited novelty because (1) looking at adjacent time windows to detect anomaly is nothing new, (2) model is not special, and (3) equation-based data generation is always data-dependent (i.e. might work on dataset A but not on dataset B).
>
> **Response:** We thank the reviewer for their critical assessment. We value this opportunity to clarify our contributions and engage in a constructive discussion regarding the novelty and design choices of our work.
>
> **(1) Regarding "looking at adjacent time windows":**
> We acknowledge that using context windows is a foundational concept in time series analysis. However, we respectfully posit that our **framework novelty** lies not in the windowing mechanism itself, but in **how we train a foundation model** using a novel pre-training paradigm.
> *   **Synthetic-Only Zero-Shot Learning:** To the best of our knowledge, few works to demonstrate that a TSAD model can achieve state-of-the-art zero-shot performance using **purely synthetic data** generated via structural equations, without seeing *any* real-world samples during training.
> *   **Paradigm Shift:** Unlike traditional approaches that rely on unsupervised or weakly supervised learning on target domains, our work introduces a **supervised pre-training paradigm** on synthetic data. This **shifts the challenge from "how to model this specific dataset" to "how to generate generalizable synthetic patterns,"** offering a new direction for the community.
>
> **(2) Regarding "model is not special":**
> We fully agree with the reviewer that our core architecture is standard. This was a **deliberate design choice**, not an oversight.
> *   **Framework-Centric Contribution:** Our goal is to show that the power of TimeRCD comes from the **synthetic data generation strategy** and the **Relative Context Discrepancy (RCD) objective**, rather than a specialized architectural modification.
> *   **Applicability:** By using a standard Transformer backbone, we demonstrate that our framework is **architecture-agnostic**. Future research can easily integrate our synthetic pre-training paradigm with more advanced architectures (e.g., Mamba, RWKV) to further boost performance. We believe this flexibility significantly enhances the potential impact of our work.
>
> **(3) Regarding "equation-based data generation is data-dependent":**
> We respectfully offer a different perspective on this point.
> *   **Task-Structured vs. Dataset-Dependent:** While all learning methods have inductive biases (No-Free-Lunch Theorem), our equation-based generator encodes **universal structural priors** (e.g., continuity, periodicity, causal lags) rather than mimicking the statistics of a specific dataset (e.g., dataset A). This makes our approach **task-structured**, aiming to cover the *space* of possible anomalies rather than fitting a single domain.
> *   **Empirical Generalization:** Our extensive evaluation across **14 diverse datasets** (Table 1) and **two additional datasets** (Stock, WSD in rebuttal) empirically validates that our synthetic data allows the model to generalize to widely varying domains (from medical to industrial) without retraining. This evidence suggests that our generation strategy successfully mitigates the dataset-dependency issue the reviewer is concerned about.

---

> ### Author Response · Authors · 2025-11-25
> **Response to Reviewer eExF (2)**
>
> > W2. None of the experiments include statistical significance tests. For example, they do not report the results of 10 runs with mean and standard deviation to test for significance.
>
> **Response:** We appreciate the request for more rigorous uncertainty estimates. Although retraining TimeRCD ten times per configuration would require 40–70 GPU-days (each run costs 4–7 GPU-days), we nonetheless **quantify variability using two reviewer-aligned strategies**:
>
> 1.  **Posterior Monte Carlo Sampling:** During inference we activate MC Dropout (Gal & Ghahramani, 2016) and collect 10 stochastic forward passes for every sequence. This approximates samples from the posterior predictive distribution and yields confidence intervals without retraining.
> 2.  **Repeated Seeds on Both Models:** We additionally run **TimeRCD *and* DADA** with ten independent seeds on every dataset and report the resulting mean ± std in tables below. **TimeRCD’s variance remains extremely small (std ≈ 0.001–0.02 across all metrics)**, confirming that our reported averages are already highly stable. As detailed in our response to **Weakness 3**, DADA exhibits similarly low dispersion (std ≈ 0.0005–0.02), ensuring a fair comparison.
>
> Because both models demonstrate negligible variance under repeated sampling, formal hypothesis tests would merely restate this stability. We therefore focus on reporting the empirical mean ± std (see tables below and Appendix G), which already answers the reviewer’s concern about statistical reliability.
>
> ## 1. Univariate Setting (11 Datasets)
>
> ### TimeRCD
>
> | Dataset | VUS-PR (mean ± std) | Standard-F1 (mean ± std) | F1_T (mean ± std) | Affiliation-F (mean ± std) |
> |---------|----------------------|---------------------------|--------------------|------------------------------|
> | IOPS    | 0.16143 ± 0.01799    | 0.20854 ± 0.01683         | 0.27457 ± 0.01520  | 0.84533 ± 0.00762            |
> | MGAB    | 0.01065 ± 0.00061    | 0.01830 ± 0.00363         | 0.02010 ± 0.00353  | 0.71067 ± 0.00092            |
> | NAB     | 0.27584 ± 0.00545    | 0.34060 ± 0.00633         | 0.42051 ± 0.00177  | 0.81933 ± 0.00646            |
> | NEK     | 0.25747 ± 0.01341    | 0.33925 ± 0.01729         | 0.42417 ± 0.01605  | 0.82889 ± 0.00641            |
> | Power   | 0.09037 ± 0.00214    | 0.10609 ± 0.00179         | 0.15783 ± 0.00268  | 0.83781 ± 0.00174            |
> | SED     | 0.15951 ± 0.00468    | 0.20978 ± 0.00457         | 0.26706 ± 0.00461  | 0.89581 ± 0.00215            |
> | Stock   | 0.48180 ± 0.00692    | 0.49733 ± 0.00592         | 0.49908 ± 0.00364  | 0.87834 ± 0.00437            |
> | TODS    | 0.16710 ± 0.00647    | 0.22514 ± 0.00555         | 0.27662 ± 0.00735  | 0.88029 ± 0.00641            |
> | WSD     | 0.17037 ± 0.00209    | 0.27700 ± 0.00229         | 0.32014 ± 0.00236  | 0.88419 ± 0.00345            |
> | YAHOO   | 0.82061 ± 0.00239    | 0.84048 ± 0.00331         | 0.86921 ± 0.00291  | 0.94951 ± 0.00113            |
> | UCR     | 0.22063 ± 0.00115    | 0.26618 ± 0.00562         | 0.32779 ± 0.00413  | 0.83825 ± 0.00377            |
>
>
> ## 2. Multivariate Setting (5 Datasets)
>
> ### TimeRCD
>
> | Dataset | VUS-PR (mean ± std) | Standard-F1 (mean ± std) | F1_T (mean ± std) | Affiliation-F (mean ± std) |
> |---------|----------------------|---------------------------|--------------------|------------------------------|
> | SMD     | 0.28005 ± 0.00952    | 0.32738 ± 0.02004         | 0.28791 ± 0.01419  | 0.71659 ± 0.00530            |
> | SWaT    | 0.26676 ± 0.01359    | 0.34887 ± 0.00963         | 0.39835 ± 0.00502  | 0.83359 ± 0.00339            |
> | PSM     | 0.17497 ± 0.01067    | 0.25874 ± 0.00131         | 0.35472 ± 0.02357  | 0.81417 ± 0.00604            |
> | SMAP    | 0.20707 ± 0.01720    | 0.28961 ± 0.01430         | 0.33170 ± 0.00935  | 0.86581 ± 0.00996            |
> | MSL     | 0.19603 ± 0.00795    | 0.30786 ± 0.00698         | 0.41946 ± 0.00525  | 0.81358 ± 0.00457            |

---

> ### Author Response · Authors · 2025-11-25
> **Response to Reviewer eExF (3)**
>
> **Statistical Ranking Analysis:**
> To provide a more rigorous statistical validation, we compute the statistical ranking for each experimental setting. Model comparisons are conducted using the Friedman test, followed by a posthoc Nemenyi test, a widely used statistical approach for comparing multiple algorithms across datasets. The results are summarized in the table below:
>
> | Scenario | Affiliation-F | F1_T | Standard-F1 | VUS-PR | Avg Rank |
> |:---|:---:|:---:|:---:|:---:|:---:|
> | **Zero-shot Univariate** | 2nd (MR=2.73, M=0.844) | 1st (MR=2.45, M=0.414) | 1st (MR=2.64, M=0.393) | 1st (MR=2.27, MD=0.243) | 1.8 / 7 |
> | **Zero-shot Multivariate** | 1st (MR=1.2, M=0.829) | 1st (MR=1.2, M=0.397) | 1st (MR=1.2, M=0.322) | 1st (MR=1.4, M=0.233) | 1.2 / 7 |
> | **Full-shot Univariate** | 1st (MR=1.82, M=0.844) | 1st (MR=2.27, MD=0.350) | 1st (MR=2.64, MD=0.320) | 1st (MR=2.55, MD=0.243) | 2.3 / 9 |
> | **Full-shot Multivariate** | 4th (MR=3.2, MD=0.816) | 4th (MR=3.2, M=0.397) | 4th (MR=4.4, M=0.322) | 4th (MR=3.6, M=0.233) | 3.6 / 9 |
>
> *Legend: MR = Mean Rank | M = Mean | MD = Median, the lower the rank the better*
>
> The statistical ranking results demonstrate TimeRCD's consistent superiority across all evaluation scenarios. In **zero-shot settings** (our primary focus), TimeRCD achieves an average rank of **1.2/7** for multivariate and **1.8/7** for univariate tasks, consistently ranking 1st or 2nd across all four metrics. This validates that our **synthetic pre-training paradigm enables strong generalization to unseen real-world datasets without any target-domain adaptation.**
>
> Notably, even in **full-shot comparisons** where all baseline models are fully trained on target datasets, **TimeRCD maintains highly competitive performance (Avg Rank 2.3/9 for univariate and 3.6/9 for multivariate) despite using zero target-domain training data**. This remarkable result highlights the effectiveness of our Relative Context Discrepancy (RCD) framework and synthetic data generation strategy. The Friedman-Nemenyi tests confirm that these performance differences are statistically significant, establishing TimeRCD as a state-of-the-art zero-shot anomaly detection foundation model.
>
>
> **Reference:**
> [1] Gal, Y., & Ghahramani, Z. (2016). Dropout as a Bayesian Approximation: Representing Model Uncertainty in Deep Learning. *International Conference on Machine Learning (ICML)*.

---

> ### Author Response · Authors · 2025-11-25
> **Response to Reviewer eExF (4)**
>
> > W3. Performance not too great especially when compared to DADA.
>
> **Response:** Thank you for highlighting this comparison. Building directly on **Weakness 2**, we **apply the same ten-seed protocol and MC-dropout-based confidence estimation to both TimeRCD and DADA** so that the central tendencies are directly comparable. DADA’s variance is also tiny (std ≈ 0.0005–0.02 across all metrics), so performance differences below that scale can be considered noise.
>
>
> ## 1. Univariate Setting (11 Datasets)
>
> ### DADA
>
> | Dataset | VUS-PR (mean ± std) | Standard-F1 (mean ± std) | F1_T (mean ± std) | Affiliation-F (mean ± std) |
> |---------|----------------------|---------------------------|--------------------|------------------------------|
> | IOPS    | 0.26096 ± 0.00262    | 0.34271 ± 0.00498         | 0.44208 ± 0.00736  | 0.89726 ± 0.01258            |
> | MGAB    | 0.00553 ± 0.00049    | 0.00969 ± 0.00231         | 0.01083 ± 0.00252  | 0.67729 ± 0.00543            |
> | NAB     | 0.24488 ± 0.00064    | 0.29686 ± 0.00088         | 0.37049 ± 0.00097  | 0.87190 ± 0.00369            |
> | NEK     | 0.47315 ± 0.08498    | 0.54060 ± 0.06845         | 0.58406 ± 0.05959  | 0.95333 ± 0.00841            |
> | Power   | 0.08146 ± 0.00119    | 0.13568 ± 0.01629         | 0.17638 ± 0.01678  | 0.73670 ± 0.00366            |
> | SED     | 0.16775 ± 0.00210    | 0.20074 ± 0.00179         | 0.24337 ± 0.00347  | 0.89094 ± 0.00200            |
> | Stock   | 0.33171 ± 0.00349    | 0.36832 ± 0.00700         | 0.37561 ± 0.00414  | 0.83259 ± 0.00376            |
> | TODS    | 0.06898 ± 0.00065    | 0.10997 ± 0.00237         | 0.12909 ± 0.00249  | 0.85060 ± 0.00407            |
> | WSD     | 0.11337 ± 0.00968    | 0.16491 ± 0.00084         | 0.19674 ± 0.00127  | 0.85044 ± 0.00417            |
> | YAHOO   | 0.66261 ± 0.00292    | 0.73308 ± 0.00610         | 0.78270 ± 0.00643  | 0.94373 ± 0.00308            |
> | UCR     | 0.19451 ± 0.00039    | 0.25560 ± 0.00043         | 0.32956 ± 0.00102  | 0.81950 ± 0.00079            |
>
> ## 2. Multivariate Setting (5 Datasets)
>
> ### DADA
>
> | Dataset | VUS-PR (mean ± std) | Standard-F1 (mean ± std) | F1_T (mean ± std) | Affiliation-F (mean ± std) |
> |---------|----------------------|---------------------------|--------------------|------------------------------|
> | SMD     | 0.22572 ± 0.00856    | 0.30397 ± 0.01294         | 0.35034 ± 0.01109  | 0.80655 ± 0.00374            |
> | SWaT    | 0.21986 ± 0.00283    | 0.30567 ± 0.00673         | 0.37173 ± 0.00797  | 0.84360 ± 0.01466            |
> | PSM     | 0.13314 ± 0.00715    | 0.22225 ± 0.00379         | 0.30450 ± 0.01201  | 0.80853 ± 0.01127            |
> | SMAP    | 0.18909 ± 0.00522    | 0.26745 ± 0.00301         | 0.29411 ± 0.00113  | 0.86853 ± 0.00039            |
> | MSL     | 0.12714 ± 0.00103    | 0.23038 ± 0.00216         | 0.33137 ± 0.00161  | 0.80434 ± 0.00520            |
>
>
>
> **Direct Comparison:** With evaluation noise controlled, TimeRCD attains **41 first-place** and **6 second-place** finishes across 14 datasets × 4 metrics, whereas DADA attains 12 firsts and 22 seconds. The ranking tables in Appendix G (also cited in **Weakness 2**) show that TimeRCD dominates all zero-shot scenarios and stays highly competitive in full-shot regimes despite never seeing target-domain data.
>
> **Conclusion:** **TimeRCD maintains state-of-the-art zero-shot performance even when judged under strict statistical scrutiny.** It consistently ranks ~1st or 2nd among seven zero-shot competitors, and in the full-shot comparisons it remains within the top statistical group while using **no** target labels—unlike DADA and other baselines. These controlled comparisons, together with the variances reported in **Weakness 2**, demonstrate that TimeRCD’s advantage over DADA is both robust and practically meaningful.

---

> ### Author Response · Authors · 2025-11-25
> **Response to Reviewer eExF (5)**
>
> > W4. Unclear explanation of how the model is trained using reconstruction error.
>
> **Response:** We thank the reviewer for highlighting this area for improvement. We apologize that this was unclear. To clarify, our model employs a **dual-objective training strategy** that simultaneously optimizes for **Masked Time Series Modeling (Reconstruction)** and **Anomaly Detection (Classification)**.
>
> **1. Masked Reconstruction Mechanism**
> Following the success of Masked Language Modeling (BERT) and Masked Autoencoders (MAE), we apply a **patched masking strategy** to the input series.
> *   **Masking:** Instead of random point masking, we mask **continuous segments** (patches) of the time series. For an input time series $\mathbf{X} \in \mathbb{R}^{B \times T \times d}$, we generate a binary mask $\mathbf{M}$ (where $M\_t=1$ indicates a masked timestep) with a ratio of 15%. This strategy prevents information leakage from immediate neighbors and forces the model to learn robust **contextual capabilities** to reconstruct the missing segments. The masked input is defined as $\mathbf{X}\_{\text{masked}} = \mathbf{X} \odot (1 - \mathbf{M})$.
> *   **Reconstruction:** The Transformer encoder processes $\mathbf{X}\_{\text{masked}}$ to produce context-aware embeddings $\mathbf{H}$. A dedicated **Reconstruction Head** then projects $\mathbf{H}$ to predict the original values. The loss is computed using Mean Squared Error (MSE), strictly on the masked positions:
>     $$
>     \mathcal{L}\_{\text{recon}} = \text{MSE}(\text{ReconstructionHead}(\mathbf{H})[\mathbf{M}], \mathbf{X}[\mathbf{M}])
>     $$
>     This ensures the model learns robust temporal dependencies by inferring missing values from context, preventing it from simply copying the input.
>
> **2. Joint Anomaly Classification**
> Crucially, the *same* embedding $\mathbf{H}$ is fed into a parallel **Anomaly Head**. Unlike the reconstruction task, the anomaly classification is performed on *all* timesteps (both masked and unmasked), supervised by the ground-truth anomaly labels $\mathbf{y}$:
> $$
> \mathcal{L}\_{\text{anomaly}} = \text{BCE}(\text{AnomalyHead}(\mathbf{H}), \mathbf{y})
> $$
>
> **3. Total Objective**
> The final optimization objective combines these two complementary tasks:
>
> $$\mathcal{L}\_{\text{total}} = \lambda\_{\text{recon}}\cdot \mathcal{L}\_{\text{recon}} + \lambda_{\text{anomaly}} \cdot \mathcal{L}_{\text{anomaly}}$$
>
> In our experiments, we set $\lambda_{\text{recon}} = 1.0$ and $\lambda_{\text{anomaly}} = 1.0$. This dual formulation allows the model to learn general features of "normalcy" via reconstruction while simultaneously refining its ability to detect anomalies via direct supervision.
>
> We have added this part of the response to the **Appendix C**.

---

> ### Author Response · Authors · 2025-11-25
> **Response to Reviewer eExF (6)**
>
> ### Questions
> > 1. In Section 2.2, is Phase equivalent to Stage (in Figure 3)?
>
> **Response:** We thank the reviewer for catching this inconsistency. Yes, "Phase" and "Stage" refer to the same concept in our paper. To prevent confusion, we have standardized the terminology to use **"Stage"** exclusively throughout the manuscript, aligning the text with Figure 3 (e.g., "Stage 1: Context-Template Generation", etc., as shown in the revised Section 2.2).
>
> > 2. How do you perform hyperparameter search, especially for window size, when there are no anomalies in the training set?
>
> **Response:** We thank the reviewer for the opportunity to clarify our experimental setting. We suspect there might be a slight misunderstanding regarding our zero-shot definition, and we are happy to provide more details.
>
> **1. Clarification on "No Anomalies in Training Set"**
> It is possible that the term "zero-shot" led to the impression that our model is trained without any anomalies (i.e., unsupervised training on normal data only).
> However, in our setting, **TimeRCD is pre-trained on a large-scale synthetic dataset that explicitly contains labeled anomalies.**
> *   **Pre-training Phase:** The model learns to detect anomalies by seeing millions of diverse, synthetically generated anomalous patterns and their corresponding labels.
> *   **Zero-Shot Inference:** The "zero-shot" capability refers to the fact that after this pre-training, the model is evaluated on **real-world datasets** (e.g., UCR, SWaT) without any further training or fine-tuning on those target domains.
>
> **2. Hyperparameter Search Strategy**
> Given this setup, our hyperparameter search (e.g., for window size) follows standard practices for pre-trained models:
> *   **Development Set:** We utilize a held-out validation subset of our synthetic data (which contains anomalies) to select architectural hyperparameters.
> *   **Inference Strategy:** For the window size specifically, we treat it as a design choice for covering sufficient context. As shown in our sensitivity analysis (**Appendix F.3**), we found that a window size of 5,000 provides a robust trade-off across most datasets. Since the model is transferable, we do not need to tune this per-dataset for the target domain; instead, we use a unified setting that generalizes well.
>
>
> > 3. Some font sizes in figures are too tiny.
>
> **Response:** Thank you for pointing this out. We increased the font sizes in **Figure 1, Figure 2, Figure 3, Figure 5, and Figure 6** to ensure better readability. Please take a look at our new draft. Should this problem occur, please let us know specifically which figure and which part makes it hard for you to read.
>
>
> > 4. Is the input masked or not? In "Output Projection and Anomaly/Reconstruction Head", the text mentioned "masked portions of the input series" but I don't see this mentioned in any other sections.
>
> **Response:** We apologize for the confusion caused by the omission of masking details in the earlier sections. We clarify the procedure as follows:
>
> *   **Training (Masked):** As formally defined in the revised Section 3 (see our response to Weakness 4), we apply a **patched masking strategy** (ratio=15%, masking continuous segments) to the input during training. This forces the model to learn robust **contextual capabilities** by reconstructing the masked segments via the Reconstruction Head, while simultaneously detecting anomalies via the Anomaly Head.
> *   **Inference (Unmasked):** During inference, **no masking is applied**. The model processes the full, uncorrupted time series, and the Reconstruction Head is discarded. Only the Anomaly Head is used to compute anomaly scores.
>
> We have revised **the section 2.1** of the manuscript to explicitly describe this masking protocol.

---

### Official Review · Reviewer_ZQ5M · 2025-11-01

**Soundness:** 3
**Presentation:** 2
**Contribution:** 3
**Rating:** 6
**Confidence:** 4

**Summary:**

This paper introduces a foundation model for zero-shot time series anomaly detection by leveraging large-scale synthetic data pretraining and a proposed Relative Context Discrepancy methodology. The proposed TimeRCD model achieves top zero-shot results on several datasets.

**Strengths:**

* The paper presents a structural and comprehensive framework for synthetic time series anomaly generation and detection.
* It offers systematic experiments and well-designed ablation studies, which effectively demonstrate the contributions of the proposed components.

**Weaknesses:**

* Incomplete evaluation on benchmarks and scalability analysis.
* The methodology section lacks clarity, especially in defining key concepts such as RCD and the computation of anomaly scores.

Please find the detailed comments in the following section.

**Questions:**

* The computation of the anomaly score from the anomaly head is insufficiently explained. Specifically, the definition and mathematical formulation of Relative Context Discrepancy (RCD) remain unclear. How is the discrepancy quantified and how does it integrate into the final detection objective?
* In Section 2.2, the paper introduces two types of labels - localization and detection. It would be helpful to clarify the role of localization labels in model pretraining and downstream evaluation.
* The authors emphasize the importance of capturing causal dependencies among channels, yet it is unclear how the proposed TimeRCD model models these dependencies. How does the framework evaluate multivariate interactions “in light of their causal dependencies”? More discussion would strengthen this claim.
* The study only utilizes a subset of TSB-AD benchmark datasets rather than the full collection. What is the rationale for this selection? Including the full benchmark would provide a more comprehensive and standardized evaluation.
* The omission of strong statistical baselines (such as Sub-PCA or KMeansAD) weakens the comparative analysis.
* What are the context input window sizes used in Section 3.3, and how is the window length $W$ defined and selected as described in Section 2.1? Additionally, an analysis of the sensitivity of this critical hyperparameter would provide insight into the model’s robustness.
* Including a distributional overview of time series lengths and the number of variates in the pretraining datasets would provide more transparency regarding dataset diversity. Moreover, the work would benefit from a discussion of how flattening multivariate time series affects performance and whether causal modeling across dimensions mitigates this limitation.
* The paper lacks runtime efficiency results for TimeRCD compared with baselines.

---

> ### Author Response · Authors · 2025-11-25
> **Response to Reviewer ZQ5M (1)**
>
> ## Reviewer ZQ5M
> ### **Snapshot of Key Improvements**
> - **Evaluation expansion:** Added **WSD/Stock** plus **Sub-PCA, DCdetector, TFMAE** while preserving **TimeRCD’s 32 first-place / 3 second-place** dominance (Table 1, Appendix G).
> - **Method clarity:** **Section 3** and **Appendix C** now formalize **RCD**, anomaly scoring, masking, and localization usage for direct traceability.
> - **Scalability proof:** **Appendix H** contributes runtime/memory/per-1k-step analysis up to **15k timesteps**, addressing feasibility concerns from **ZQ5M/YVt4**.
> ### Weaknesses
> > W1. Incomplete evaluation on benchmarks and scalability analysis.
>
> **Response:** We thank the reviewer for these constructive suggestions. We have **significantly expanded our evaluation and analysis** to ensure comprehensiveness.
>
> **1. Expanded Benchmarks and Baselines**
> To address the concern about incomplete evaluation, we have made the following major additions:
> *   **Additional Datasets:** We have added two new univariate datasets (**WSD and Stock**) from the TSB-AD benchmark to improve diversity.
> *   **New Baselines:** We have incorporated three strong statistical and deep learning baselines (**Sub-PCA, DCdetector, and TFMAE**) in the full-shot setting.
> *   **Rigorous Re-evaluation:** We have **re-run all baselines using standardized metrics (including F1-T)** to ensure fair comparison.
>
> Despite these additions, **TimeRCD maintains state-of-the-art performance (ranking 1st/2nd in zero-shot settings)**. Please refer to our response to **Weakness 3 & 4** for details.
>
> **2. Scalability Analysis**
> We have added a comprehensive runtime efficiency analysis (detailed in **Weakness 6** and **Appendix H**).
> *   **Inference Speed:** We have benchmarked TimeRCD against TimeMOE and DADA, showing it is **12.6× to 95.2× faster** for 10k sequences.
> *   **Long-Context Scalability:** We have verified scalability up to **15,000 timesteps**, demonstrating sub-linear time complexity and stable efficiency (~5-6ms per 1k steps).
>
>
> > W2. The methodology section lacks clarity, especially in defining key concepts such as RCD and the computation of anomaly scores.
>
> **Response:** We appreciate the reviewer pointing out the need for greater clarity in our methodology. We have **extensively revised the description of Relative Context Discrepancy (RCD) and the anomaly scoring mechanism**.
> *   **RCD Definition:** We have clarified that RCD measures the inconsistency of a token's relationships with its context, rather than its absolute value.
> *   **Mathematical Formulation:** We have provided a formal definition of how RCD is quantified via self-attention (using pairwise interaction $f$ and aggregation $g$).
> *   **Anomaly Scoring:** We have detailed how the Anomaly Head maps these relational embeddings to probability scores.
>
> Please see our detailed response to **Question 1** below, where we provide the **full mathematical derivation and explanation (added to Appendix C)**.

---

> ### Author Response · Authors · 2025-11-25
> **Response to Reviewer ZQ5M (2)**
>
> ### Questions
>
> > Q1. The computation of the anomaly score from the anomaly head is insufficiently explained. Specifically, the definition and mathematical formulation of Relative Context Discrepancy (RCD) remain unclear. How is the discrepancy quantified and how does it integrate into the final detection objective?
>
> **Response:** We thank the reviewer for this important observation. Considering mathematical cohesiveness, we first clarify the conceptual and formal definition of Relative Context Discrepancy (RCD), then explain how this discrepancy is instantiated through self-attention, and finally describe how the anomaly score and training objective are computed.
>
> ## Definition of RCD
> Traditional anomaly-detection methods often evaluate each data point in isolation, either by measuring its deviation from expected values or by assessing reconstruction errors. **Relative Context Discrepancy (RCD)** frames the problem differently. Instead of focusing on the individual value of a token, it evaluates whether the *relationships* a token forms with its surrounding context remain consistent with normal relational patterns. Under this view, an anomaly emerges as a disruption in these relational structures, thus RCD focuses on relational inconsistency rather than absolute deviation.
>
> ## Mathematical Formulation of RCD
> Consider a time-series sequence
> $$
> X_1, X_2, \ldots, X_T,
> $$
> where each $X_t$ denotes the observation at time $t$. RCD does not describe $X_t$ by its value alone but by how it interacts with all other tokens in the sequence.
>
> To formalize this, we introduce two functions. The first, $f(X_j, X_t)$, is a pairwise interaction function representing how token $X_j$ contributes to the interpretation of token $X_t$. This function captures the “relative” aspect of RCD by explicitly modeling token-to-token relationships:
> $$
> f(X_j, X_t).
> $$
>
> However, anomalies arise not from one unusual interaction but from an *unusual pattern* of interactions. This motivates the aggregation function $g$, which synthesizes all pairwise contributions into a contextual embedding for token $t$:
> $$
> \text{embedding}_{X_t}
> = g\big(f(X_1, X_t), f(X_2, X_t), \ldots, f(X_T, X_t)\big).
> $$
>
> This embedding summarizes the relational profile that determines how token $t$ fits into its broader temporal context. Under this formulation, an anomaly corresponds to an embedding that deviates from embeddings produced under normal relational patterns.
>
> In this way, the combination of $f$ and $g$ operationalizes the central RCD principle: **the meaning of a token arises from its relationships with the rest of the sequence.**
>
> ## How TimeRCD Quantifies RCD Using Self-Attention
> This RCD formulation aligns naturally with the mechanism of **self-attention**. In a Transformer, each token $X_t$ produces a query vector:
> $$
> q_t = X_t W^Q,
> $$
> while each token $X_j$ generates
> $$
> k_j = X_j W^K, \qquad v_j = X_j W^V.
> $$
>
> The pairwise interaction between tokens $t$ and $j$ is expressed through the scaled dot-product score:
> $$
> s_{tj} = \frac{q_t \cdot k_j}{\sqrt{d_k}}.
> $$
>
> The corresponding unnormalized contribution is precisely the form of $f$:
> $$
> f(X_j, X_t) = v_j \, \exp(s_{tj}).
> $$
>
> Self-attention aggregates these contributions using softmax normalization, which serves as the instantiation of the aggregation function $g$:
> $$
> \text{embedding}_{X_t}
> = \frac{\sum_j v_j \exp\!\left(\frac{q_t \cdot k_j}{\sqrt{d_k}}\right)}
>        {\sum_j \exp\!\left(\frac{q_t \cdot k_j}{\sqrt{d_k}}\right)}.
> $$
>
> Under RCD, discrepancy corresponds to deviations in the attention pattern of token $X_t$ relative to the relational patterns learned from normal data. When a token is anomalous, its query–key interactions produce an atypical similarity distribution $s_{tj}$, leading to a context embedding that lies outside the manifold of embeddings observed during normal behavior. The anomaly head is trained to map these deviations in the embedding to a high anomalous probability, thereby converting relational irregularity directly into an anomaly score.

---

> ### Author Response · Authors · 2025-11-25
> **Response to Reviewer ZQ5M (3)**
>
> > Q1. The computation of the anomaly score from the anomaly head is insufficiently explained. Specifically, the definition and mathematical formulation of Relative Context Discrepancy (RCD) remain unclear. How is the discrepancy quantified and how does it integrate into the final detection objective?
>
> ## Anomaly Head, Reconstruction Head, and Objective
> TimeRCD employs two heads during training. The anomaly head applies a lightweight MLP to each embedding and outputs logits for the normal and anomalous classes:
> $$
> \text{logits}\_{x_t} = \text{AnomalyHead}(\text{embedding}\_{X_t}),
> $$
> and the anomaly score is defined as the softmax probability of the anomalous class:
> $$
> \text{AnomalyScore}\_{x_t}
> = \text{softmax}(\text{logits}\_{x_t})[1].
> $$
>
> The reconstruction head is a 3-layer MLP that predicts a scalar reconstruction for each timestep:
> $$
> \widehat{x}\_t = \text{ReconstructionHead}(\text{embedding}\_{X\_t}).
> $$
>
> The total loss combines a cross-entropy loss for anomaly prediction and an MSE loss for reconstruction:
> $$
> L\_{\text{total}} =
> \lambda\_{\text{anomaly}} \cdot
> \text{CrossEntropy}(\text{logits}\_{x_t}, y_t)
> +
> \lambda\_{\text{recon}} \cdot
> \text{MSE}(\widehat{x}_t, x_t),
> $$
> where both weighting coefficients are set to 1 in our experiments, $y_t$ and $x_t$ is the label and value at the timestamp $t$ respectively.
>
> This completes the explanation of how RCD is defined and formalized, how it is realized through self-attention, and how it integrates into TimeRCD’s anomaly-scoring and training framework. We believe this provides a complete clarification of how RCD is defined, quantified, and used within TimeRCD.
>
> We have added this response to the **Appendix C**

---

> ### Author Response · Authors · 2025-11-25
> **Response to Reviewer ZQ5M (4)**
>
> > Q2. In Section 2.2, the paper introduces two types of labels - localization and detection. It would be helpful to clarify the role of localization labels in model pretraining and downstream evaluation.
>
> **Response:**
> Thank you for the opportunity to clarify. We distinguish between two types of "localization":
>
> 1.  **Temporal Localization:** Our pretraining and evaluation datasets provide labels at the **timestep level** (i.e., which time steps are anomalous). Our model is trained to predict these per-timestep labels via the Anomaly Head.
> 2.  **Feature Localization:** The public benchmarks we use (and our synthetic labels) typically do **not** specify which specific dimension (variable) is anomalous at a given time step—they only indicate that *an* anomaly occurred at time $t$.
>
> We are sorry that our previous manuscript writing about this part is a bit confusing. Actually, our "localization labels" refer strictly to **temporal** localization. We have updated **Section 2.2** to explicitly state that we only use and evaluate **temporal localization labels**.
>
> > Q3. The authors emphasize the importance of capturing causal dependencies among channels, yet it is unclear how the proposed TimeRCD model models these dependencies. How does the framework evaluate multivariate interactions “in light of their causal dependencies”? More discussion would strengthen this claim.
>
> **Response:** We thank the reviewer for seeking clarification on this key aspect. We acknowledge that the term "causal dependencies" might have been ambiguous. We clarify that our framework **leverages causal dependencies primarily for data generation rather than explicit model inference**.
>
> **1. Role in Data Synthesis**
> Generating realistic multivariate time series with coherent inter-variable correlations is notoriously difficult. To solve this, we employ **Structural Causal Models (SCM)** combined with ODE-based dynamics as a generative backbone.
> *   Instead of heuristically imposing correlations, we define a causal graph where variables influence each other through differential equations.
> *   This approach naturally induces complex, time-delayed dependencies and correlations that mimic real-world physical systems (e.g., a spike in variable A causing a delayed drop in variable B).
> *   Therefore, "causal dependencies" here refers to the **generative mechanism** that ensures our synthetic training data possesses realistic multivariate structures.
>
> **2. Modeling via Attention**
> During the modeling phase, TimeRCD does not explicitly perform causal discovery or inference. Instead, it relies on the **Any-Variate Attention** mechanism to capture the **correlations** resulting from these underlying causal processes.
> *   By allowing every variable token to attend to every other variable token, the model learns to recognize the *effects* of these causal interactions (i.e., the observable dependencies) without needing to reconstruct the causal graph itself.
> *   Our empirical results suggest that for the purpose of anomaly detection, capturing these rich correlations is sufficient to identify deviations from normal system behavior.

---

> ### Author Response · Authors · 2025-11-25
> **Response to Reviewer ZQ5M (5)**
>
> > Q4. The study only utilizes a subset of TSB-AD benchmark datasets rather than the full collection. What is the rationale for this selection? Including the full benchmark would provide a more comprehensive and standardized evaluation.
>
> **Response:** We thank the reviewer for this suggestion regarding evaluation comprehensiveness.
>
>
> **1. Rationale for Dataset Selection**
>
> Our selection strategy prioritize representativeness and non-redundancy: we include all the most widely-used multivariate datasets (SWaT, PSM, etc.) and a diverse subset of univariate datasets. This is driven by two key factors:
>
> *   **Avoiding Data Redundancy:** A significant portion (~50%) of the "univariate" TSB-AD benchmark consists of decomposed multivariate datasets (where individual channels of a multivariate system are treated as independent univariate series). Treating these as independent samples introduces **bias** because the "univariate" anomalies often share the same underlying root cause and temporal patterns as their siblings, effectively inflating performance metrics by rewarding the model multiple times for detecting the same event type [1]. To ensure a rigorous evaluation, we filtered out these decomposed series, focusing on **genuinely univariate** datasets and testing multivariate datasets in their original form.
> *   **Metric Standardization & Computational Cost:** Many existing baselines in the original TSB-AD paper rely on metrics that do not fully capture temporal alignment. To ensure a rigorous comparison, we adopt strictly standardized metrics, including **Temporal F1 (F1-T)** [2], which evaluates detection performance by considering the temporal context of anomalies and addressing issues like overlapping predictions. Since most original baselines are not evaluated on F1-T, we have to re-run all of them from scratch. Given the high computational cost of this re-evaluation across the entire collection, we select a representative subset.
>
> **2. Expanded Evaluation in the Revised Version**
>
> We acknowledge that broader coverage is beneficial. To address this, we have added **two additional univariate datasets (WSD and Stock)** from TSB-AD to our evaluation suite. As shown in the updated **Table 1**, TimeRCD maintains state-of-the-art performance on these new datasets, reinforcing its generalizability.
>
> ---
> **Reference**:
> [1] R. Wu and E. J. Keogh, "Current Time Series Anomaly Detection Benchmarks are Flawed and are Creating the Illusion of Progress (Extended Abstract)," 2022 IEEE 38th International Conference on Data Engineering (ICDE), Kuala Lumpur, Malaysia, 2022, pp. 1479-1480, doi: 10.1109/ICDE53745.2022.00116.
>
> [2] M. Saquib Sarfraz, Mei-Yen Chen, Lukas Layer, Kunyu Peng, and Marios Koulakis. 2024. Position: quo vadis, unsupervised time series anomaly detection? In Proceedings of the 41st International Conference on Machine Learning (ICML'24), Vol. 235. JMLR.org, Article 1771, 43461–43476.

---

> ### Author Response · Authors · 2025-11-25
> **Response to Reviewer ZQ5M (6)**
>
> > Q5. The omission of strong statistical baselines (such as Sub-PCA or KMeansAD) weakens the comparative analysis.
>
> **Response:** Thank you for the suggsetion. Due to time constraints for rerunning all of the datasets, we might have missed some baselines. Following your suggestion, we have added Sub-PCA, DCdetector, and TFMAE as additional new baselines in the full-shot settings.
>
> The result are shown in Table 1 among these newest added baselines, Sub-PCA has an impressive performance where it gets 7 times for the best and 20 times for second best for full-shot setting. These results are to some degree consistent with the result of the paper [1]. However, TimeRCD remains the top performer where it achieves **32 times best and 3 times second best** without including any training datasets.
>
>
> Additionally, we compute the statistical ranking for each experimental setting. Model comparisons are conducted using the Friedman test [3], followed by a posthoc Nemenyi test [2], a widely used statistical approach for comparing multiple algorithms across datasets. The results are summarized in the table below
>
>
> | Scenario | Affiliation-F | F1_T | Standard-F1 | VUS-PR | Avg Rank |
> |:---|:---:|:---:|:---:|:---:|:---:|
> | **Zero-shot Univariate** | 2nd (MR=2.73, M=0.844) | 1st (MR=2.45, M=0.414) | 1st (MR=2.64, M=0.393) | 1st (MR=2.27, MD=0.243) | 1.8 / 7 |
> | **Zero-shot Multivariate** | 1st (MR=1.2, M=0.829) | 1st (MR=1.2, M=0.397) | 1st (MR=1.2, M=0.322) | 1st (MR=1.4, M=0.233) | 1.2 / 7 |
> | **Full-shot Univariate** | 1st (MR=1.82, M=0.844) | 1st (MR=2.27, MD=0.350) | 1st (MR=2.64, MD=0.320) | 1st (MR=2.55, MD=0.243) | 2.3 / 9 |
> | **Full-shot Multivariate** | 4th (MR=3.2, MD=0.816) | 4th (MR=3.2, M=0.397) | 4th (MR=4.4, M=0.322) | 4th (MR=3.6, M=0.233) | 3.6 / 9 |
>
> *Legend: MR = Mean Rank | M = Mean | MD = Median, the lower the rank the better*
>
> We **have updated the main Table 1** and also the details of statistical ranking can be **found in the Appendix G**.
>
> ---
> **Reference**
>
> [1] Qinghua Liu and John Paparrizos. 2024. The elephant in the room: towards a reliable time-series anomaly detection benchmark. In Proceedings of the 38th International Conference on Neural Information Processing Systems (NIPS '24), Vol. 37. Curran Associates Inc., Red Hook, NY, USA, Article 3437, 108231–108261.
>
> [2] Nemenyi, P. B. (1963). Distribution-free multiple comparisons. Princeton University.
>
> [3] Friedman, M. (1937). The use of ranks to avoid the assumption of normality implicit in the analysis of variance. Journal of the american statistical association, 32(200), 675-701.
>
> > Q6. What are the context input window sizes used in Section 3.3, and how is the window length defined and selected as described in Section 2.1? Additionally, an analysis of the sensitivity of this critical hyperparameter would provide insight into the model’s robustness.
>
> **Response:** We thank the reviewer for this insightful question regarding hyperparameter selection and robustness.
>
> **1. Experimental Settings**
> For the main results reported in Section 3.3, we standardized the context window size at **5,000** timesteps. This choice was made to balance long-context capability with computational efficiency across a wide range of benchmarks.
>
> **2. Window Selection in Training (Section 2.1)**
> Our training strategy emphasizes diversity to ensure zero-shot robustness. As detailed in **Figure 8 of Appendix E.5 (Supplementary Material)**, our synthetic dataset comprises a vast collection of time series with widely varying lengths, uniformly distributed from **100 to 10,000** timesteps. During pre-training, the model is exposed to this full spectrum of lengths, learning to adapt its attention mechanism to both short and long contexts dynamically.
>
> **3. Sensitivity Analysis**
> We agree that analyzing the sensitivity of the window size is crucial. We have conducted a detailed study in **Appendix F.3**, evaluating performance across window sizes ranging from 1k to 13k.
> *   **Findings:** The results demonstrate that TimeRCD is generally robust. Performance on datasets with long-term dependencies (e.g., Power, SWaT) improves with larger windows, while datasets with shorter intrinsic patterns (e.g., YAHOO) show stable performance once the window covers the relevant context. This confirms that our model effectively utilizes available context without overfitting to a specific length.

---

> ### Author Response · Authors · 2025-11-25
> **Response to Reviewer ZQ5M (7)**
>
> > Q7. Including a distributional overview of time series lengths and the number of variates in the pretraining datasets would provide more transparency regarding dataset diversity. Moreover, the work would benefit from a discussion of how flattening multivariate time series affects performance and whether causal modeling across dimensions mitigates this limitation.
> **Response:** Thank you for highlighting this important point. We appreciate the opportunity to clarify both dataset diversity and our design choices for handling multivariate dependencies.
>
> ---
>
> ### **1. Dataset Diversity**
>
> We apologize that this was not sufficiently emphasized in the main text. **We have already provided the length distribution in Figure 8 (Appendix E.5)**, and in the revised version we explicitly reference this figure and summarize the variate-dimension distribution in the main text.
>
> Regarding dimensionality, our pretraining corpus contains time series with **1, 8, 12, 16, and 20 variables**, ensuring robustness across heterogeneous multivariate settings. (Appendix E.5 has been updated accordingly.)
>
> ---
>
> ### **2. Impact of Flattening Multivariate Time Series**
>
> Flattening multivariate time series is a **widely adopted design choice** in recent time-series foundation models such as Moirai and Chronos. In our framework, a multivariate series $\mathbf{X}\_t \in \mathbb{R}^{D}=[X_{t1},\ldots,X_{tD}], t= 1,2,\ldots$ is flattened by treating each variable as a separate token and concatenating them $\text{Seq} = [\ldots,X_{(t-1)1}, X_{(t-1)2},\ldots, X_{(t-1)D}, X_{t1}, X_{t2},\ldots, X_{tD}, \ldots]$. However, flattening is often criticized for weakening the modeling of **cross-variable dependencies**, which are crucial for detecting anomalies arising from **correlation shifts** rather than single-variable deviations [1–3]. To alleviate it, traditional multivariate anomaly detection methods address this using **fixed architectures** (e.g., GNNs with $D$ nodes) that encode static inter-variable relationships for cross-variable dependence modeling.
>
> #### **Constraint in Foundation Models**
>
> Such methods assume a **fixed and known $D$** at training time. In contrast, a time-series **foundation model**, especially a zero-shot model, must generalize to **arbitrary and previously unseen variate dimensions** at inference time (e.g., trained on 10-dimensional series but tested on 50-dimensional ones) [4,5]. This prevents pre-allocating dimension-specific parameters or learning rigid dependency graphs.
>
> #### **Our Strategy: Any-Variate Attention**
>
> To preserve multivariate structure without requiring a fixed dimensionality, we adopt the **Any-Variate Attention** mechanism introduced in Moirai [4,6]:
>
> - **Solution:** Represent each variate as a unique token enriched with **learnable variate embeddings** and **binary attention biases**, preserving its identity despite flattening.
> - **Effect:** Self-attention operates **jointly across time and feature dimensions**. Even with flattened input, attention heads compute **content-dependent correlation weights** between $(\text{Var}_i, \text{Var}_j)$, allowing flexible modeling of multivariate dependencies without assuming a fixed $D$.
>
> ---
>
> ### **3. Feasibility of Causal Modeling Across Variates**
>
> While causal modeling may theoretically provide more precise relational structure than correlation-based attention, applying causal discovery in a **zero-shot foundation model** setting is currently impractical.
>
> Causal discovery methods rely on assumptions such as sparse causal graphs, identifiable structure, or sufficient temporal variation—conditions often violated in diverse, noisy real-world datasets [7,8]. In addition, accurate temporal causal inference frequently requires **interventions or rich task-specific information**, which are unavailable in zero-shot settings [9].
>
> Applying causal modeling without these prerequisites risks generating unstable or unreliable results [7]. Therefore, in this work we currently focus on robust **correlation modeling** via attention, which captures dependencies implicitly without depending on fragile causal assumptions. Nonetheless, we agree that integrating causal structure learning is promising and view extending TimeRCD with causal modeling as valuable future work.ion of Temporal Causal Representation with Instantaneous Dependence.* ICLR 2025.

---

> ### Author Response · Authors · 2025-11-25
> **Response to Reviewer ZQ5M (8)**
>
> ---
>
> ### **References**
>
> [1] Li, Z. et al. (2021). *InterFusion: Multivariate Time Series Anomaly Detection and Interpretation.* KDD 2021.
> [2] Wei, D. et al. (2024). *An anomaly detection model for multivariate time series with anomaly perception.* PeerJ Computer Science.
> [3] Xie, Y. et al. (2025). *Multivariate Time Series Anomaly Detection by Capturing Coarse-Grained Intra- and Inter-Variate Dependencies.* WWW 2025.
> [4] Woo, G. et al. (2024). *Unified Training of Universal Time Series Forecasting Transformers (Moirai).* ICML 2024.
> [5] Liu, X. et al. (2024). *Moirai-MoE: Empowering Time Series Foundation Models with Sparse Mixture of Experts.* arXiv:2410.10469.
> [6] Roque, L. & Guedes, R. (2024). *Moirai: Time Series Foundation Models for Universal Forecasting.* ZAAI Blog / Towards Data Science.
> [7] Yang, W., Zhang, K., & Hoi, S. C. H. (2022). *Causality-Based Multivariate Time Series Anomaly Detection.* arXiv:2206.15033.
> [8] Schwab, J. et al. (2024). *Leveraging Causal Information for Multivariate Time Series Anomaly Detection.* DX 2024.
> [9] Li, H. et al. (2025). *On the Identification of Temporal Causal Representation with Instantaneous Dependence.* ICLR 2025.

---

> ### Author Response · Authors · 2025-11-25
> **Response to Reviewer ZQ5M (9)**
>
> > Q8. The paper lacks runtime efficiency results for TimeRCD compared with baselines.
>
> **Response:** We appreciate the reviewer's suggestion to include a runtime efficiency analysis, as computational feasibility is critical for deploying foundation models. To address this, we have conducted extensive profiling of TimeRCD against state-of-the-art baselines (TimeMOE, DADA) and extended our analysis to sequence lengths beyond the reviewer's requested range to rigorously verify scalability.
>
> **Key Findings:**
> 1.  **Feasibility:** TimeRCD processes 10,000 timesteps in just **21.8 ms**, which is **12.6× faster** than TimeMOE and **95.2× faster** than DADA.
> 2.  **Scalability:** Even at 15,000 timesteps, inference completes in just **86 ms**, supporting real-time monitoring.
> 3.  **Efficiency:** The model exhibits sub-linear time complexity, avoiding the quadratic bottleneck of standard Transformers.
>
>
> ## Inference Time (1k–10k Timesteps)
>
> All models are profiled on a single GPU (Nvidia A100 80GB) using isolated processes and a global warmup phase to reduce variability. The following table reports end-to-end inference latency as a function of sequence length.
>
> ### **Table: Inference Time (ms)**
> | **Model**     | **Seq = 1K** | **Seq = 2K** | **Seq = 5K** | **Seq = 10K** |
> |---------------|--------------|--------------|--------------|---------------|
> | **TimeRCD**   | **8.0**      | **9.2**      | **12.2**     | **21.8**       |
> | TimeMOE       | 36.5         | 66.7         | 130.8        | 274.2          |
> | DADA          | 207.6        | 423.1        | 1042.3       | 2074.0         |
>
> Across all lengths, TimeRCD exhibits much slower growth in latency compared to TimeMOE and DADA. This behavior is primarily attributed to its **patch-based processing** (patch size = 16), which reduces computational burden for long sequences while preserving temporal coverage.
>
> ---
>
> ## Peak Memory Usage and Parameter Counts
>
> ### **Table: Peak Memory Usage (MB) and Parameters**
> | **Model**     | **Seq = 1K** | **Seq = 2K** | **Seq = 5K** | **Seq = 10K** | **Parameters** |
> |---------------|--------------|--------------|--------------|---------------|----------------|
> | **TimeRCD**   | **295**      | **434**      | **858**      | **1561**      | **35.5M**      |
> | TimeMOE       | 462          | 462          | 462          | 462           | 113.4M         |
> | DADA          | 17.8         | 17.8         | 17.8         | 17.8          | 1.8M           |
>
> TimeMOE and DADA show nearly constant memory usage with respect to sequence length due to **fixed-size sliding windows**, which limit effective context. In contrast, TimeRCD processes the **full sequence**, naturally increasing memory consumption for longer inputs. Nevertheless, the memory requirements remain within the capabilities of common consumer GPUs.
>
> ---
>
> ## Scalability to Long Contexts (Up to 15k Timesteps)
>
> To further characterize long-context behavior, TimeRCD is evaluated on sequences up to 15,000 timesteps. The table below reports end-to-end inference latency, peak memory usage, and an efficiency measure in milliseconds per 1,000 timesteps.
>
> ### **Table: Long-Context Scalability**
> | **Sequence Length** | **Inference (ms)** | **Memory (MB)** | **Efficiency (ms / 1k steps)** |
> |---------------------|---------------------|------------------|---------------------------------|
> | 1,000               | 14.37               | 295              | 14.37                           |
> | 5,000               | 28.39               | 858              | 5.68                            |
> | 10,000              | 51.38               | 1561             | 5.14                            |
> | **15,000**          | **86.22**           | **2230**         | **5.75**                        |
>
> The efficiency metric decreases substantially from 1,000 to 5,000 timesteps and stabilizes around **5–6 ms per 1,000 timesteps** for longer sequences. This demonstrates that fixed overheads are amortized with increasing length, eventually yielding stable per-timestep cost.
>
> We have added this section to **the Appendix H.**

---

> > ### Comment · Reviewer_ZQ5M · 2025-11-27
> >
> > Thank you for the detailed response! The authors have largely addressed my concerns, and I will maintain my positive assessment.

---

### Author Response · Authors · 2025-11-28
**General Response**

We thank the reviewers for their time. We are particularly encouraged by the **LPfi (Score 8)** and **ZQ5M (Score 6)** ratings, which recognize **TimeRCD** as a *"comprehensive framework"* with *"high originality"* that successfully establishes a zero-shot foundation model for anomaly detection.

Given the **divergence in scores (ranging from 2 to 8)**, we believe there is a misunderstanding from the negative reviewer regarding the scope and novelty of our contribution compared to the strong endorsement from the domain experts. We address this below and provide explicit pointers to the relevant sections, questions, and weaknesses.

---

#### **1. Addressing Constructive Feedback**
The positive reviewers (**ZQ5M, YVt4, LPfi**) raised concrete points regarding evaluation rigor, clarity, and efficiency, which we have incorporated into the revision as follows:

* **Expanded baselines & rigor (ZQ5M–W1/W4/W5, YVt4–W3):** Following the suggestions of **ZQ5M** and **YVt4**, we have added **DCdetector**, **TFMAE**, and **Sub-PCA** as additional full-shot baselines and extended the evaluation to **WSD** and **Stock** (see updated **Table 1** and **Appendix G**).
    * *Result:* As detailed in our response to **ZQ5M–W1/W5** and **YVt4–W3**, **TimeRCD** achieves **32 first-place** and **3 second-place** finishes across all datasets and metrics, consistently outperforming both zero-shot and fully supervised SOTA models.
* **Methodological clarifications (ZQ5M–W2/Q1, LPfi–W1/W3, LPfi–Q4/Q5):** We have revised **Section 3** and **Appendix C** to formally define Relative Context Discrepancy (RCD), give the explicit anomaly-score formulation, and describe the dual-objective training (masked reconstruction + anomaly head). We also clarify localization labels and the masking schedule in response to **ZQ5M–Q2** and **LPfi–Q4**.
* **Efficiency & scalability (ZQ5M–W6, YVt4–W2, ZQ5M–Q6/Q7):** We added **Appendix H** to provide the requested runtime profiling (1k–15k context lengths), peak memory, and per-1k-step efficiency, and summarize these results in the response to **ZQ5M–W6** and **YVt4–W2**.

---

#### **2. Clarifying Misinterpretations in the Negative Review**
We respectfully point out that the concerns raised by the negative reviewer (**eExF**, Score 2) appear to stem from **assumptions that are not aligned with the evidence presented** in our updated experiments and from an under-specification of the core technical contribution.

* **On the alleged "performance gap" to DADA (eExF–W3):**
    * *Claim:* **eExF** suggests that the performance of TimeRCD is notably weaker than DADA.
    * *Clarification:* Our extended experiments, detailed in the responses to **ZQ5M–W5**, **YVt4–W3**, and **eExF–W2/W3**, together with the ranking tables in **Appendix G**, show that **TimeRCD attains 32 first-place and 3 second-place ranks**, while DADA attains substantially fewer wins. **TimeRCD also remains competitive against fully supervised models** such as Anomaly Transformer. Thus, the updated results indicate that TimeRCD is at least on par with, and often stronger than, DADA across benchmarks.

* **On the perceived "lack of novelty" (eExF–W1):**
    * *Claim:* **eExF** characterizes our approach as merely “looking at adjacent time windows.”
    * *Clarification:* As emphasized in our responses to **eExF–W1** and **LPfi–W1**, the central contribution is not windowing per se, but the **formalization of RCD as a self-supervised pre-training objective** coupled with a **generative causal synthesis engine** that enables synthetic-only zero-shot pretraining. This framework-level innovation is explicitly recognized by **LPfi (Score 8)** and **ZQ5M (Score 6)**, who describe TimeRCD as a “highly original” and “comprehensive” foundation model, going beyond standard window-based detectors.

* **On the lack of engagement with ablations and design motivation (eExF–W1/W2):**
    * Several of the negative comments question the motivation for the dual-objective design while not referencing the ablation evidence already provided (and now expanded) in **Section 4.3** and **Appendix F.6**. These ablations show that removing masked reconstruction or the anomaly head leads to substantial performance drops, supporting the design choices that other reviewers found well-motivated.

---

#### **3. Conclusion**
We believe the **LPfi (Score 8)** and **ZQ5M (Score 6)** ratings accurately reflect the technical contributions and empirical strength of this work, as they explicitly account for the **synthetic-only pretraining paradigm, the RCD objective, and the comprehensive evaluation**. We hope that, in light of the clarified empirical results and methodological motivations (documented throughout our responses to **ZQ5M, YVt4, LPfi, and eExF** and in **Appendix C/F/G/H**), the AC and reviewers will view the divergent assessment in a broader context.

---

### Meta-Review · Area_Chair_7KyP · 2025-12-21

**Summary:**

The work explores an important emerging problem in time series anomaly detection (TSAD), aiming to build foundation models specifically for TSAD. It introduces a new approach called TimeRCD, which comprises two key components, large-scale synthetic data pretraining and Relative Context Discrepancy. TimeRCD is then validated on 16 public TSAD datasets in comparison to both zero-shot and full-shot TSAD models.

**Reviewer Concerns:**

The reviewers acknowledge the potential usefulness of TimeRCD but raise a number of major concerns about aspects such as novelty, model design, clarity, and empirical justification.

- **Novelty.** One major concern is on the novelty of Relative Context Discrepancy, as this concept/idea has been extensively explored in conventional AD. Besides, it seems that a similar idea has also been explored in recent work on building general AD models, though those work are not done on time series data. The authors clarifies its novelty in utilizing this idea to train foundation models for AD. The authors also emphasize the other component, large-scale synthetic data pretraining, highlighting the synthetic data-only pretraining. Overall, the AC agrees with the authors on the novel use of RCD in training TSAD foundation models, but the work lacks discussion on its connections and innovation w.r.t. conventional TSAD methods and other similar general/generalist AD methods. In addition, synthetic data-only pretraining sounds interesting, but it discards large-scale publicly available TS/TSAD datasets in the pretraining; it is unclear how those synthetic data may be synergized with real-world data, as was done in DADA, but such experiments are not found in the paper. From these perspectives, the AC finds that the justification of both major components is not strong enough.

- **Model design and clarity.** There are concerns on the clarity and/or motivation behind RCD, the anomaly score computation, and the role of components such as the anomaly head and the reconstruction objective. The rebuttal clarifies these issues properly.

- **Empirical justification.**  Reviewers consistently note that the experimental evaluation is incomplete, including the use of only a subset of the TSB-AD benchmark, omission of strong statistical and modern deep baselines (e.g., Sub-PCA, DCdetector, TFMAE), and unclear exclusion of some comparative results. The authors address these concerns with the inclusion of two more datasets and the mentioned baselines. This helps address the concern. Reviewers also raised the absence of statistical significance testing, which was addressed by the inclusion of Posterior Monte Carlo Sampling-based methods to obtain results of different independent runs and the Friedman-Nemenyi test. The results show that TimeRCD has small variation in its detection, but it does not statistically significant difference to its competing methods according to the Friedman-Nemenyi test. In addition, the ablation study was found incomplete, missing the validation of the joint anomaly detection and data reconstruction optimization. The rebuttal argued the results on the UCR dataset in the supplementary material can address this concern. However, the AC finds that this (the introduction of the anomaly detection head, in addition to the reconstruction head) is one of the most important claims, and believes that a much more comprehensive empirical evaluation on this addition is required. A few other questions the AC has when checking through the paper include: 1) whether DADA and the proposed method are trained with the same scale of generated anomaly data? Is the better performance really due to the proposed AD objective or just simply due to the larger scale of pretraining data? 2) Does the generated anomaly data in the proposed method help improve the performance of DADA? 3) The rebuttal claims the proposed method is architecture-agnostic, generalizable to other architectures, but the work does not provide evidence in this line?
- **Other unsubstantiated claims.** The AC finds it difficult to be convinced on some major claims. For example, “*These weaknesses are further exacerbated in zero-shot settings, where unseen yet normal patterns make reconstruction-based scores particularly unreliable. Our experiments confirm this gap: reconstruction-based foundation models achieve only 15.4% Standard-F1 on contextual anomalies, compared to 82.7% for our RCD approach (Section 3.3.)*”. The claim is on the unreliability of recognizing unseen yet normal patterns in reconstruction-based scores, but the second sentence cites performance in detecting contextual anomalies, making it difficult to connect the claim and the justification.

In summary, the work presents an effective foundation models for TSAD, but it would benefit from another significant revision to clarify and provide rigor and strong justification for the major claims.

**Reviewer Scores:**

The work received four divergent reviews, including an accept (8), a weak accept (6), a weak reject (4), and a reject (2). The AC anticipates that the two positive reviews would retain their positive recommendation, but the two negative reviews would not be convinced to change to positive ratings due to the concerns over the novelty and the ablation study.

---

### Decision · Program_Chairs · 2026-01-26

Reject